


# Estimating background contributions and U.S. anthropogenic enhancements to maximum ozone concentrations in the northern U.S.

David D. Parrish[1,2,3]

[1] Cooperative Institute for Research in Environmental Sciences, University of Colorado, Boulder, USA

[2] NOAA/ESRL Chemical Sciences Division, Boulder, Colorado, USA

[3] David.D.Parrish, LLC, Boulder, Colorado, USA

*Correspondence to*: David D. Parrish (David.D.Parrish@noaa.gov)

**Abstract.** U.S. ambient ozone concentrations have two components: U.S. background ozone and enhancements produced from the country's anthropogenic precursor emissions; only the enhancements effectively respond to national emission controls. We investigate the temporal evolution and spatial variability of the largest ozone concentrations, i.e., those that define the ozone design value (ODV) upon which the National Ambient Air Quality Standard (NAAQS) is based, within the northern tier of U.S. states. We focus on two regions: rural western states, with only small anthropogenic precursor

emissions, and the urbanized northeastern states, which include the New York City urban area, the nation's most populated. The U.S. background ODV (i.e., the ODV remaining if U.S. anthropogenic precursor emissions were reduced to zero) is estimated to vary from 54 to 63 ppb in the rural western states, and to be smaller and nearly constant (45.8 ± 1.7 ppb) throughout the northeastern states. These U.S. background ODVs correspond to 65 to 90% of the 2015 NAAQS of 70 ppb. Over the past two to three decades U.S. emission control efforts have decreased the anthropogenic ODV enhancements at an

approximately exponential rate with an e-folding time constant of ~22 years. These ODV enhancements are small in the rural western states (2.4 ± 1.2 ppb in 2000), with much larger state maximum ODV enhancements (~35-64 ppb in 2000) in the northeastern states. The U.S. background ODV contribution is significantly larger than the present-day ODV enhancements due to photochemical production from U.S. anthropogenic precursor emissions in the urban as well as the rural regions investigated. Forward projections of past trends suggest that average maximum ODVs in northeastern U.S. will drop below

the NAAQS of 70 ppb by about 2021, assuming that the exponential decrease of the ODV enhancements can be maintained and the U.S. background ODV remains constant. This estimate is much more optimistic than in the Los Angeles urban area, where a similar approach estimates ~2050 for the maximum ODV to reach 70 ppb (Parrish et al., 2017). The primary reason for this large difference is the significantly higher U.S. ODV background (62.0 ± 2.0 ppb) estimated for the Los Angeles urban area. The approach used in this work has some unquantified uncertainties that are discussed. Models can also estimate

U.S. background ODVs; some of those results are shown to correlate with the observational estimates derived here ($r^2$ values for different models are ~0.31 to 0.85), but are on average systematically lower by 4 to 12 ppb. Further model improvement



is required until their output can accurately reproduce the time series and variability of observed ODVs, and the uncertainties in the two approaches can be reduced through additional comparisons.

## 1 Introduction

The U.S. has a long-standing air quality problem associated with elevated ozone concentrations (e.g., NRC, 1991).
Fortunately, this situation has been greatly improved over the past 3 to 5 decades, particularly in urban areas. For example, through the 1960s and 1970s the Los Angeles urban area (i.e., California's South Coast Air Basin – SoCAB) endured maximum 1-hr average and maximum daily 8-hr average (MDA8) ozone mixing ratios that exceeded 500 and 300 ppb, respectively (ppb = nmole ozone per mole air) (Parrish and Stockwell, 2015). The National Ambient Air Quality Standard (NAAQS) is based on the ozone design value (ODV), which is defined as the 3-year average of the annual fourth-highest
daily maximum 8-hour average (MDA8) ozone concentration; in 2015 the NAAQS was lowered, now mandating that ODVs not exceed 70 ppb. A fit to the long-term trend of the maximum ODVs recorded in the SoCAB indicates that these highest ozone concentrations decreased from 289 to 102 ppb over the 36-year, 1980 to 2015 period (Parrish et al., 2017). This decrease demonstrates that controls on U.S. ozone precursor emissions have been remarkably effective in reducing maximum ambient ozone concentrations. However, much additional emission reduction efforts are required to reach the
NAAQS of 70 ppb. A critical question has relevance to policy development for managing U.S. ozone concentrations: What is the limit to which ODVs can be reduced by controlling U.S. anthropogenic emissions? One goal of this work is to provide an approximate answer to this question.

Both natural and anthropogenic processes interact to determine the temporal and spatial distribution of surface ozone concentrations in both urban and rural areas. Thus, even if U.S. anthropogenic emissions of ozone precursors were
completely eliminated, ambient ozone concentrations throughout the U.S. would be non-zero due to contributions from natural sources of ozone, enhanced by anthropogenic contributions from other countries. Parrish et al. (2017) estimate that this remaining ODV (denoted as U.S. background ODV) would be 62.0 ± 1.9 ppb in the Los Angeles urban area. This contribution is the limit to which the ODVs can be reduced by U.S. emission controls alone; it is so large that there is little margin for further enhancement of ambient ozone concentrations by photochemical production from U.S. anthropogenic
precursor emissions before the NAAQS of 70 ppb is exceeded. In this paper we extend the analysis of Parrish et al. (2017) to the northern tier of U.S. states, which includes the most populated U.S. urban area (the New York City metropolitan area), as well as vast, sparsely populated regions.

The northern U.S. states can be conveniently divided into four regions (Figure 1): three Pacific Northwest states, three rural western states, three Midwest states, and eight northeastern states. The northeastern states are the most heavily populated,
with the New York City; Boston, Massachusetts; and Providence, Rhode Island urban areas. Two regions are moderately



populated: the Midwest states (with Detroit, Michigan; Minneapolis-St. Paul, Minnesota; and Milwaukee, Wisconsin) and the Pacific Northwest states (with Seattle, Washington and Portland, Oregon). These urban areas all have estimated 2017 populations > 1.5 million. In contrast, the rural western states (Montana, North Dakota and South Dakota) have no cities with populations >260,000. Wyoming could potentially be included with these three rural western states; however,

Wyoming's maximum ODVs occur in winter due to photochemical ozone production within a basin containing intense oil and gas production activities (Oltmans et al., 2014). Since this is a very different situation from that in the other rural states, only the Yellowstone National Park site from this state is considered; it is adjacent to Montana and distant from the oil and gas activities. This site is included because it has a measurement record that has been examined in previous analyses of long-term trends of U.S. background ozone concentrations (e.g., Lin et al., 2017).

The temporal histories of the ambient ozone concentrations measured in these four regions approximately correlate with the magnitude of the urban populations (Figure 2). The largest ODVs have been recorded in the northeastern states, the lowest in the rural western states, with intermediate ODVs in the other two regions. As expected, the ODVs in none of these regions have approached the maximum ODVs recorded in the SoCAB (indicated by blue lines in Figure 2). The analysis that follows in this paper considers only the two more extreme regions – the rural western states, where the average ODVs have remained

approximately constant at relatively small values over the 39 years of measurements, and the northeastern states, where the largest ODVs in the northern U.S. have been recorded.

The northeastern U.S. and the rural western states present dramatic contrasts. Although the eight northeastern states total only ~40% of the land area of the three western states, their population is a factor of ~16 larger. These differences imply a factor of ~40 greater population density in the northeast, despite large rural areas remaining within that region (e.g., parts of

Maine, Vermont and New Hampshire). This comparison emphasizes the concentration of population in urban areas in the northeast, with the 2017 New York City urban area population of > 20 million compared to no cities significantly exceeding 1% of that population in the rural western states. Consistent with this population distribution, the Northeast U.S. has suffered from ozone air pollution for decades (e.g., Wolff and Lioy, 1980). There are no designated ozone nonattainment areas in the western rural states, while, despite substantial improvements in ozone air quality (e.g., Lin et al., 2000), the U.S. EPA's

"Green Book" (https://www.epa.gov/green-book) currently lists five ozone nonattainment areas in the eight northeastern states. The two nonattainment areas designated as "Moderate" - New York-Northern New Jersey-Long Island, NY-NJ-CT and Greater Connecticut, CT - include the New York City urban area and the regions directly downwind. The other three, designated as "Marginal", include southern New Jersey in the Philadelphia-Wilmington-Atlantic City, PA-NJ-MD-DE exceedance area, plus two sparsely populated areas - Dukes County, MA with a population of ~17,000 and Jamestown,

NY with a population of ~95,000. Transport of ozone into these latter two areas is accepted as the dominant cause of the observed high ozone concentrations that led to their designations (Wilcox, 1996).



In this paper we examine the temporal and spatial variability of the highest ozone concentrations (i.e., the ODVs) observed over the past three to four decades in these two contrasting regions of the northern U.S. The analysis is designed to separately quantify the U.S. background ODVs and the enhancements of the ODVs above that background contribution due to U.S. anthropogenic precursor emissions. The U.S. background ODVs will be taken as estimates of the maximum ozone

concentrations that would exist in these regions in the absence of U.S anthropogenic precursor emissions. We also aim to quantify the temporal evolution and spatial variability of the US anthropogenic enhancements of the ODVs, and based on past trends, project the expected time required for the maximum ozone concentrations to decrease to the 70 ppb NAAQS in the northeastern U.S. Photochemical modeling systems are generally utilized for such quantifications and projections (e.g., Dolwick et al., 2015; Emery et al., 2012; and Fiore et al., 2014). However, present model quantifications of U.S. ozone

concentrations have quite large uncertainties (Jaffe et al., 2018; Guo et al., 2018). In contrast, Parrish et al. (2017) present an observational based analysis that utilized the temporal record of measured maximum ODVs in eight air basins in Southern California for these purposes; here we apply this latter approach in the northern U.S. Comparison of these observational based results with those from modeling efforts will help to identify needs for further research.

The analysis approach in this paper relies on differences in the temporal behavior of the U.S. background ODV (assumed

approximately constant) and ODV enhancements due to US anthropogenic precursor emissions (assumed to have decreased exponentially over the past decades in response to U.S emission controls.) One complication arises from previously published studies that have identified a multitude of additional processes that potentially can make systematic contributions on a variety of time scales to the variability of ozone concentrations at U.S. surface sites, including: stratospheric intrusions, which can bring particularly high ozone concentrations to the surface (Langford et al., 2009, 2014; Lin et al., 2012a, 2015);

increasing Asian anthropogenic emissions, which are believed to raise ozone concentrations over the U.S. (Jacob et al., 1999; Lin et al., 2012b); increasing frequency of wildfires, which can produce episodic ozone enhancements (McKeen et al., 2002; Jaffe, 2008, 2013; Pfister et al., 2016); variable meteorological conditions, which can lead to changes in transport patterns (Wang et al., 2016) or changes in the conditions conducive to photochemical ozone production (Shen and Mickley, 2017; Shen et al., 2017), increasing methane, which is argued to increase global ozone concentrations (Fiore et al., 2008, and

references therein); and a warming climate, which has been argued may partially offset air quality improvement from regional emission controls by (Fiore et al., 2015). However, there has been little in the way of systematic, quantitative analysis of the effects of these additional processes on ODVs across the U.S. Figure 2 shows clear overall reductions of ODVs across four decades, consistent with control of U.S. ozone precursor emissions, but there is also significant variability of ODVs about the long-term trends. In this work, we first quantify the U.S. background ODVs and the temporal effects of

US anthropogenic precursor emission reductions, and then discuss the influence of other processes through analysis of the fraction of the ODV variability that cannot be accounted for by the first two factors.



Many papers have investigated U.S. surface ozone trends (see Lin et al., 2017 and references therein). These studies have treated a variety of statistics (medians, means, and various percentiles) to characterize the ozone concentration distribution. It is important to note that all trends investigated in this paper are based on the ODV statistic. The reason for this choice is that the NAAQS is based on the ODV statistic, and so is most relevant for policy considerations. Assuming that the four highest

MDA8 ozone concentrations occur in the warmer half of the year when the highest ozone concentrations generally occur, the ODV corresponds to ~98th percentile of the MDA8 concentrations during the ozone season. As a consequence, the U.S. background ODVs that we discuss are significantly larger than average or median background ozone concentrations that are often discussed in the literature. Given these different choices, care must be taken in comparing trends derived in this work with those from other analyses.

The sources of data and the analysis methods are discussed in the next section, followed by the applications of those methods to quantify the U.S. background ODVs and the enhancements from U.S. anthropogenic ozone precursor emissions in the western rural region (Section 3.1) and the northeastern U.S. (Section 3.2). The larger temporal ODV trends and the greater spatial variation of those trends in the northeastern U.S. provide the basis for the elucidation of several features of regional ozone trends, which are briefly discussed in the Appendix. Section 4 gives a summary of the approach and the results,

discusses implications of those results, and identifies needs for further research.

## 2 Data and Methodology

This work considers Ozone Design Values (ODVs) reported from the beginning of U.S. ozone monitoring in the mid-1970s through 2017 in seventeen northern U.S. states; in-depth analyses focus on three, sparsely populated, western rural states and on eight states in the much more heavily populated northeast U.S. The ODV, the statistic upon which the U.S. NAAQS is

based, is calculated each year for each ozone monitoring station in the U.S. if the measurements achieve the specified completeness criteria. Each year the ODVs recorded at all monitoring stations in the U.S. are added to EPA's AQS data archive (https://www.epa.gov/aqs). The ODVs reported for all years at all sites in all seventeen northern states were downloaded from this archive; only the ODVs marked as valid were retained for analysis. Exceptional events that have concurrence from the U.S. EPA were excluded. Most notably the Connecticut Department of Energy and Environmental

Protection submitted an exceptional event demonstration to the U.S. EPA showing that emissions from the 2016 Fort McMurray wildfire caused elevated ozone levels throughout Connecticut on May 25 and 26, 2016, to which the U.S. EPA concurred.
(http://www.ct.gov/deep/lib/deep/air/ozone/2016_exceptional_event_request/ct_deep_ft._mcmurray_ee_demo_concurrence_letter_and_tsd_7_31_2017.pdf). Table S1 summarizes these archived ODVs for each of the states, including the number of

monitoring sites, the years spanned by the reported ODVs, and their maximum and minimum values. The earliest ODV was reported in 1973 in New York. The number of monitoring sites in the states varies from 83 in Wisconsin to 4 in Rhode Island, with a total of 457 in the seventeen states. The reported ODVs span the range from 169 ppb to 41 ppb. It should be





noted that very few sites have continuous measurements over the complete time spans indicated, and that many sites operated for only short periods. All reported ODVs are included in this analysis, even if only a single ODV was reported for a particular site. It is implicitly assumed that the temporal discontinuities associated with initiation or termination of individual sites does not prevent an accurate quantification of temporal trends of ODVs within the regions selected for

analysis.

An examination of long-term trends of ODVs has been reported for southern California air basins (Parrish et al., 2017) that utilized an exponential function with a constant positive offset (Eq. 1) to quantify the temporal evolution of the ODVs in each air basin:

$$ODV = y_0 + A \exp\{-(year\text{-}2000)/\tau\}. \tag{1}$$

Mathematically, the first term of Eq. 1, $y_0$, is the asymptotic value toward which the ODVs approach as the year of the ODV increases, and the second term is the enhancement of the ODVs above $y_0$, which decreases exponentially with an e-folding time constant of $\tau$ years. Thus, $A$ is the enhancement of the ODVs above $y_0$ in a reference year, defined in this work as the year 2000. Parrish et al. (2017) show that a single value of $\tau = 21.9 \pm 1.2$ years, a single value of $y_0 = 62.0 \pm 1.9$ ppb, and a different value of $A$ in each air basin provided an excellent fit ($r^2 = 0.984$) to the maximum ODVs recorded in six southern

California air basins over the 1980-2015 period (shorter periods ending in 2015 in two of those air basins). In this work Eq. 1 with $\tau$ set to the same value (21.9 years) is used to derive fits to the evolution of ODVs in the northern U.S. states.

A well-established conceptual model (e.g., Parrish et al., 1986) provides qualitative support for the application of Eq. 1. At any given location in the U.S., we can consider ambient ozone concentrations to be composed of two contributions: 1) background ozone and 2) enhancements resulting from ozone produced from photochemical processing of U.S.

anthropogenic emissions of ozone precursors. The first contribution is defined as the ozone that would be present in the absence of U.S. emissions of ozone precursors from anthropogenic sources; this is ozone transported into the U.S., plus that produced over the U.S. from naturally emitted precursors, modified by loss processes. The U.S. Environmental Protection Agency (EPA) has defined this contribution as U.S. background ozone (e.g., Dolwick et al., 2015). The second contribution accounts for ozone produced from U.S. anthropogenic precursor emissions, which can be further divided into two separate

contributions - that transported into a region from upwind U.S. sources, and that produced locally and regionally. We identify the first term of Eq. 1, $y_0$, as an estimate of the ODV that would result from U.S. background ozone alone (i.e., U.S. background ODV), and the second term as an estimate of the enhancement of observed ODVs above $y_0$ due to the sum of the two contributions from U.S. anthropogenic precursor emissions. This interpretation implies that the average of the annual fourth-highest MDA8 ozone concentration in the absence of U.S. anthropogenic emissions would equal $y_0$, and that in the

year 2000 the long-term trend in ODVs is best fit by the sum of $y_0$ plus $A$; hence $A$ is identified as the ODV enhancement





above $y_0$ in the year 2000. It should be noted that in the absence of anthropogenic emissions, the four days of highest MDA8 ozone concentrations that would determine $y_0$ are likely not the same days that determined the actual ODV in any particular year. Indeed Parrish et al. (2017) show that there has been a systematic shift of the days with maximum ozone concentrations to earlier in the year as the U.S. anthropogenic ozone contribution, with a summertime maximum due to its photochemical

source, has decreased, and the U.S. background ozone contribution, which has a springtime maximum, has come to dominate to a progressively greater extent.

Equation 1 implies that the two U.S. anthropogenic contributions have decreased exponentially as emission control efforts reduced ozone produced from all U.S. anthropogenic sources. Associating U.S. background ozone with the parameter $y_0$ is equivalent to extrapolating long-term trends of observed ODVs to the limit of zero U.S. anthropogenic emissions, when the

exponential term in Eq. 1 becomes zero. Parrish et al. (2017) used 36 year ODV records (1980-2015) in southern California to derive a value of $\tau$ = 21.9 years for that region of the country. As will be shown in the following discussion, the ODV records in the northern U.S. do not show consistent decreasing trends over such long time periods. As a consequence it is not possible to precisely extract 3 parameters from the regression fits of the measurement records to Eq. 1. Here we take the value of $\tau$ = 21.9 years derived from California to apply to all of the northern U.S., which implicitly assumes that control

strategies have produced approximately equal relative reductions in anthropogenic ozone enhancements throughout the country. Fixing the value of $\tau$ allows precise determinations of the other two parameters ($y_0$ and $A$) from the fits to Eq. 1.

Two differences between the application here and that of Parrish et al. (2017) should be noted. First, the former work chose 1980 as the reference year, while here we choose the year 2000. The curves derived from the fits to Eq. 1 and the values derived for the $y_0$ parameter do not depend on the choice of reference year, while the values derived for the $A$ parameter do.

Consequently, the $A$ parameters derived here cannot be directly compared with those given for California by Parrish et al. (2017); however, the second term of Eq. 1 with the parameters derived in this work and year set to 1980 does give values that can be directly compared to the $A$ parameter values from that earlier work. Second, the former work considered only the maximum ODV recorded in each year at any of the sites within a given air basin, while here we primarily analyze the ODVs from all sites recorded in a given year in selected regions, although we do also consider the maximum ODVs recorded in

each of the northeastern U.S. states. Since $y_0$ is an estimate of the U.S. background ODV, the value derived for this parameter is expected to be independent of whether the fit of Eq. 1 is made to all ODVs or to only the maximum ODVs in a particular region.

As discussed in the Introduction, the conceptual model utilized for interpretation of Eq. 1 is simplified; it assumes that decreasing U.S. anthropogenic emissions is the only cause of ODV variability at a particular location. Other factors such as

rising anthropogenic emissions in Asia, variable occurrence of wild fires, and interannual meteorological and climate variability can also potentially affect observed ODVs, but they cannot be simply included in Eq. 1. The approach taken here



is to interpret the observed ODVs solely on the basis of Eq. 1, and then to examine the fraction of the ODV variability captured by that interpretation. The remaining fraction of the variability is then attributed to other factors, including those listed above. Here we use two statistics to quantify the fraction of the variability not captured by the fits to Eq. 1. The root-mean-square deviation (RMSD) between the derived fit and the observed ODVs gives an absolute measure (in ppb) of the

ODV variability not captured by Eq. 1. The square of the correlation coefficient ($r^2$) between the observed ODVs and the values derived from the fit to Eq. 1 gives a measure of the fraction of the total variability of the ODVs that is captured by that fit; the difference between unity and that $r^2$ value is then a relative measure (as a fraction) of the ODV variability not captured by Eq. 1. Other factors must then account for the variability quantified by those statistics. In the earlier study of ODVs in southern California air basins (Parrish et al., 2017), the derived $r^2 = 0.984$ and the RMSD ≈ 4 ppb indicate that all

factors not included in Eq. 1 account for no more than 1.6% of the total variability in the basin maximum ODVs analyzed in that work, and contribute a RMSD to those ODVs of no more than ~ 4ppb.

A potential complication in the interpretation of the two terms of Eq. 1 arises if there is a significant fraction of U.S. anthropogenic ozone precursor emissions that has not been reduced by emission controls. Ozone produced from such emissions would not have decreased in the same manner as that produced from most U.S. anthropogenic emissions, which

could raise the derived value of $y_0$, above the actual U.S. background ODV. Parrish et al. (2017) have discussed this issue with regard to the emissions associated with the intense agricultural activity in the Imperial Valley of the Salton Sea air basin, where the derived $y_0$ is higher than in other southern California air basins. The final section of this paper briefly considers the possible impact of this complication in the northeastern U.S. states.

In this work we consistently give 95% confidence limits for derived parameters, unless indicated otherwise. Most of the

analysis in this work is based on non-linear, least-squares regression fits of the archived ODVs to Eq. 1, and interpretation of the derived values for the parameters $y_0$ and $A$. In this interpretation it is important to properly consider the uncertainty of the derived parameter values. We derive the 95% confidence limits from the least-squares fitting routine that have been adjusted to account for the known covariance between the recorded ODVs. Each ODV is a three-year running mean; therefore only every third ODV is independent from the others determined at a given site. Consequently, the number of independent ODVs

in each fit is approximately a factor of three smaller than the number of reported ODVs. Thus, the fitting routine underestimates the true confidence limits of the derived parameters; all derived confidence limits have been increased by a factor of $3^{1/2}$ to account for this covariance, giving more realistic confidence limits. There are additional sources of covariance between the ODVs included in any particular fit. The ODVs from different sites within a region can co-vary due to regionally coherent interannual variability. Also temporal interannual variability may possibly lead to covariance between

ozone concentrations measured in successive years at a site. We are not able to account for the effect of this additional covariance; the derived confidence limits are thus lower limits for the true confidence limits of the derived parameters.



**3 Results**

Here we fit the time series of ODVs from the western rural states (Section 3.1) and the northeastern states (Section 3.2) to Eq. 1 and discuss the results in the context of the conceptual model introduced above. This model considers the recorded ODVs to comprise two contributions: 1) an approximately constant U.S. background ODV (i.e., the ODV that would exist if

U.S. anthropogenic ozone precursor emissions were reduced to zero), which is identified with $y_0$ in Eq. 1, and 2) enhancements due to local and regional photochemical production of ozone from the actual U.S. anthropogenic ozone precursor emissions, which are approximated by the second term in Eq. 1. These photochemical enhancements have been progressively reduced over the past decades by U.S. emission controls. Section 3.2 and the Appendix discuss further details of the spatial and temporal variability of ODVs in the northeastern U.S.

**3.1 ODVs in western rural states**

The sparsely populated, three-state, western rural region generally lies on the Northern U.S. Great Plains downwind of more mountainous terrain to the west. Figure 3 shows a topographical map of the region, with the locations of the ozone monitoring sites indicated. The region gradually slopes to the east and north. With the exception of Yellowstone NP at 2.43 km, all of the monitoring sites lie below 1.55 km elevation.

The histories of the ODVs recorded in the region (Figure 4 with data statistics and fitted curves summarized in Table 1) indicate that throughout the measurement period, there has been little enhancement of ozone concentrations due to anthropogenic influences and little variability due to any cause. The 283 tabulated ODVs recorded over 39 years at 35 sites in the three states average 59.3 ppb with a standard deviation of 3.7 ppb – strong support for the assumption of an approximately constant U.S. background ODV within this region. At the individual sites and within each state the entire

measurement records are all well described by averages with small standard deviations (Table 1): < 3 ppb in Montana and North Dakota, and < 4 ppb in South Dakota, the state whose sampling sites span the largest elevation range (0.34 to 1.55 km). U.S. background ODVs increase with the elevation of the sampling site (e.g., see discussion in Jaffe et al., 2018), so larger variability is expected when the monitoring sites within a state span a larger range of elevations. Fits to Eq. 1 included in Figure 4 show small, statistically significant anthropogenic enhancements of ODVs in Montana and North Dakota, with $A$

values of 2.1 ± 2.1 ppb and 2.5 ± 1.5 ppb, respectively. These fits reduce the variability slightly – the RMSD of all of the data about the fitted curves in Figure 4 is 2.7 ppb, compared to the standard deviation of 3.7 ppb for the average of all ODVs in the region. The derived values for the $y_0$ parameter all lie within a range of 9 ppb, but there are some statistically significant differences, with a maximum in South Dakota (62.7 ± 5.1 ppb), a minimum in Montana (53.7 ± 1.9 ppb) and North Dakota in between (57.0 ± 1.5 ppb). Consistent with the site elevation differences, the average ODV at Yellowstone

NP (elevation 2.43 km) is significantly larger than that at Glacier NP (elevation 0.96 km): 64.0 ± 2.1 and 54.5 ± 1.3 ppb, respectively (where the standard deviation of each data set is indicated); however statistical uncertainty prevents an



attribution of the cause of this difference to either larger $y_0$ values at Yellowstone NP (59.1 ± 6.1 vs. 54.8 ± 2.9 ppb) or larger $A$ values (6.8 ± 8.3 vs. -0.3 ± 3.1 ppb).

### 3.2 ODVs in northeastern states

A topographical map showing the network of ozone monitoring sites in the eight northeastern U.S. states is given in Figure
5. All of the ODVs recorded in four of the eight states are plotted in Figures 6 and 7, along with curves showing fits of Eq. 1 to the ODVs from selected groups of sites over selected time periods. These groups of sites, and similarly selected groups in the other four states, represent different environments within each state. Figures S1-S8 of the Supplement show detailed ODV temporal plots for all eight states with fits of Eq. 1 to the ODVs for all selected groups of sites; the locations of these groups are indicated on maps included in these figures. In all, seventeen groups of sites within the eight states were selected; they are listed in Table 2. Superimposed curves in Figures 6, 7 and S1-S8 indicate the fits of Eq. 1 to the ODVs from all groups of sites; the parameters derived from those fits are included in Table 2. The fits to the ODVs in the rural western states include the full time span of all available results. However in the northeastern U.S., for most of the groups of selected sites, a consistent decrease in ODVs is not defined by the available data over the full measurement record (e.g., the four groups of selected sites in Figures 6 and 7). The strategy utilized here is to fit the ODVs over the time period beginning when a clear, consistent decrease in ODVs is first established, and continuing through 2017, the most recent ODV available. In all cases these fits begin by 2000, with some beginning earlier - either at the start of measurement record, 1990 or 1995 determined by the best, consistent fit to the functional form of Eq. 1.

There are some consistent general features of the ODV time series and the corresponding fits that inform the following analysis:

- Throughout the measurement record, the largest ODVs are found in the states that contain the New York City metropolitan area (New York, New Jersey and southwestern Connecticut), or that lie directly downwind (coastal Connecticut and Long Island, New York). Such sites compose two of the selected groups of sites in New York and Connecticut (see highlighted points in that area in Figure 5), whose ODVs and fits of Eq. 1 are highlighted in Figure 6.
- In several states, the largest ODVs are recorded at coastal sites (i.e., Connecticut, Massachusetts, New Hampshire and Maine in Figures 6, 7, S3, S5, S6 and S8). Two Massachusetts coastal sites are relatively isolated, located on the offshore island of Martha's Vineyard and near the tip of Cape Cod; however, these sites record some of the highest ODVs within that state (see Figure S5). Dukes County, which includes only Martha's Vineyard and nearby smaller islands, with a total population of ~17,000 has been designated as a marginal non-attainment area for ozone. The high ODVs at coastal sites emphasize the important, widely-discussed (e.g., Wolff and Lioy, 1980; Wilcox, 1996) role of transport in bringing high ozone concentrations from the major East Coast urban areas far downwind, particularly when that transport occurs over the waters of Long Island Sound and the Coastal Atlantic Ocean.





- In the past, ODVs at rural, generally upwind sites on the western border of New York (green symbols in on the left in Figure 5) were significantly smaller than in the northeastern U.S. urban areas, although in recent years that difference has diminished (Figure 6). These upwind rural areas in New York, and similar sites in Vermont (Figure S7), experienced ozone concentrations exceeding 80 ppb throughout the measurement record until about 2005. These high concentrations
caused Chautauqua County, N.Y., with a population of ~95,000, to be designated as a marginal non-attainment area, again emphasizing the importance of ozone transport in the northeastern U.S., although in this case the source of the transported ozone is not as clearly established.

Additional systematic features of the ODV time series in the northeastern U.S. are discussed in the Appendix.

The fits of Eq. 1 to the long-term trends of the ODVs from the seventeen selected groups of sites shown in Figures 6, 7 and
S1-S8 elucidate the temporal and spatial variability of ODVs in the northeastern U.S. All of the curves derived from these fits are compared in Figure 8, with the corresponding parameters included in Table 2. Considerable similarity is apparent between the fits to the ODVs from the different selections of sites in the eight states. Except for the four fits denoted by the colored dotted and dashed curves, all are similar in the sense that they exhibit the same relative long-term decrease and are asymptotically approaching approximately the same value of $y_0$. The same relative long-term decrease is necessarily forced
by the use of the same value of $\tau$ = 21.9 years in all fits. However, the derived $A$ and $y_0$ values do provide information regarding the spatial and temporal variation of ODVs over the past two to three decades in this region. Three of the four curves with noticeably different behavior are from fits to the groups of sites with the highest recently reported ODVs (Connecticut, especially the coastal sites, and the New York sites highlighted in Figures 5 and 6); these are discussed further in the Appendix. The fourth exception is the one high elevation site (Mt. Washington in New Hampshire at an elevation of
1.9 km), which is also discussed separately in the Appendix. The parameters in Table 2 provide the basis for quantitatively comparing the fits throughout the northeastern U.S. in the next two sections.

### 3.2.1 Estimation of U.S. background ODV in northeastern states

The arithmetic mean of all $y_0$ values in Table 2 (excluding the four exceptions indicated in Figure 8) is 45.9 ppb with a standard deviation of 3.2 ppb. The average of these $y_0$ values weighted with the inverse square of the respective confidence
limits is 45.8 ± 1.7 ppb, where the 95% confidence limit of this average is indicated. All of the $y_0$ values in Table 2 agree (excluding the four exceptions noted above) with these average values within their indicated confidence limits. Recalling earlier discussion, we identify $y_0$, as an estimate of U.S. background ODV, i.e., the ODV expected if U.S. anthropogenic emissions were reduced to zero. We take the latter value of $y_0$ = 45.8 ± 1.7 ppb as the best estimate of this U.S. background ODV throughout the northeastern U.S.; there is no discernable spatial variability in the U.S. background ODV within this
region. This value is significantly smaller than the value of 62.0 ± 1.9 ppb derived for southern California (Parrish et al., 2017); however even in northeastern U.S., the U.S. background ODV amounts to 65% of the 70 ppb NAAQS.



### 3.2.2 Estimation of U.S. anthropogenic ODV enhancements in northeastern states

The fits to Eq. 1 with $\tau = 21.9$ years provide estimates of $A$, the enhancements of the ODVs above $y_0$ in the reference year 2000; Table 2 lists these values for the 17 selected groups of sites from two-parameter fits, i.e., fits with $y_0$ and $A$ as independent parameters determined from the least-squares fits themselves. The discussion above estimated a constant value

of $y_0 = 45.8 \pm 1.7$ ppb for the entire northeastern U.S. region. Based on this result, we can derive one-parameter fits to Eq. 1 with $y_0$ held constant at this value of 45.8 ppb; the results of these fits are included in Table 1 as the $A^*$ values. (Such a fit is not included for the Mt. Washington results, since $y_0$ is evidently greater than 45.8 ppb as discussed in the Appendix). The $A^*$ values generally agree with the $A$ values from the two-parameter fits within their confidence limits, which are smaller, since only one parameter need be derived. The exceptions to the agreement between $A$ and $A^*$ are the fits to the exceptions

discussed earlier - the two groups of Connecticut sites and the New York maximum ozone sites, which are the upper three colored curves in Figure 8. In Table 2 the $A$ values for these three groups of sites are anomalously small compared to the results from neighboring groups of sites (i.e., New Jersey, Rhode Island, Massachusetts/coastal); the $A^*$ values for all of these neighboring groups of sites agree more closely. In the following discussion we take these $A^*$ values as the best estimate for the U.S. anthropogenic ODV enhancements in the northeastern states.

An overview of the spatial variation of the ODV enhancements across the northeastern U.S. can be obtained from a contour plot (Figure 9) derived from the $A^*$ values in Table 2. The groups of selected sites fit to Eq. 1 give only coarse spatial resolution across the northeastern U.S.; thus the contour plot has uncertainties not apparent from the smooth spatial variability of this figure. This uncertainty has been mitigated in the contour plot calculation by inclusion of duplicate values added at the site locations in each selected group of sites; these additions ensure that the contouring program reproduces a

more nearly constant value over the sometimes large regions covered by the selected groups of sites from which the individual A* values were derived. Despite the uncertainties, the contour plot does give a useful, semi-quantitative representation of the magnitude and regional variation of the ODV enhancements in the northeastern U.S. Note that the contour plot and the $A$ and $A^*$ values of Table 1 describe the ODV enhancements as they existed in the year 2000. As is apparent from Eq. 1 and the illustrated temporal trends in the figures, the ODVs have decreased throughout the last two to

three decades. We have used an e-folding time of $\tau = 21.9$ years to represent this decrease. This implies that between the reference year of 2000 and 2017, the ODV enhancements decreased by a factor of 2.2. Hence, dividing the year 2000 ODVs in the contour plot by that factor gives an approximation of the 2017 ODV enhancements (the latest year of data included in this analysis).

The ability of Eq. 1 to accurately reproduce observed ODVs can be judged by comparing those observed ODVs with the

values predicted from the fits derived with $y_0 = 45.8$ ppb and $\tau = 21.9$ years. Figure 10a shows that comparison as a correlation plot. The fits for ODVs recorded at all sites in the eight northeastern states over the entire measurement period



are calculated from the $A^*$ values at each site interpolated from the contour plot of Figure 9. The correlation is high ($r^2$ = 0.71) for the 1719 separate ODV values recorded at the 148 sites over the 2000-2017 period, but significantly lower for earlier years as expected from the figures illustrating the derived fits. A general decrease in ODVs throughout the region did not begin until 2000, which is about the time that the U.S. EPA "NOx SIP Call" began reducing power plant NOx emissions

across much of the eastern U.S. (Aleksic et al., 2013). There is significant scatter about the 1:1 line in the comparison in Figure 10a; the RMSD between observed and calculated ODVs is 5.6 ppb for the 2000-2017 period.  Much of this scatter is due to variability in ODVs recorded at different sites within a given region, which arises from differences in local photochemical ozone production and transport patterns. This variability can be reduced by comparing state maximum ODVs, rather than individual site ODVs. Figure 11 plots the time series of the maximum ODVs recorded in each year for the eight

northeastern states with respective fits over the 2010-2017 period. The derived $A^*$ values (given in Table 3) are somewhat larger than would be expected from the contour plot in Figure 9, consistent with consideration of only the maximum ODVs recorded in each state. Stronger correlation ($r^2$ = 0.89) is found for the fits to the state maximum ODVs as expected, since considering only the largest of the states' ODVs in a given year removes much of the regional variability across the state.

**4 Discussion and Conclusions**

The analysis presented in this paper has two complementary parts. First, time series of the highest ozone concentrations (i.e., the ODVs, the statistic upon which the NAAQS is based) observed in selected regions in the northern U.S. are fit to the simple mathematical relationship given by Eq. 1. This equation has two terms - one constant and one that exponentially decreases with an e-folding time of 21.9 years. Equation 1 provides fits to the most recent two to four decades of all time series of ODVs considered here through two variable parameters: $y_0$, the magnitude of the constant term, and $A$, the

magnitude of the exponentially decreasing term in the year 2000. The success of this mathematical process is judged through standard statistical tests that (1) quantify how well Eq. 1 captures the variability of the ODV time series to which the equation is fit, and (2) quantify the uncertainty of the derived parameter values. The second part of the analysis is the physical interpretation of the parameters derived from the fits to Eq. 1; $y_0$ is taken as an estimate of the U.S. background ODV (i.e., the ODV that would exist if U.S. anthropogenic emissions of ozone precursors were reduced to zero), and the

second term of Eq. 1 is interpreted as an estimate of the regional enhancement of the observed ODVs above that U.S. background ODV due to photochemical production of ozone from U.S. anthropogenic precursor emissions. These estimates apply to the region of the ODV determinations, and vary between selected regions. Thus, the derived values of $y_0$ and $A$ with their calculated uncertainties quantify the estimates of the two contributions to observed ODVs. Parrish et al. (2017) applied this same analysis approach to large areas of southern California, including the Los Angeles urban area.

In this paper the above-described analysis is applied to the northern tier of U.S. states, with a focus on two contrasting regions – three sparsely populated rural western states (Montana, North Dakota, and South Dakota, with map in Figure 3)





and eight northeastern states (New York, New Jersey, and the six New England states, with map in Figure 5). This selection is made in order to use the marked regional contrasts to judge the value of the physical interpration of Eq. 1. The northeastern states contain the New York City urban area – the most populous U.S. urban area – while the western rural states contain no large cities. The time series of ODVs recorded in these two regions differ markedly. The rural western

states span a distance of ~1700 km and cover a land area of >0.75 million km$^2$, but the ODVs recorded at 35 different sites throughout this area over a 39-year period show remarkably little variability (Figures 2 and 4) with an overall standard deviation of 3.7 ppb. In contrast, the ODVs recorded in the northeastern states vary from >160 to <50 ppb (Figures 2, 6, 7 and S1-S8). The fits of Eq. 1 to time series of ODVs in selected groups of sites in these two regions allow the estimation of both the U.S. background ODV and the enhancements of the ODVs due to U.S. anthropogenic precursor emissions, and

elucidate the spatial variability of both.

There are significant spatial variations in the derived U.S. background ODVs. Although ODVs have varied little, either spatially or temporally, within the rural western states through the entire measurement record, small but statistically significant regional variability in the U.S. background ODV and the small anthropogenic enhancements are quantified (Table 1); accounting for their associated variability reduces the standard deviation of the residuals to 2.7 ppb from the overall

standard deviation of 3.7 ppb for all ODVs recorded over 39 years in the western rural states. Within the northern U.S., the largest $y_0$ value is found for South Dakota (62.7 ± 5.1 ppb), which agrees within statistical confidence limits with that derived for southern California (62.0 ± 1.9 ppb). Within the western rural states, Montana has the smallest U.S. background ODV (53.7 ± 1.9 ppb) with North Dakota (57.0 ± 1.5 ppb) between those two values. The U.S. background ODV in the northeastern U.S. states (45.8 ± 1.7 ppb) is significantly smaller than in any of the western U.S. regions. For context, these

U.S. background ODVs account for 65 to 90% of the 2015 NAAQS of 70 ppb. No statistically significant differences are discernable over the northeastern U.S. states, except at the one relatively high elevation site (Mt. Washington in New Hampshire at 1.9 km ), which has a larger $y_0 = 66 ± 7$ ppb that is believed to be characteristic of the lower free troposphere (see Appendix for further discussion).

In the western rural states, the anthropogenic ODV enhancements, as quantified by the derived $A$ parameters, are quite small

with a wieghted average of 2.4 ± 1.2 ppb calculated for the three states plus Yellowstone NP (see Table 1 for derived parameters). In the northeastern U.S. the $A$ parameters have a systematic spatial dependence as shown by the contour plot in Figure 9, with the largest values (>54 ppb) immediately downwind of New York City decreasing to <22 ppb over northeastern Maine. Importantly, these derived $A$ parameters quantify the anthropogenic ODV enhancements in the year 2000. By 2017 these enhancements had decreased by a factor of 2.2; thus the largest ODV enhancements immediately

downwind of New York City have decreased to ~25 ppb.



### 4.1 Implications of the results

This analysis presented here and the results of Parrish et al. (2107) demonstrate that throughout diverse regions of the country (i.e., northern rural states, the northeastern U.S., and California) the U.S. background ODV contribution is significantly larger than the present-day ODV enhancements due to photochemical production from U.S. anthropogenic

precursor emissions. This comparison is true not only in rural areas, but also in the two most populous U.S. urban areas, New York City and Los Angeles. Since these ODVs are the largest observed ozone concentrations upon which the NAAQS is based, degraded air quality due to elevated ozone concentrations can be attributed primarily to the hemisphere-wide source that is responsible for the U.S. background ODV, with local and regional photochemical production enhancing that background to a significant, but smaller amount.

Forward projections of the fits to the maximum ODVs (shown in Figures 11 and 12a) allow an estimate of future trends of ODVs in the northeastern U.S. under the assumptions that 1) the U.S. background ODV (i.e., $y_0$) remains constant at 45.8 ppb throughout the region, and 2) the exponential decrease in ODV enhancements can be maintained with an e-folding time, $\tau$, of 21.9 years, by means of continued emission reduction efforts. These projections suggest that the maximum ODVs throughout the northeastern U.S. will drop below the 2015 NAAQS of 70 ppb by about 2021. However, these projections do

not account for the variability of observed maximum ODVs (i.e., RMSD of 3.9 ppb in the northeastern U.S.) about the fitted curves, so that even after 2020 this variability will likely result in the continued occasional recording of ODVs above 70 ppb.

These forward projections cannot account for any systematic deviations of the ODVs from the behavior given by Eq. 1. The recent temporal evolution of ODVs in Connecticut appears to differ significantly from the general regional behavior (see Figures 6-8 and 11). In the discussion of the fit to Eq. 1 of the Connecticut ODVs, as well as the maximum New York City

ODVs this difference was noted (see dashed colored curves in Figure 8), but nevertheless the temporal evolution was forced with $y_0$ = 45.8 ppb in deriving the $A^*$ values given in Table 2 and in deriving the contour plot of Figure 9. The different behavior and fits for Connecticut are due to the most recent five years of ODV values lying above the expected trend, as most clearly shown by Figure 11. The cause of this difference is not understood. Whether this difference is simply a statistical fluctuation cannot be determined at this time; however, random fluctuations of similar magnitude are only rarely

apparent in the temporal records of ODVs in the states discussed. McDonald et al. (2018) have recently discussed a class of ozone precursor emissions, i.e., volatile chemical products - including pesticides, coatings, printing inks, adhesives, cleaning agents, and personal care products - that have not been addressed by emission controls to the same extent as other emission sectors. The impact of this emission sector on ODVs has not been quantified, but is expected to be most significant in areas of largest population density, exactly the regions where the significant differences in temporal evolution of ODVs are noted.



The higher U.S. background ODV ($y_0$) in southern California of 62.0 ± 1.9 ppb (Parrish et al., 2017) compared to the value of 45.8 ± 1.7 ppb derived here for the northeastern U.S. implies much less difficulty in achieving the 2015 ozone NAAQS of 70 ppb in the New York City (NYC) urban area compared to Los Angeles (LA). This situation arises because the northeastern U.S. has a much larger margin for anthropogenic enhancement of ODVs while still attaining the NAAQS.

Figure 12 compares these U.S. background ODVs and the maximum ODVs in these two urban areas. In 2015 these curves indicated maximum ODVs of 78 and 102 ppb in NYC and LA, respectively. To lower the maximum ODVs to 70 ppb would require respective decreases in total ODVs of 10% in NYC and 31% in LA. However, only the ODV enhancements due to U.S. anthropogenic emissions can be addressed by local and regional controls of ozone precursor emissions. In 2015 these enhancements were about 25% larger in LA than in NYC (32 and 40 ppb, respectively). To reach a maximum ODV of 70

ppb requires ODV enhancement reductions of 25% in NYC and 80% (i.e. a factor of 5 reduction) in LA. The exponential term of Eq. 1 projects that such reductions of the 2015 ODV enhancements will require 5 years in NYC and 35 years in LA; hence the projected years of 2021 and 2050 in NYC (discussed above) and LA (Parrish et al., 2017), respectively. From the perspective of lowering maximum ODVs to the ozone NAAQS, the most important difference between NYC and LA urban areas is the higher U.S. background ODV in LA, although the 25% larger anthropogenic ODV enhancements in LA play a

secondary role. This comparison provides an insightful context for the consideration of relative anthropogenic enhancements of ozone concentrations across the country.

It should be noted that from the perspective of human health, continuing efforts to reduce ambient ozone concentrations are beneficial, despite the difficulty of achieving the NAAQS. Recent studies establish human health impacts from long-term ozone exposure over several years (Turner et al., 2016; Di et al., 2017; Berger et al., 2017). Therefore, any reduction in

ozone concentrations below present levels, regardless of whether or not ODVs remain above 70 ppb, will benefit U.S. human health.

### 4.2 Possible shortcomings of the analysis

In this work we have used Eq. 1 to quantitatively describe the temporal evolution of ODVs throughout the northern U.S.; this equation accounts for the U.S. background ODV and enhancements of the ODVs due to U.S. anthropogenic emissions. As

noted in the Introduction, a multitude of additional processes (i.e., systematic departures of the temporal evolution of the ODVs from a purely exponential decrease, possible variation in U.S. background ODVs from a single $y_0$ value associated with stratospheric intrusions, rising anthropogenic emissions in Asia, variable or increasing occurrence of wild fires, meteorological and climate variations driving differences in ozone photochemical production between years, and transport differences associated with interannual meteorological and climate variability) have been identified as possibly affecting

ozone concentrations on a variety of time scales at U.S. surface sites. However, there has been little in the way of systematic, quantitative analysis of the effects of these additional processes on ODVs across the U.S. Our approach here is to quantify



the fraction of the variation of the ODVs captured by Eq. 1 in the rural western states and in the northeastern states, and then to attribute the remaining, unexplained variability to these additional influences. Section 2 discusses the use of the $r^2$ and RMSD statistics derived from the fits of Eq. 1 to the observed ODVs to quantify the variability contributions.

The analysis of the ODVs in the western rural states (Section 3.1) demonstrates that in this vast region, the U.S. background
ODV contribution overwhelmingly dominates the observed ODVs; the 283 ODVs reported from 35 sites over 39 years of measurements in the three states have a standard deviation of only 3.7 ppb. At the individual sites and within each state the entire measurement records are all well described by averages with even smaller standard deviations (Table 1). For example, Glacier NP is a single site with a 27-year measurement record that is often utilized for characterizing background ozone concentrations (see Lin et al., 2017 and references therein); the ODVs at this site have a standard deviation of only 1.4 ppb
over the entire record. Fits to Eq. 1 included in Figure 4 show only small indications of anthropogenic enhancement of ODVs above $y_0$ (see Table 1). These fits reduce the remaining variability in the ODVs to a limited extent – the RMSD of all of ODVs about the fitted curves in Figure 4 is 2.7 ppb. Thus, contributions of all factors to ODV variability, except for changing anthropogenic precursor emissions, including the multitude of effects discussed above plus any spatial variability between sites within the individual states, are small in these western rural states - no larger than the overall 2.7 ppb RMSD
about the fitted curves.

The northeastern U.S. states present a strong contrast to the western rural states. In this region variation in the anthropogenic ODV enhancements dominates the total variability, and Eq. 1 captures the large majority of this variability. In Figure 10 the $r^2$ values for 18 years (2000-2017) indicate that Eq. 1 captures more than two-thirds of the variability of the 1719 ODVs reported from all 147 sites in the northeastern U.S., and 89% of the variability of the maximum ODVs in the eight states. The
difference between these percentages is attributed to spatial variability in the ODV enhancements not accurately represented by the contour plot of Figure 9, plus interannual variability in the spatial distribution of ODVs within the states. The RMSD between observed and calculated values is 5.6 and 3.5 ppb for all ODVs and the state maximum ODVs, respectively; these values provide estimates of the total influence of all other factors affecting ODVs over the region. Together, the root mean square contribution from these factors to the variability in the state maximum ODVs is no more than 3.5 ppb, which accounts
for no more than about 11% of their total variability over the 2000-2016 period. There are indications that the influence of these additional factors is even smaller than the 3.5 ppb RMSD of the state maximum ODVs, as several of the RMSD values for groups of sites in Table 2 are smaller than that value. In summary, Eq. 1 is remarkably successful at capturing a large fraction of the ODV variability in the northeastern U.S. states from 2000-2017; the combined influences from all other factors make only relatively minor contributions to the ODV variability across this entire region. Guo et al. (2018) discuss a
contrasting result; they suggest that monthly regional mean U.S. background MDA8 ozone concentrations vary by up to 15 ppb from year to year, and that a 3-year averaging period (as is used to define the ODV) is not long enough to eliminate interannual variability in background ozone on the days of highest observed ozone. This is not a direct comparison, but it is



clear that Guo et al. (2018) overestimate the actual variability of the observed ODVs in the two northern U.S. regions examined in this work and in southern California examined by Parrish et al. (2017).

An uncertainty in this analysis is the definition of the time constant, $\tau$, for the exponential decay of the ODV enhancement term in Eq. 1. The clear decrease in ODVs across the entire northeastern U.S. did not begin until about 2000; the 18 year

period that decrease has continued is not long enough for fits of Eq. 1 to accurately derive all three parameters. The approach we have taken is to use $\tau = 21.9$ years, the value determined for southern California (Parrish et al., 2017), in these two northern U.S. regions. It is not clear how the time scales of ODV enhancement reductions compare between California and the northeastern U.S. In California, precursor emission reductions may have been faster, because that state may have had more aggressive emission control measures, but they may also have been slower because controls on eastern coal-fired

power plants dramatically reduced NOx emissions. This latter reduction would not have occurred in California where such power plants are located downwind, out-of-state. On the other hand, emission reduction rates could be roughly the same, as most northeastern U.S. states have adopted the California on-road light-duty motor vehicle emission control program, and this is a large source sector both in California and the Northeast. The $y_0$ values derived from the fits are sensitive to the selected $\tau$ value, with a larger value of $\tau$ giving a smaller value for $y_0$, and vice versa. For example, using the Boston data

from Figure S5, increasing the assumed value of $\tau$ by 10% (from 21.9 to 24.1 years) decreases the derived $y_0$ estimate by 6% (from 45.8 to 43.3 ppb). Setting $\tau = 21.9$ years means that the confidence limits derived in this analysis are necessarily lower limits.

Finally, Eq. 1 implicitly assumes that all sectors of anthropogenic U.S. ozone precursor emissions have been reduced by emission controls at approximately the same rate. However, in some respects this is a poor approximation in that some

emission sectors have received lesser efforts than others. Any emissions that have not been reduced would tend to lead to an overestimate in the U.S. background ODV, since ozone produced from those emissions would not have decreased. Parrish et al. (2017) note that continuing agricultural emissions in the Salton Sea Air Basin may account for the anomalously high $y_0$ value derived for that region, and the possible influence volatile chemical products (McDonald et al., 2018) in the northeastern U.S. is mentioned above. It is not possible to account for uncertainties in the results that may arise from this

issue.

### 4.3 Needs for further research efforts

Accurately quantifying the U.S. background contribution to ODVs (i.e., the limit to which ODVs can be reduced through U.S. anthropogenic emission reductions alone) is important from the perspective of determining the extent of emission reductions required to attain the ozone NAAQS. In this work we have determined the value of the parameter $y_0$ of Eq. 1

within relatively small uncertainties (95% confidence limits of ~2ppb). These uncertainties are derived from the scatter in the





observed ODVs about the fits to Eq. 1. However, identifying the value of $y_0$ with U.S. background ODVs brings in additional possible uncertainties (see discussion in the preceding section) that have not been quantified. Traditionally, models have been used to estimate U.S. background ozone (see Jaffe et al., 2018 and references therein), but the models utilized in these efforts have significant shortcomings (e.g., see discussion in Parrish et al., 2017), that lead to large

uncertainties in the results. Jaffe et al. (2018) estimate an uncertainty in modeled seasonal mean U.S. background ozone of about ±10 ppb, with greater uncertainty for individual days (such as those that define the ODV), and Guo et al. (2018) find biases as high as 19 ppb in modeled seasonal mean MDA8 ozone. Thus, two approaches are available for estimating U.S. background ODVs, but each has significant, poorly quantified uncertainties.

The estimates for the U.S. background ODV derived in this work can be compared with model results. Fiore et al. (2014)

compare calculations of the fourth highest MDA8 North American background (NAB) ozone (also called policy-relevant background (PRB) ozone) from two global models. The NAB concentration is that which would be present if anthropogenic emissions were reduced to zero throughout North America, not just in the U.S. NAB ozone concentrations are therefore somewhat smaller than U.S. background ozone concentrations, but for the purposes of this comparison, we can ignore this difference. The color scales in their Figures 2 and 10 allow estimates of the U.S. background ODV from the GEOS-Chem

and AM3 models, respectively. Similarly, the color scale in Figure 6 of Emery et al. (2012) allows estimates of results from a different version of the GEOS-Chem model for the fourth highest MDA8 PRB. Figure S9 and Table S2 compare the model results with the observationally based estimates of U.S. background ODV derived in this work. These model results do have some skill in calculating the U.S. background ODVs. For five regions (three western rural states, the northeastern U.S. region, and the South Coast Air Basin) the model-observation correlations give $r^2$ values varying from 0.31 to 0.85, but the

model results are on average systematically lower by 4 to 12 ppb. Importantly, the model results disagree with each other, as well as with the observationally based results.

In summary, effective air quality management can be usefully informed by quantification of U.S. background ODVs. However, given the relatively small differences between estimated U.S. background ODVs and the 2015 ozone NAAQS of 70 ppb, these quantifications will be of more utility if they are accurate to within a couple of ppb (see Figure 12 and

associated discussion). Currently, two general approaches are available for estimating U.S. background ODVs (the observational based method discussed here and in Parrish et al. (2017) and a variety of modelling approaches), but the limited comparisons of results from these two approaches indicate differences much larger than ideal. However, the magnitudes of these disagreements are within the uncertainty of the model estimates as discussed by Jaffe et al. (2018) and Guo et al. (2018). Further improvement is required in modeling systems until their output can accurately reproduce the

magnitude and variability of the time series of observed ODVs discussed here; these model calculations can then provide accurate determination of the U.S. background ODVs, the ODV enhancements from U.S. anthropogenic emissions, and robust interpretations of the parameters $y_0$ and $A$ derived in this work from the fits to Eq. 1 to observed ODVs. Until that



model improvement is accomplished, the observationally based approach utilized in this work can provide useful estimates for air quality management guidance, as well as for comparison with evolving model calculations.

**Appendix A. Additional features of ODV time series in the northeastern states**

The text of the paper briefly described some consistent general features of the ODV time series and the corresponding fits to
Eq. 1 for selected groups of sites in the northeastern U.S. that guided the analysis. Here some additional features of interest are briefly discussed:

- New York has two non-attainment areas. In addition to the New York-N. New Jersey-Long Island, NY-NJ-CT moderate non-attainment area with a population of more than 20 million, there is the Chautauqua County (Jamestown), NY marginal non-attainment area with a population of less than 100,000. In Figure S1 the two sites in this latter non-
attainment area are highlighted in purple; the ODVs from these two sites do not differ markedly from the other upwind sites on the western border of the state. In this analysis ODVs from all of the upwind sites are considered together.

- Sites in the New York urban area and regions downwind with over-water transport paths from that urban area have recorded the largest observed ODVs. Consistent with this identification, Vermont, the only state with neither major urban areas nor an over-ocean transport path from the New York City area, records the smallest maximum ODVs (see
Table 3 and Figure S7).

- Although some sites the New York urban area record high ODVs, some other sites in central urban areas in the northeast U.S. are record the lowest ODVs (e.g., New Haven, Connecticut; Providence, Rhode Island, particularly before 2000; and Boston Massachusetts in; see Figures S3, S4 and S7, respectively). This behavior is consistent with fresh NOx emissions in urban areas reducing the ozone concentrations in air masses transported into those areas. This is evidently a
very localized phenomenon, as the suburban sites adjacent to Boston (Figure S5) exhibit ODVs similar to other coastal sites in the state.

- The farthest downwind coastal monitoring site in northeast Maine (Figure S8) records significantly lower ODVs than other coastal sites, suggesting that ozone concentrations may decrease during transport due to dilution and/or ozone loss to surface deposition.

- Interestingly, Connecticut had much higher maximum ODVs than any other state before 1985 (all points above 140 ppb in Figure 2); their cause is unknown. Since 1985 Connecticut ODVs have been similar to those of neighboring states.

- Through the measurement record, the differences between maximum and minimum ODVs have decreased, both within individual states and throughout the entire region.

- There is one monitoring site at a relatively elevated location in the northeastern U.S. - Mt. Washington in New
Hampshire at 1.9 km above sea level (asl). Although the ODV record at this site (Figure S6) is generally not higher than others recorded in New Hampshire, the fit to Eq. 1 shows a much smaller decrease than seen at any other site in the entire region. These ODVs followed a temporal evolution different from any of the other sites in the region (see curves



in Figure 8 and parameters in Table 2). The $A$ value ($8 \pm 8$ ppb) is much smaller than that of any other selected set of sites, and the U.S. background ODV ($y_0 = 66 \pm 7$ ppb) is significantly higher than the common $y_0$ value of $45.8 \pm 1.7$ ppb derived for the entire northeastern U.S. This difference is attributed to the vertical gradient of ozone over the northeastern U.S. Ozone concentrations in the free troposphere increase with altitude (e.g., see Figure 2 of Fehsenfeld et

al., 2006), and it is these higher altitude air parcels that impact Mt. Washington. The value of $y_0$ derived at Mt. Washington is in reasonable accord with the average ozone concentrations measured over the eastern U.S. by the MOZAIC program in the years near 2000 (Fehsenfeld et al., 2006). The enhancement of the ODVs (i.e., the in $A$ value) in the free troposphere observations at Mt. Washington is much smaller than the enhancements seen at the other sites, which are all located within the planetary boundary layer. Note that the temporal evolution described by the parameters

in Table 2 and illustrated in Figures 9 and S6 implies that the Mt. Washington summit site will soon record the highest ODVs in New Hampshire and higher than other sites in the northeastern U.S. outside of and immediately downwind from the New York City urban area; in 2017 Mt. Washington did report the largest ODV in New Hampshire.

- The Cadillac Mountain site at in Maine is at a somewhat elevated location (0.47 km asl). In contrast to Mt. Washington, the Cadillac Mountain ODVs (Figure S8) are generally similar to, although slightly higher than, others recorded at the

southwest Maine coastal sites. Evidently Cadillac Mountain receives primarily bounday layer air masses.  One coastal site is at a relatively high elevation (0.47 km) on Cadillac Mountain.

*Competing interests.* The author declares that he has no conflict of interest. He is the sole proprietor of David.D.Parrish, LLC, which has had contracts funded by several state and federal agencies and private companies. One of those contracts funded some of this work.

*Acknowledgements.* Some of the content of this paper was originally developed as a report submitted in fulfillment of the Technical Services Agreement between the Northeast States for Coordinated Air Use Management (NESCAUM) and David.D.Parrish, LLC funded under Agreement No. 101132 from the New York State Energy Research and Development Authority (NYSERDA).  NYSERDA has not reviewed the information contained herein, and the opinions expressed in this report do not necessarily reflect those of NYSERDA or the State of New York. The author appreciates the comments and

discussion provided by Paul Miller of NESCAUM and Tom Ryerson, Fred Fehsenfeld, Owen Cooper and Andrew Langford of NOAA ESRL CSD.  The scientific results and conclusions, as well as any views or opinions expressed herein, are those of the author, and do not necessarily reflect the views of NESCAUM, NOAA, or the Department of Commerce.





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





**Table 1. Results of least-squares fits to Eq. 1 illustrated in Figure 1; RMSD indicates the root-mean-square deviation between the observed ODVs and the derived fit.**

| State/Site | Avg. ± Std. Dev. (ppb) | $y_o$ (ppb) | $A$ (ppb) | RMSD (ppb) | years fit |
|---|---|---|---|---|---|
| Montana | 55.4 ± 2.2 | 53.7 ± 1.9 | 2.1 ± 2.1 | 2.0 | 1979-2017 |
| Glacier NP | 54.5 ± 1.3 | 54.8 ± 2.9 | -0.3 ± 3.1 | 1.4 | 1991-2017 |
| Yellowstone NP | 64.0 ± 2.1 | 59.1 ± 6.1 | 6.8 ± 8.3 | 1.7 | 1999-2017 |
| North Dakota | 59.3 ± 2.7 | 57.0 ± 1.5 | 2.5 ± 1.5 | 2.5 | 1982-2017 |
| South Dakota | 61.5 ± 3.8 | 62.7 ± 5.1 | -1.9 ± 7.8 | 3.8 | 1990-2017 |

**Table 2. Results of least-squares fits to Eq. 1 illustrated in Figures 6-8 and S1-S8; RMSD indicates the root-mean-square deviation between the observed ODVs and the derived fit.**

| State/sites | $y_0$ (ppb) | $A$ (ppb) | RMSD (ppb) | $A*$ (ppb) | years fit |
|---|---|---|---|---|---|
| New York/maximum | 53 ± 6 | 43 ± 9 | 3.9 | 53 ± 2 | 2000-2017 |
| New York/rural upwind | 42 ± 7 | 50 ± 10 | 5.1 | 44 ± 2 | 2000-2017 |
| New Jersey/all sites | 43 ± 4 | 57 ± 6 | 4.6 | 54 ± 2 | 2000-2017 |
| Connecticut/all sites | 57 ± 5 | 40 ± 7 | 5.0 | 55 ± 2 | 2000-2017 |
| Connecticut/ coastal | 61 ± 6 | 36 ± 8 | 4.1 | 57 ± 3 | 2000-2017 |
| Rhode Island/all sites | 49 ± 8 | 44 ± 12 | 4.0 | 49 ± 3 | 2000-2017 |
| Massachusetts/Boston | 46 ± 6 | 27 ± 6 | 3.1 | 27 ± 2 | 1990-2017 |
| Massachusetts/suburban | 41 ± 10 | 52 ± 14 | 3.3 | 45 ± 3 | 2000-2017 |
| Massachusetts/coastal | 44 ± 9 | 52 ± 13 | 3.2 | 49 ± 3 | 2000-2017 |
| New Hampshire/coastal | 49 ± 6 | 35 ± 8 | 3.7 | 38 ± 2 | 1995-2017 |
| New Hampshire/northwest | 45 ± 6 | 29 ± 9 | 3.7 | 28 ± 2 | 2000-2017 |
| New Hampshire/Mt. Washington | 66 ± 7 | 8 ± 8 | 2.9 | --- | 1993-2017 |
| Vermont /all sites | 45 ± 7 | 34 ± 10 | 2.7 | 33 ± 2 | 2000-2017 |
| Maine/interior | 44 ± 8 | 23 ± 10 | 5.8 | 21 ± 3 | 1990-2017 |
| Maine/NE coast | 47 ± 5 | 22 ± 5 | 2.0 | 23 ± 2 | 1991-2017 |
| Maine/SW coast | 49 ± 5 | 36 ± 5 | 4.1 | 39 ± 2 | 1990-2017 |
| Maine/Cadillac Mtn. | 52 ± 16 | 36 ± 20 | 5.2 | 44 ± 5 | 1997-2017 |

**Table 3. Results of least-squares fits of Eq. 1 to the state maximum ODVs illustrated Figure 11; $y_0$ and $\tau$ were held constant at 45.8 ppb and 21.9 years, respectively. The absolute root-mean-square deviations between the observed ODVs and the derived fits are indicated. Year$_{NNAQS}$ indicates the projected year that the fit to the state maximum ODV dropped to, or will drop to the NAAQS of 70 ppb.**

| State | $A*$ (ppb) | RMSD (ppb) | Year$_{NNAQS}$ |
|---|---|---|---|
| Connecticut | 61 ± 7 | 5.8 | 2021 |
| Maine | 48 ± 4 | 3.2 | 2015 |
| Massachusetts | 53 ± 5 | 3.9 | 2017 |
| New Hampshire | 43 ± 4 | 3.0 | 2013 |
| New Jersey | 64 ± 5 | 3.7 | 2021 |
| New York | 58 ± 4 | 3.0 | 2019 |
| Rhode Island | 52 ± 4 | 3.4 | 2017 |
| Vermont | 35 ± 3 | 2.1 | 2008 |





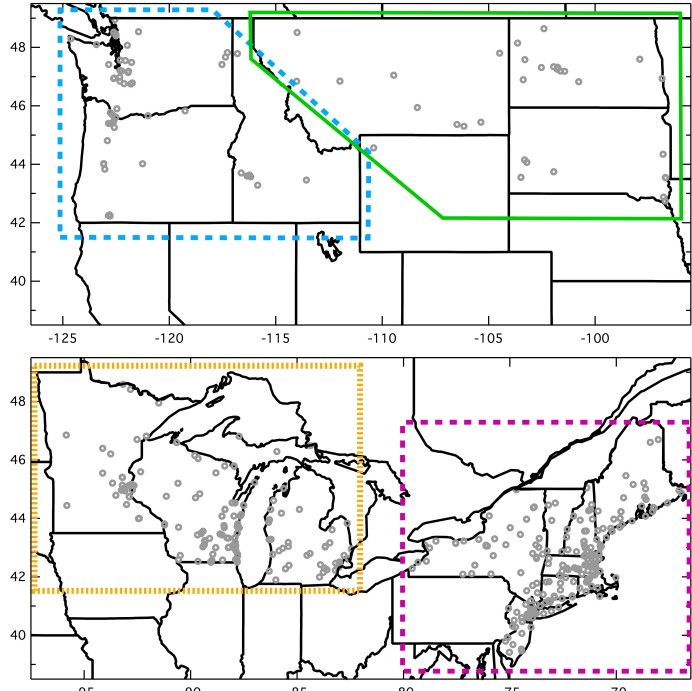

**Figure 1: Maps of the northern U.S. with all ozone monitoring sites indicated by grey circles. The colored lines indicate the four**
5    **regions considered: three Pacific Northwest states (blue dashed), three rural western states (green solid), three midwestern states**
**(dotted orange), and eight northeastern states (purple dashed).**





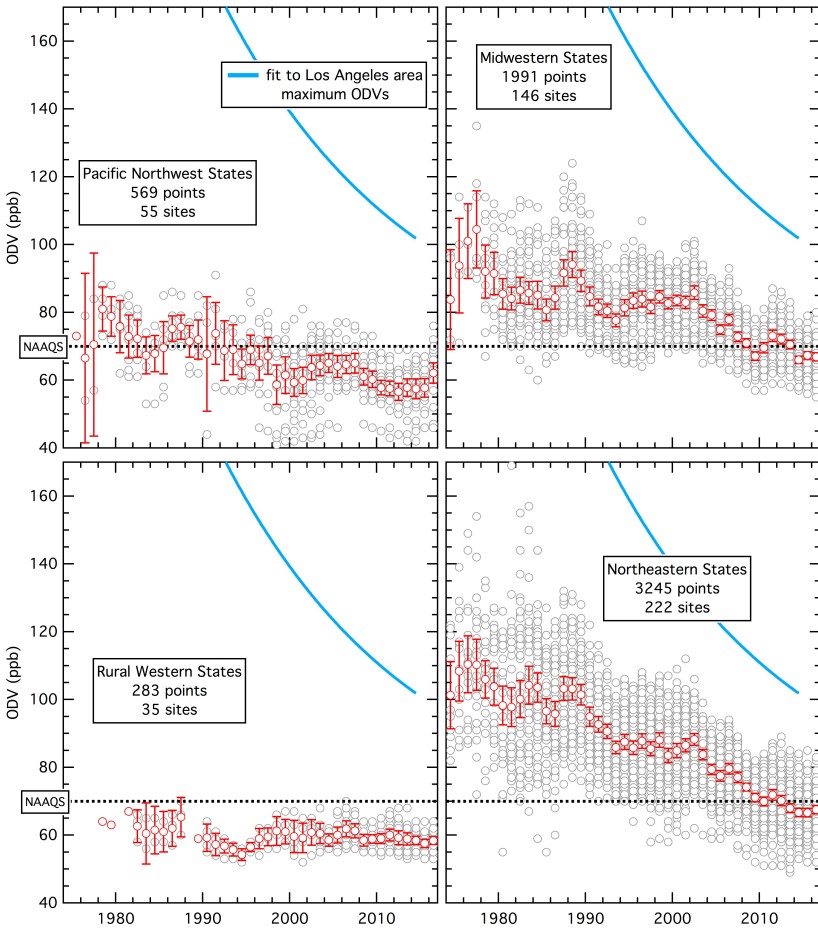

**Figure 2: Time series of all ODVs (grey symbols) reported from all monitoring sites in the four northern U.S. regions shown in Figure 1. The numbers of monitoring sites and reported ODVs are annotated for each region. The red symbols give the averages and 2-σ confidence limits for all ODVs reported in each year. For comparison, the blue curve in each panel indicates a fit to the time history of the maximum ODVs recorded in the Los Angeles urban area (Parrish et al., 2017). The dotted line indicates the 2015 NAAQS of 70 ppb.**



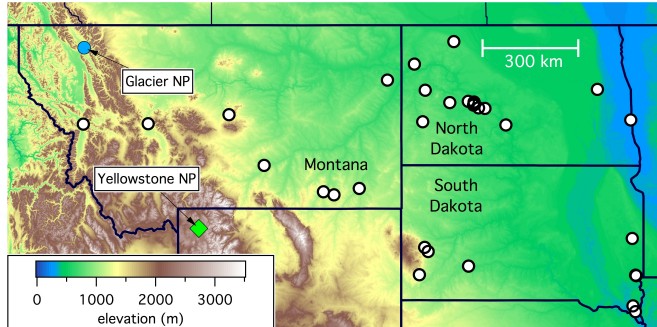

**Figure 3: Topographical map of the three rural western states with symbols indicating the locations of the monitoring sites. The two colored symbols indicate two long-term sites in national parks that are discussed in detail. (Note that Yellowstone National Park is located in Wyoming, but is nevertheless considered here.**





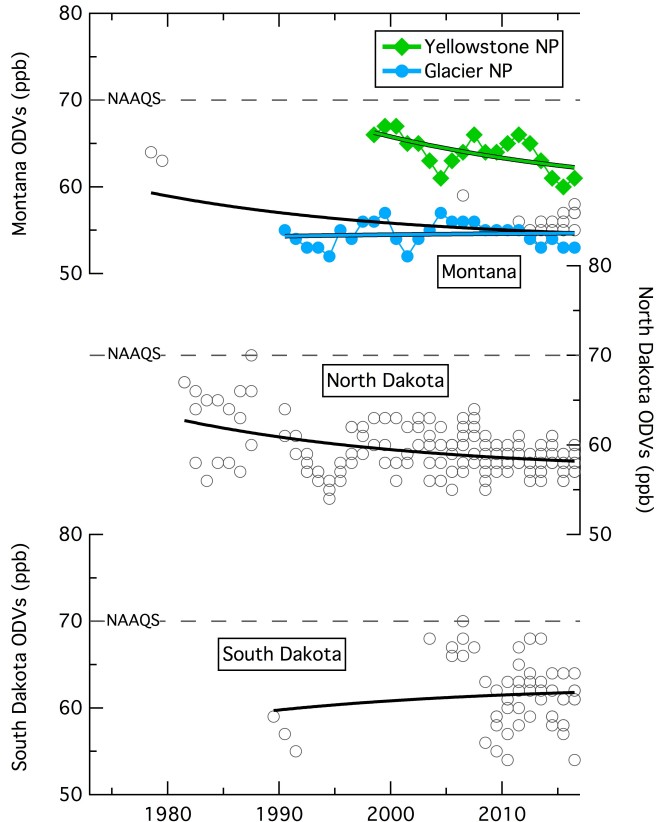

**Figure 4: Time series of all ODVs (grey symbols) reported from all monitoring sites in three rural western states, plus Yellowstone NP located in Wyoming. The two sets of colored symbols are results from two long-term sites in national parks. The curves are fits**
5 **of Eq. 1 to all data from the three states (black curves), and to the national park data in the respective colors.**



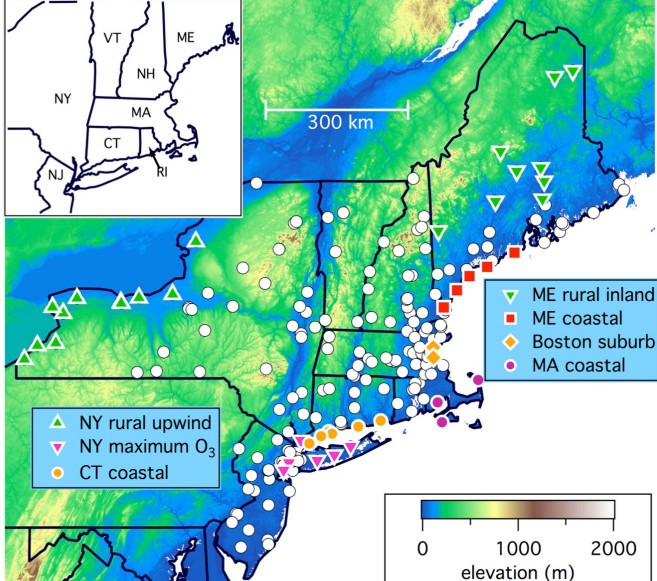

**Figure 5: Topographical map of the eight northeastern states with symbols indicating the locations of the ozone monitoring sites. Three groups of colored symbols indicate groups of sites that are discussed in detail. The inset gives the abbreviations for each of the eight states.**





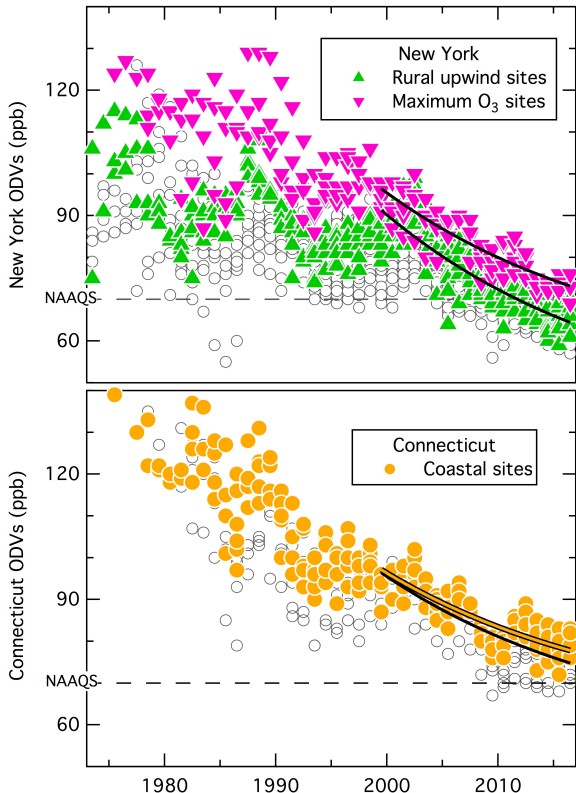

**Figure 6: Time series of all ODVs (grey symbols) reported from all monitoring sites in New York and Connecticut. The three sets of colored symbols indicate the results from groups of sites that are discussed in detail. The curves are fits of Eq. 1 to respective colored symbols, and to all data points for Connecticut.**



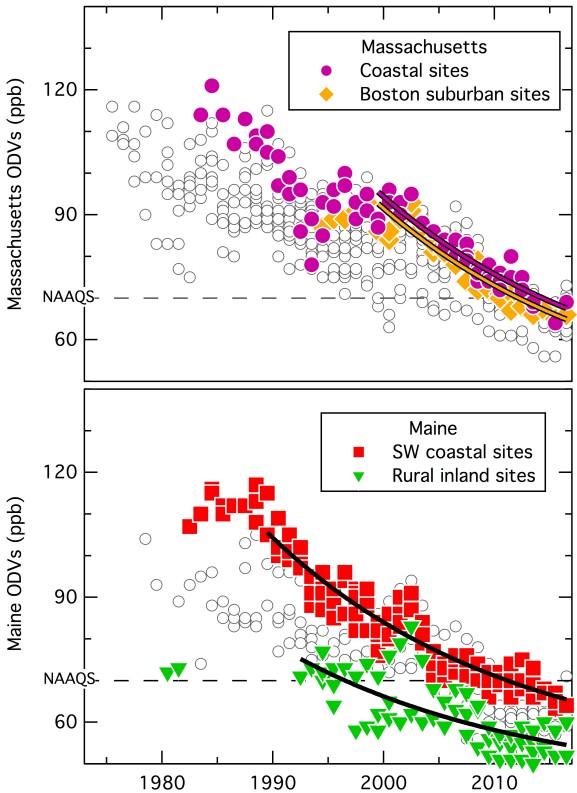

**Figure 7: Time series of all ODVs (grey symbols) reported from all monitoring sites in Massachusetts and Maine. The four sets of colored symbols indicate the results from groups of sites that are discussed in detail. The curves are fits of Eq. 1 to respective colored symbols.**



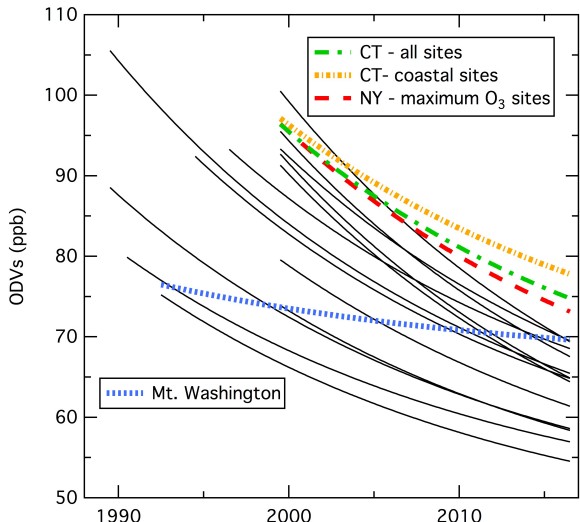

**Figure 8: Comparison of fits of the ODVs shown in Figures 6, 7 and S1-S8 to Eq. 1. The parameters of these fits are included in Table 2.**

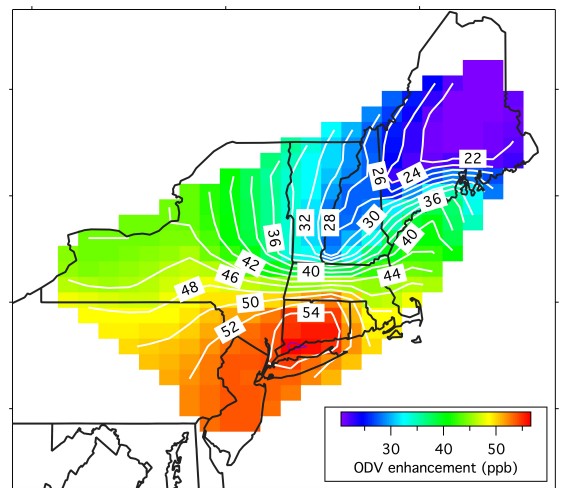

**Figure 9: Approximate contour plot of the enhancement of ozone design values due to photochemical production from U.S. anthropogenic emissions in the year 2000, estimated from the *A\** values given in Table 2.**





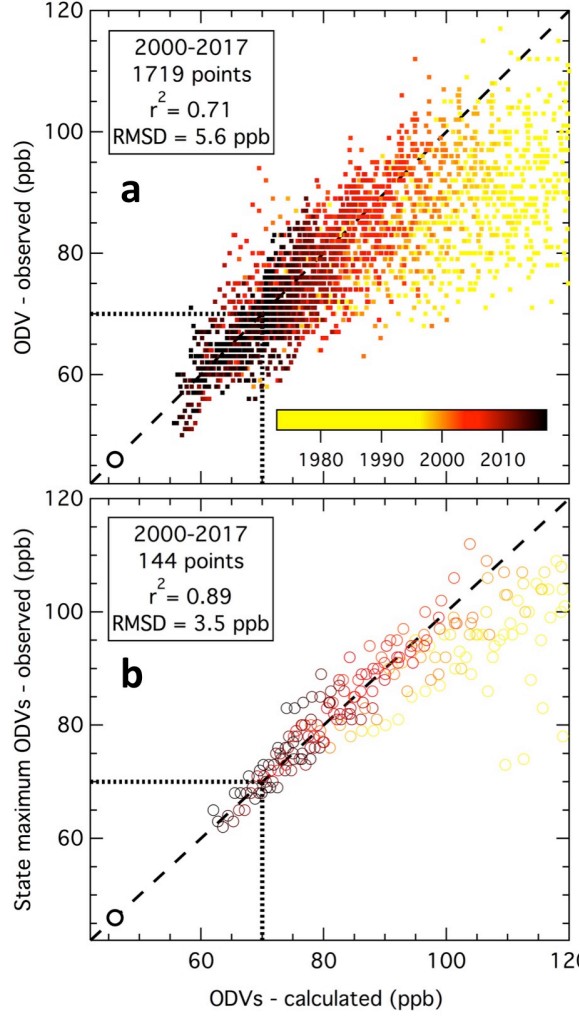

**Figure 10. Comparison of observed ODVs color-coded by year with those calculated from Eq. 1 for a) all monitoring sites and b) for the maximum observed in each state. The dashed lines indicate the 1:1 relationships with $y_0$ indicated by the larger circle, and the dotted lines the NAAQS. The number of data, square of the correlation coefficient, and the root-mean-square difference between the observed and calculated ODVs for 2000-2017 are annotated.**



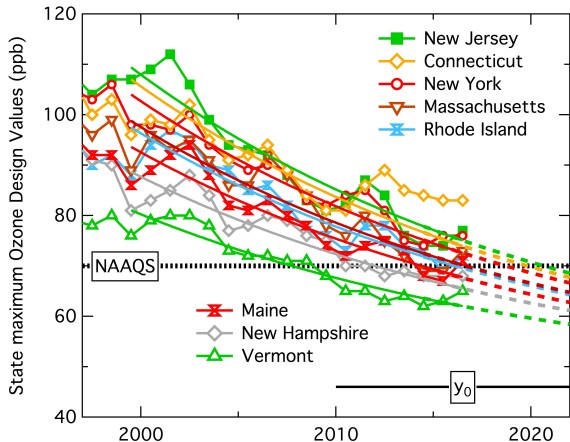

**Figure 11: Time series of maximum ODVs reported from any site within each of the eight northeastern states. The solid curves colors are fits of Eq. 1 to respective colored symbols for the 2010-2017 period. The derived $A*$ values from these latter fits are given in Table 3. The dashed lines are projections of the solid curves.**

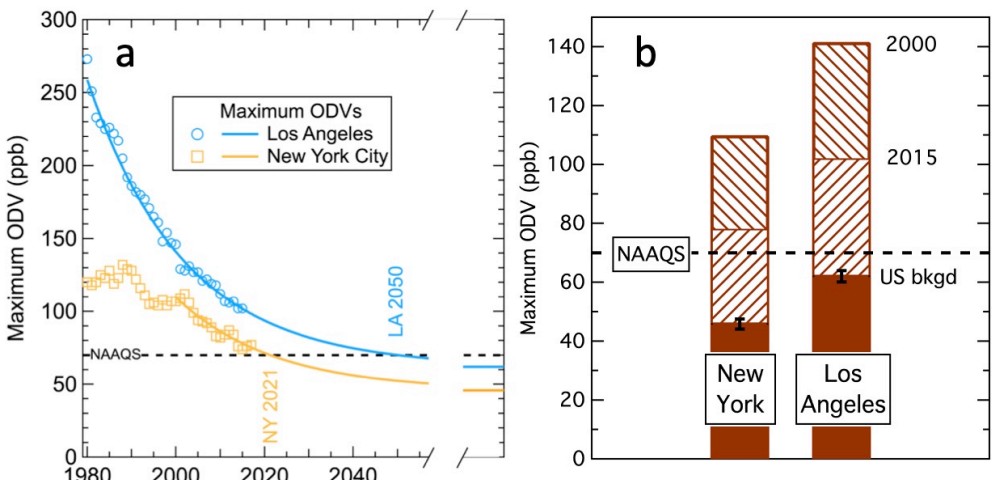

10    **Figure 12. Comparison of maximum observed ODVs in the New York City and Los Angeles urban areas. a) Temporal trend of observations (symbols) and fit to Eq. 1, including extrapolation to infinite time; annotations indicate year that extrapolations decrease to 70 ppb. The New York City results are the maxima from either the states of New York or New Jersey, and the Los Angeles results are those for the South Coast Air Basin (Figure 8 of Parrish et al., 2017). b) Bar graph indicating maximum ODVs in 2000 and 2015 (hatched bars) and the estimated U.S. background ODV (solid bars); the maximum ODVs are derived from the**
15    **fits to Eq. 1 included in a).**