# Peer review of "Estimating background contributions and U.S. anthropogenic enhancements to maximum ozone concentrations in the northern U.S."

_Atmospheric Chemistry and Physics, 2018_

## Referee Comment (RC1) · Anonymous Referee #1 · 9 Dec 2018

This paper fits long-term trends of ozone design values (ODVs) in the northeastern and rural western US to exponential decay forms with a pre-derived decay scale (21 years) from prior work for Los Angeles, and infers US background ODVs from the asymptote. It concludes that the ODV in the northeastern US is 45.8 ppb, points out that it represents a large fraction (65%) of the current NAAQS over which the US has no regulatory authority, and that it is much larger than models implying that models have large errors.

I have a number of problems with this paper, and not sure that they can be fixed, so ACP may need to do arbitration or seek another reviewer. As I see it, there is no reason

that the 21-year ODV decay time scale from LA would apply to other regions. The rural western US is mostly flat, and the northeastern US has a very different trend history initiated with the early 2000s NOx SIP Call. In fact, it seems from Figure 6 that the trend in the Northeast since 2000 could be fit to a linear decrease just as well as to an exponential decrease, and the linear decrease would imply a zero backgound ODV – which does not make sense of course but makes the point that there is no robustness to the estimate of background ODV presented here from the aymptote to the exponential decay curve. As the paper points out, a 10% change in the decay time scale would lead to a 5 ppb change in background ODV – but there is much more than 10% leeway to the fit in Figure 6.

There is also no physical rationale for a single time scale in the exponential decay of ODV, and in the absence of such a rationale any interpretation or extrapolation can be very foolish. The decrease of ODV in the Northeast is thought to be driven mainly by US NOx emissions, which have decreased linearly since 2000 according to EPA although Jiang et al. (PNAS 2018) suggest that they have been flat since 2009 – in any case, I don't see how either scenario would drive an exponential decay of ozone. Even if the response of ozone to NOx emissions was exponential, we would need a sum of exponentials to describe the ozone decrease because different anthropogenic NOx sources have decreased at different rates, and NOx emissions from fertilizer use and small industries have not decreased at all according to EPA. The effect of anthropogenic emissions on ozone is thus much more complicated than can be explained with a single exponential, and even if one can achieve such a fit to the data there is no rationale for extrapolation without understanding. Considering that NOx emissions from fertilizer use and small industries have not decreased, and that some VOC emissions have not decreased (reference in the paper to McDonald), one must conclude that the background ODV derived in this paper is biased high, possibly by a large amount.

Indeed, a punch line of the paper is that the background ODVs inferred from the exponential fit are 65-90% of the NAAQS. That seems like a big fraction, but the background

ODV estimates are biased high (see above). In addition this is misleading, considering that the ODV is depleted under polluted conditions. In the northeastern US in particular, the ozone background is highest in subsiding northerly flow, whereas the ODV exceedances are under stagnant conditions with southerly flow where background ozone is much lower.

As the paper points out, model estimates of background ozone are much lower than what is presented in this paper. The paper attributes this to error in the models. It is fair to say that there is a $\pm$ 10 ppb uncertainty in model estimates, as quoted in the paper. But that uncertainty is not a bias, whereas the background estimate in this paper is unarguably biased high. The paper does point out to some extent the uncertainties in its estimate of background ODVs, but that disappears in the abstract where the message is about the high contribution of the background to the NAAQS and how the models need to be corrected.

Aside from these basic issues of scientific content, I found the paper to be much longer than it needs to be. Figures 1 and 2 show the Pacific Northwest and the Midwest but these then drop from the radar screen, why even bother? Descriptive discussions of population, topography, etc. don't seem necessary. There's a lot of chattiness and repetition.
* * *

---

## Author Comment (AC1) · 11 Dec 2018

The author is grateful for the referee's thoughtful comments regarding this paper. However, I believe that these comments are incorrect in some respects, as detailed below, and more importantly miss the significant value of the analysis presented. U.S. policy makers must set national ambient air quality standards (NAAQS) for criteria air pollutants including ozone. A major uncertainty they face is the contribution to measured ambient concentrations made by transported background ozone. It is possible that a standard could be set at a concentration below that background contribution, at least in some regions of the country, and thus be impossible to meet through control of U.S.

[Figure]

precursor emissions. If it is also possible, indeed likely, that the background contribution varies over the country, so any standard would be more difficult to meet in some regions of the country than others. Currently policy makers must rely on estimates of the background ozone contribution calculated by models of atmospheric transport and chemistry. However, these estimates vary widely among different models, and are recognized to be so uncertain that their utility to policy makers is limited. The value of the reviewed paper is that it presents an observationally based estimate of the background contribution that policy makers can compare and contrast with model estimates. As noted by the referee, this observationally based approach is not perfect, but I believe that the results are more accurate than the model results. Each of the referee's comments is reproduced below (*in italics*) followed by my response (in plain text). For reference, I have numbered the paragraphs from the referee's comment; if a paragraph included multiple comments, those are addressed separately with numbers and letters (e.g., paragraph 3 is divided into 3a and 3b).

*1) This paper fits long-term trends of ozone design values (ODVs) in the northeastern and rural western US to exponential decay forms with a pre-derived decay scale (21 years) from prior work for Los Angeles, and infers US background ODVs from the asymptote. It concludes that the ODV in the northeastern US is 45.8 ppb, points out that it represents a large fraction (65%) of the current NAAQS over which the US has no regulatory authority, and that it is much larger than models implying that models have large errors.*

This summary is mostly accurate, if perhaps overly concise. However, the last phrase is not accurate; the response to comment 5 below discusses the issues that imply large model errors.

*2) I have a number of problems with this paper, and not sure that they can be fixed, so ACP may need to do arbitration or seek another reviewer. As I see it, there is no reason that the 21-year ODV decay time scale from LA would apply to other regions. The rural western US is mostly flat, and the northeastern US has a very different trend*

*history initiated with the early 2000s NOx SIP Call. In fact, it seems from Figure 6 that the trend in the Northeast since 2000 could be fit to a linear decrease just as well as to an exponential decrease, and the linear decrease would imply a zero background ODV – which does not make sense of course but makes the point that there is no robustness to the estimate of background ODV presented here from the aymptote to the exponential decay curve. As the paper points out, a 10% change in the decay time scale would lead to a 5 ppb change in background ODV – but there is much more than 10% leeway to the fit in Figure 6.*

The referee correctly identifies one of the more difficult aspects of the analysis presented in this paper. In the northeastern U.S., the long-term decreases of ozone design values have not continued in a consistent manner long enough for all three parameters of Equation 1 to be precisely extracted from the available data sets. Thus, applying the exponential decay time from the entire southern California region (not just LA) is a convenient approximation. However, there are reasons to believe that this is appropriate, as discussed in Section 4.2 of the paper. The significance of the referee's comment on topography is not clear to me, since the time constant reflects the decrease in the anthropogenic precursor emissions, which (as far as is known) is independent of topography. The northeastern U.S. does have a mixed trend history before 2000, but ODVs over much of this region have decreased similarly to those in California. Figure 12a illustrates the great value of comparing the two most populous U.S. urban areas in this manner.

The referee suggests that a linear fit to the data would be appropriate. Indeed, the estimation of background ODVs can be based on linear fits, as discussed by Parrish et al. (2017) with reference to their Figure 4. For the northeastern U.S. a similar analysis is possible as illustrated in Figure 1 below. Importantly, a linear decrease by itself does not imply a zero background ODV, or any other specific value; rather it implies that the ODVs can decrease indefinitely, even to increasingly negative ODVs – which indeed does not make sense. However, two linear fits to different time series of

ODVs in a region with a uniform background ODVs, do imply a particular background ODV value, which is given by the intersection of the extrapolations of the two linear fits. This follows because the background ODV does not depend upon the magnitude of the anthropogenic enhancement of ODVs, so both time series should meet at that intersection when U.S. emissions of anthropogenic ozone precursors are reduced to zero. For two extreme data sets from the northeastern U.S. (maximum ODVs observed in the New York City urban area, and all of the ODVs observed in the much more rural state of Vermont) the background ODV from the intersection of the linear fits (48.5 ppb) is in reasonable accord with the asymptotes approached by the two exponential curves for New York City and Vermont (45.8 and 45.6 ppb, respectively). The important point of this illustration is not that the linear fits are realistic (they are not), but rather that the estimation of the background ODV is not strongly dependent on the validity of the exponential fits, because they can be estimated nearly as well from linear fits. Thus, the estimates of background ODVs are quite robust.

*3a) There is also no physical rationale for a single time scale in the exponential decay of ODV, and in the absence of such a rationale any interpretation or extrapolation can be very foolish. The decrease of ODV in the Northeast is thought to be driven mainly by US NOx emissions, which have decreased linearly since 2000 according to EPA although Jiang et al. (PNAS 2018) suggest that they have been flat since 2009 – in any case, I don't see how either scenario would drive an exponential decay of ozone. Even if the response of ozone to NOx emissions was exponential, we would need a sum of exponentials to describe the ozone decrease because different anthropogenic NOx sources have decreased at different rates, and NOx emissions from fertilizer use and small industries have not decreased at all according to EPA. The effect of anthropogenic emissions on ozone is thus much more complicated than can be explained with a single exponential, and even if one can achieve such a fit to the data there is no rationale for extrapolation without understanding.*

There is a strong rationale for a single time scale in the exponential decay of ODVs,

but that rationale is more mathematical than physical. In the entire southern California region considered by Parrish et al. (2017), a single time scale in the exponential decay of ODVs captured 98.4% of the total variability in the maximum ODVs in 7 air basins, as indicated by the r2=0.984 for the correlation between the values predicted from the single time scale in the exponential decay of Equation 1 (see Figure 5 of Parrish et al., 2017). Such strong correlations are only rarely encountered in geophysical research. In 4 of the 7 air basins, this exponential decay covered the entire 36-year 1980-2015 time series of ODVs. In the present manuscript, a single time scale in the exponential decay of Equation 1 captures 89% of the total variability in the maximum ODVs in the 8 northeastern U.S. states, but only over the 18-year 2000-2017 period. The shorter time period accounts for the smaller r2 found in the present work. This strong agreement between the fits to Equation 1 and the observations in these two diverse U.S. regions is the primary rationale for using a single time scale in the exponential decay, and this strong agreement clearly demonstrates that the effect of anthropogenic emissions on ozone is not more complicated than can be explained with a single exponential. It would be of interest to understand the origin of this strong agreement, but until robust model calculation can reproduce the observed long-term trends in ODVs such an understanding is not available. The response to comment 2) shows that the extrapolation and interpretation of the exponential decay is not foolish.

*3b) Considering that NOx emissions from fertilizer use and small industries have not decreased, and that some VOC emissions have not decreased (reference in the paper to McDonald), one must conclude that the background ODV derived in this paper is biased high, possibly by a large amount.*

The referee is correct that the background ODVs derived in this paper can be biased high. This is because the fits to Equation 1 separate the ODV time trend into two contributions, one time independent, $y_0$, and a second that is time dependent. Any contribution to ODVs from emissions that have not decreased would contribute to $y_0$, and thus bias its estimate high compared to the true U.S. background ODV. However,

generally speaking there are no indications that this bias is significant, and we have some checks on the magnitude of the potential bias. For example, the same $y_0$ value is found throughout the northeastern U.S. in regions that are highly industrialized with no significant agriculture (i.e., the major metropolitan areas), rural areas with significant agriculture (e.g., the Hudson River Valley), and forested regions with little agriculture. If NOx emissions from fertilizer use or small industries were important, this importance is expected to be reflected in spatial variability of the derived $y_0$ values. A similar argument was made by Parrish et al. (2017), where the same $y_0$ value was found for 6 of the 7 California air basins, where the degree of industrialization and land use varied from the Los Angeles urban area, to the rich agricultural areas of the San Joaquin Valley, and to the sparsely populated Mojave desert. Some significant examples of derived $y_0$ values that are biased high have been identified and discussed. In the present paper, the $y_0$ values derived for the most densely populated areas (see discussion of Figure 8) are higher than expected from comparison with nearby areas; this is discussed as possibly caused by emissions of the volatile chemical products discussed by McDonald et al. (2018). Parrish et al. (2017) discuss the source of the high bias of the derived $y_0$ value for the Salton Sea air basin (NOx emissions from the highly fertilized agriculture of California's Imperial Valley). Beyond these limited examples, there is no indication of a general high bias in the results.

*4) Indeed, a punch line of the paper is that the background ODVs inferred from the exponential fit are 65-90% of the NAAQS. That seems like a big fraction, but the background ODV estimates are biased high (see above). In addition this is misleading, considering that the ODV is depleted under polluted conditions. In the northeastern US in particular, the ozone background is highest in subsiding northerly flow, whereas the ODV exceedances are under stagnant conditions with southerly flow where background ozone is much lower.*

It is difficult to respond to this comment, as several issues are combined. First, there is no evidence that the background ODV estimates are generally biased high by a signifi-

cant amount, although there are some specific instances of biases that are discussed, and their influence avoided (see response to comment 3b). It is not clear what the referee means by the statement "that the ODV is depleted under polluted conditions"; the ODV is defined as the 3-year average of the annual fourth-highest MDA8 ozone concentration, and these highest MDA8 ozone concentrations do occur under polluted conditions. If they are the highest concentrations, how can they be depleted? The analysis presented in the paper makes no assumption about, and does not depend upon, any particular flow regime, whether subsiding northerly flow or stagnant conditions with southerly flow. I can find no relevance in this comment.

*5) As the paper points out, model estimates of background ozone are much lower than what is presented in this paper. The paper attributes this to error in the models. It is fair to say that there is a $\pm 10$ ppb uncertainty in model estimates, as quoted in the paper. But that uncertainty is not a bias, whereas the background estimate in this paper is unarguably biased high. The paper does point out to some extent the uncertainties in its estimate of background ODVs, but that disappears in the abstract where the message is about the high contribution of the background to the NAAQS and how the models need to be corrected.*

Model estimates of background ozone are both higher and lower than the result presented in this paper. The $45.8 \pm 1.7$ ppb estimate of the background ODV derived in our analysis for the northeastern US is smaller than the model results illustrated in a recent assessment of background ozone over the U.S. (Jaffe et al., 2018); the color scale of their figure 3 indicates a U.S. background ODV in the northeastern U.S. in the range of 50-60 ppb. Our paper does cite other model calculations that give results that are 4 to 12 ppb lower than the 45.8 ppb estimate.

The uncertainty in model estimates of ODVs is actually significantly larger than $\pm 10$ ppb. The paper does have the sentence: "Jaffe et al. (2018) estimate an uncertainty in modeled seasonal mean U.S. background ozone of about $\pm 10$ ppb, with greater uncertainty for individual days (such as those that define the ODV), and Guo et al.

(2018) find biases as high as 19 ppb in modeled seasonal mean MDA8 ozone." Much of the model uncertainties are indeed due to biases. The referee's reference to a high bias in our analysis is discussed in the response to comment 3b) above.

It is clear from the large systematic, average differences among the results from different models that models have large errors, and that to be more useful to policy makers, the models do need to be improved.

The abstract does focus on the high contribution of the background to the NAAQS and the need for further model improvement, since those are among the primary conclusions of the paper. During revision, a brief statement will be added to the abstract regarding the uncertainty of the analysis.

*6) Aside from these basic issues of scientific content, I found the paper to be much longer than it needs to be. Figures 1 and 2 show the Pacific Northwest and the Midwest but these then drop from the radar screen, why even bother? Descriptive discussions of population, topography, etc. don't seem necessary. There's a lot of chattiness and repetition.*

The description of the Pacific Northwest and the Midwest are necessary for two reasons: first, to show that the ozone design values measured throughout the entire tier of northern U.S. states follow a common pattern (i.e., the magnitude of ODV enhancements correlate with population density, and have decreased in at least a qualitatively similar manner), and second, to justify the focus on two of the four regions, namely on the two regions presenting the greatest extremes. This paper is intended for a journal with an international readership, so I believe that descriptive discussions of population and topography are needed for readers that are not familiar with these U.S. details. During revision, the paper will be reviewed to remove chattiness and repetition.

[Figure]

[Figure]

**Fig. 1.** Comparison of linear and exponential fits to the maximum observed ODVs in the New York City urban area and all of the ODVs observed in Vermont. The dotted lines are extrapolations of the linear fits.

---

## Referee Comment (RC2) · Anonymous Referee #2 · 4 Jan 2019

The analysis is designed to separately quantify the U.S. background ozone design values (ODVs) and the enhancements of the ODVs above that background contribution due to U.S. anthropogenic precursor emissions. The U.S. background ozone design value is assumed to be the maximum ozone DV that would exist in these regions in the absence of U.S anthropogenic precursor emissions. The US background and US anthropogenic increment are derived from a simple exponential function, analogous to the function derived for California subregions in Parrish et al. (2017).

Although the idea of a simple model to describe design value behavior is appealing, there are several problems with this approach.

[Figure]

(1) The simple exponential function has been applied to separate the contributions of anthropogenic US precursor emissions to the ozone design values from the contributions by US background ozone (i.e., ozone that would be present in the absence of US anthropogenic precursor emissions). This formulation of the ozone problem is based upon a chemical transport modeling definition of US background ozone; the US background ozone can be estimated by "zeroing-out" US anthropogenic emissions in a chemical transport model. In areas far from the Pacific Coast, where US background concentrations enter the country, it is difficult to see how this simple observational model can untangle the interactions between US biogenic emissions (part of the background) and US anthropogenic emissions (part of the anthropogenic component). It is not at all clear that the asymptotic value approached by the exponential equation in this manuscript represents US background, or some mixture of US background (e.g., biogenic VOCs) combined with an especially persistent US anthropogenic (NOx) component that hasn't yet been substantially reduced by control strategies.

(2) Interannual variation of ozone data is smoothed because three years of data are averaged together to get a design value. The rationale for using the three-year average of the fourth high is that attainment of the ozone standard is linked to a three-year average, and hence, it is important to study the behavior of this somewhat unwieldy metric in order to reach policy-relevant conclusions.

But a design value is defined for a specific metropolitan area. The highest three-year average of the fourth high at any monitoring site within the metro area is the design value. Since the analysis does not examine ODVs for individual metro areas, or select the fourth high for each metro area for each year, the ODVs described in the paper are not actually the ODVs used in regulatory applications. It can be argued that this distinction is scientifically trivial, but in this case we are discussing policy, not science, so the distinction is important. The author could redefine the regions according to the EPA's definition of nonattainment areas to match the policy definition. But the statistical analysis would still be somewhat clumsy; the three-year averages smooth out much of

the interannual variability, and can cause autocorrelation issues, as the author notes. There are, however, other metrics just as relevant to policy as the ODVs used in this study. A better metric for individual monitoring sites would be the 4th high ( 98th percentile) maximum daily eight-hour ozone concentration for each year at each site. If the fourth-high/98th percentile metric for each year at each site were analyzed, there would be no overlap among years, eliminating problems with autocorrelation and excessive smoothing of interannual variations among years, yet the analysis would be at least as relevant to regulatory status as the current analysis. The physical interpretation of the data would be simplified as well, because the metric itself would be more closely tied to the observations of a single site and single year, instead of being smeared over three years.

Using a different metric would help resolve an issue related to the smoothing of interannual variation. The author asserts that the simple exponential model of ODV trends has achieved a degree of success in describing the variation, based upon the confidence intervals. These confidence intervals have been modified to account for covariance due to lack of independence among ODVs. But the interannual variation would be larger if the analysis had been performed on annual 98th percentiles rather than ODVs for each site, and it is unclear whether the modification of confidence intervals to account for covariance also accounts for the reduction of interannual variation. The results would be more compelling if the interannual variation had not been shaved down by using three-year running averages.

(3) Increasing the interannual variation, however, would probably worsen another issue: the inability of the model to converge on a solution for the three model parameters. As the author notes on page 7, lines 10-16, the shorter data record for the northern regions appears to be preventing estimation of the three parameters of the exponential function. To resolve this issue, the author has assumed that one of the parameters can be set at the same value as derived for California. As the author notes, the value of $\tau$ïĂă= 21.9 years derived from California implicitly assumes that control strategies

have produced approximately equal relative reductions in anthropogenic ozone enhancements throughout the country. This assumption is questionable, and the results for the northeastern states seems to show that it is unwarranted.

(4) Table 2 shows the derived values of y0 (US background ozone) and A (US anthropogenic component) for subsets of monitoring sites. The values of US background ozone for low altitude sites vary from 41±10 ppb in suburban Massachusetts to 61±6 ppb in coastal Connecticut. This is a large variation over a short distance for a value that is supposed to reflect relatively unvarying background ozone. One interpretation is that the Connecticut "background O3" includes a lot of ozone generated in the New York City area, and that therefore the simple exponential model cannot determine US background. The author chose a different interpretation, and re-set the US background to a lower value, which increased the US anthropogenic component to more acceptable levels. This portion of the analysis is not convincing, and seems to be an attempt to compensate for the simple model's shortcomings. In fact, it is essentially an admission that the original simple model cannot be used to distinguish between US background ozone and US anthropogenic ozone.

Ultimately, I have concluded that the assertions claiming that US background and anthropogenic increment can be derived from the simple exponential function are not compelling, especially for the northeastern states. It is possible that the analysis could be re-worked, by changing the ozone metric to the annual 98th percentile, but the failure of the method to derive the three parameters of the model even with three-year running averages, and its inability to distinguish between US background and US anthropogenic in the northeastern states suggests that this simple approach is flawed for the regions to which it has been applied in this study. I thought that the study was interesting, and the author did a commendable job in explaining the uncertainties and possible shortcomings of the approach. This admirable transparency in describing the methods is worthy of emulation, but I do not think the study should be published in its present form.

---

## Author Comment (AC2) · 7 Jan 2019

The author is grateful for the time and thought reflected in the referee's comments regarding this paper. They will lead to improved discussion in the revised paper. However, I believe that these comments are incorrect in some respects, as detailed below, and very importantly miss the significant value of the analysis presented. U.S. policy makers must set national ambient air quality standards (NAAQS) for criteria air pollutants including ozone. A major uncertainty they face is the contribution to measured ambient concentrations made by transported background ozone. Currently policy makers must rely on estimates of the background ozone contribution calculated by models

of atmospheric transport and chemistry. However, these estimates vary widely among different models, and are recognized to be so uncertain that their utility to policy makers is limited. The value of the reviewed paper is that it presents an observationally based estimate of the background contribution that policy makers can consider in their work. As noted by the referee, this observationally based approach is not perfect, but I believe that the results are more accurate than the model results. Each of the referee's comments is reproduced below (*in italics*) followed by my response (in plain text).

*The analysis is designed to separately quantify the U.S. background ozone design values (ODVs) and the enhancements of the ODVs above that background contribution due to U.S. anthropogenic precursor emissions. The U.S. background ozone design value is assumed to be the maximum ozone DV that would exist in these regions in the absence of U.S anthropogenic precursor emissions. The US background and US anthropogenic increment are derived from a simple exponential function, analogous to the function derived for California subregions in Parrish et al. (2017).*

*Although the idea of a simple model to describe design value behavior is appealing, there are several problems with this approach.*

*(1) The simple exponential function has been applied to separate the contributions of anthropogenic US precursor emissions to the ozone design values from the contributions by US background ozone (i.e., ozone that would be present in the absence of US anthropogenic precursor emissions). This formulation of the ozone problem is based upon a chemical transport modeling definition of US background ozone; the US background ozone can be estimated by "zeroing-out" US anthropogenic emissions in a chemical transport model. In areas far from the Pacific Coast, where US background concentrations enter the country, it is difficult to see how this simple observational model can untangle the interactions between US biogenic emissions (part of the background) and US anthropogenic emissions (part of the anthropogenic component). It is not at all clear that the asymptotic value approached by the exponential equation in this manuscript represents US background, or some mixture of US background (e.g.,*

*biogenic VOCs) combined with an especially persistent US anthropogenic (NOx) component that hasn't yet been substantially reduced by control strategies.*

The referee speculates that there may be "an especially persistent US anthropogenic (NOx) component that hasn't yet been substantially reduced by control strategies," which can confound the analysis presented in the manuscript. As we discuss in the paper, this is a concern. However, there is no evidence of such a component with an NOx emission magnitude comparable to the well-known emission sectors (mobile, industrial and power plant sources) that have been effectively reduced by control strategies. The influence of such an emission component must be kept under consideration, but in that absence of any evidence of such a component, there is no justification for rejecting the results of the analysis presented in the manuscript.

Notably, the emission inventories typically used for regional photochemical modeling do not include any such "especially persistent US anthropogenic (NOx) component". If such a component did exist, photochemical modeling results would be biased. Such a possibility further emphasizes the value of the present observationally based estimate of U.S. background ozone, so that biases in either the model results or the observationally based estimate can be understood through comparison of results, and the results obtained from each approach improved.

*(2) Interannual variation of ozone data is smoothed because three years of data are averaged together to get a design value. The rationale for using the three-year average of the fourth high is that attainment of the ozone standard is linked to a three-year average, and hence, it is important to study the behavior of this somewhat unwieldy metric in order to reach policy-relevant conclusions.*

*But a design value is defined for a specific metropolitan area. The highest three-year average of the fourth high at any monitoring site within the metro area is the design value. Since the analysis does not examine ODVs for individual metro areas, or select the fourth high for each metro area for each year, the ODVs described in the paper*

*are not actually the ODVs used in regulatory applications. It can be argued that this distinction is scientifically trivial, but in this case we are discussing policy, not science, so the distinction is important. The author could redefine the regions according to the EPA's definition of nonattainment areas to match the policy definition.*

The referee's description of ozone design values is not entirely complete. As discussed in the manuscript, an ODV is defined for each monitor in the U.S. The ODVs analyzed in this paper are indeed the ODVs that are considered in regulatory applications. EPA's process for defining nonattainment areas involves many considerations beyond the tabulated ODVs. One of these considerations is the ODV for the area; the area ODV is simply the highest ODV of those recorded at all of the sites within the area during each year. Since the highest site ODVs are included in our analysis, we are analyzing the ODVs actually used in regulatory applications. To assuage the referee's concern, Figure 1 below illustrates the analysis for nonattainment area ODVs, and Table 1 gives the derived parameters. (The four nonattainment areas that lie entirely within the northeastern states are included; the fifth nonattainment area is only partially in the area under consideration, so it is not included.) These nonattainment areas are described in the manuscript. As can be seen from comparing Table 1 below with Table 2 of the manuscript (and properly considering the parameter confidence limits included in the tables), the analysis presented in the manuscript is directly relevant to the nonattainment area ODVs. The results for the NY-NJ-LI-CT and Greater CT nonattainment areas in Table 1 are in close accord with the Connecticut and New York/maximum results given in Table 2 of the manuscript. Similarly, the results for the Jamestown, NY and Dukes Co, MA nonattainment areas in Table 1 are in close accord with the New York/rural upwind and Massachusetts/coastal results given in Table 2 of the manuscript. The analysis for the nonattainment areas is entirely consistent with the analysis and discussion included in the manuscript; nothing new would be gained by including this additional analysis in the manuscript; however, it could be included in the Supporting Information of a revised submission.

*But the statistical analysis would still be somewhat clumsy; the three-year averages smooth out much of the interannual variability, and can cause autocorrelation issues, as the author notes. There are, however, other metrics just as relevant to policy as the ODVs used in this study. A better metric for individual monitoring sites would be the 4th high (98th percentile) maximum daily eight-hour ozone concentration for each year at each site. If the fourth-high/98th percentile metric for each year at each site were analyzed, there would be no overlap among years, eliminating problems with autocorrelation and excessive smoothing of interannual variations among years, yet the analysis would be at least as relevant to regulatory status as the current analysis. The physical interpretation of the data would be simplified as well, because the metric itself would be more closely tied to the observations of a single site and single year, instead of being smeared over three years.*

*Using a different metric would help resolve an issue related to the smoothing of interannual variation. The author asserts that the simple exponential model of ODV trends has achieved a degree of success in describing the variation, based upon the confidence intervals. These confidence intervals have been modified to account for covariance due to lack of independence among ODVs. But the interannual variation would be larger if the analysis had been performed on annual 98th percentiles rather than ODVs for each site, and it is unclear whether the modification of confidence intervals to account for covariance also accounts for the reduction of interannual variation. The results would be more compelling if the interannual variation had not been shaved down by using three-year running averages.*

The preceding two paragraphs critique the statistical fitting technique utilized in the manuscript; however, the referee's discussion is incorrect. The goal of the statistical analysis is to extract the systematic, long-term change in a set of observed ODVs (or fourth-high/98th percentile) as accurately as possible, given the interannual variability about those long-term changes. No statistically significant information regarding the long-term change is lost by working with 3-year means (i.e., the ODVs) rather than

annual mean data (i.e, fourth-high/98th percentile). A qualitative explanation for this can be given. Deriving 3-year means from annual data involves an averaging process, which minimizes the sum of the squares of the deviations of the annual data from the derived 3-year means. The fitting procedure employed in the analysis to derive the long-term change minimizes the sum of the squares of the deviations of the 3-year ODVs from the derived long-term change. The final result is independent of whether the sum of the squares of the deviations is minimized in two steps (3-year mean calculation followed by the fit to the long-term change), or in one step (extracting the long-term change directly from the higher frequency annual data). We work with 3-year mean ODVs because they are of most policy relevance, and we have properly dealt with autocorrelation issues.

Additionally, working with annual mean data (i.e, fourth-high/98th percentile) would worsen a separate autocorrelation concern. Interannual meteorological and climate variations can drive differences in ozone photochemical production and transport; such differences can persist over multiple years, so that annual mean data may be autocorrelated. Working with 3-year mean ODVs reduces the influence of this autocorrelation.

*(3) Increasing the interannual variation, however, would probably worsen another issue: the inability of the model to converge on a solution for the three model parameters. As the author notes on page 7, lines 10-16, the shorter data record for the northern regions appears to be preventing estimation of the three parameters of the exponential function. To resolve this issue, the author has assumed that one of the parameters can be set at the same value as derived for California. As the author notes, the value of tau = 21.9 years derived from California implicitly assumes that control strategies have produced approximately equal relative reductions in anthropogenic ozone enhancements throughout the country. This assumption is questionable, and the results for the northeastern states seems to show that it is unwarranted.*

As discussed in my response immediately preceding this comment, no statistically significant information regarding the long-term change is lost working with 3-year means.

Increasing the interannual variation by working with annual means would not worsen the uncertainty in deriving the three model parameters; the greater number of independent data available for the fitting procedure would closely compensate for the larger interannual variation.

The referee is correct that the assumption of tau = 21.9 years is questionable, and that question is addressed in detail in my response to Comment 2 of referee 1. Notably, the referee does not state in what way "the results for the northeastern states seems to show that it (the assumption) is unwarranted"; thus I am unable to respond to the last part of this comment.

*(4) Table 2 shows the derived values of $y_0$ (US background ozone) and A (US anthropogenic component) for subsets of monitoring sites. The values of US background ozone for low altitude sites vary from 41±10 ppb in suburban Massachusetts to 61±6 ppb in coastal Connecticut. This is a large variation over a short distance for a value that is supposed to reflect relatively unvarying background ozone.*

The referee correctly points out that he values in Table 2 derived for $y_0$ for the low altitude sites vary from 41±10 ppb in suburban Massachusetts to 61±6 ppb in coastal Connecticut and suggests that this variation is large. This large variation arises from two sources. First, as reflected in the 95% confidence limits given in the table, there is statistical uncertainty in deriving the parameter values from the fits of Equation 1 to the ODVs; the confidence limits for the two extreme results that the referee quotes indicate that the statistically significant portion of the variation is not as large as the referee suggests. Second, as discussed in the paper, some of the variation does arise from real departures of the derived $y_0$ values from the true U.S. background ozone. Figure 2 below illustrates an analysis of the distribution of the derived $y_0$ values that allows these two sources of variation to be examined separately. The ordinate scale of this figure is designed so that a normal distribution defines a straight line. In this figure, 13 of the 17 derived $y_0$ values do define a normal distribution with a median of 47.7 ppb and a standard deviation of 4.5 ppb. We believe that these 13 values accurately reflect

U.S. background ozone; this result is in close accord with the $45.8 \pm 1.7$ ppb estimate of this quantity given in the manuscript. The 4 of 17 derived $y_0$ values in Figure 2 lie in a high value tail of the distribution; these are the 4 $y_0$ values that we discuss separately in the manuscript. This figure nicely illustrates the value of our analysis based upon the simple model encapsulated in Equation 1 – the large majority of the results provide a robust estimate of the regional U.S. background ozone, and deviations from a uniform result point toward issues that deserve further investigation.

*One interpretation is that the Connecticut "background O3" includes a lot of ozone generated in the New York City area, and that therefore the simple exponential model cannot determine US background.*

This interpretation suggested by the referee is not valid. Connecticut ODVs certainly do reflect a lot of ozone generated in the New York City area. However, this New York City generated ozone has been decreasing over the years; thus, our analysis would properly include it in Connecticut's time varying term (A*exp[-(year-2000)/tau]), not in Connecticut's constant $y_0$ term. The simple exponential model can indeed determine U.S. background ozone, within the caveats thoroughly discussed in the manuscript.

*The author chose a different interpretation, and re-set the US background to a lower value, which increased the US anthropogenic component to more acceptable levels. This portion of the analysis is not convincing, and seems to be an attempt to compensate for the simple model's shortcomings. In fact, it is essentially an admission that the original simple model cannot be used to distinguish between US background ozone and US anthropogenic ozone.*

The referee incorrectly describes the reasoning in our manuscript. We followed two clearly distinct steps. The first step is the purely mathematical fitting of Equation 1 to the observed ODVs. The success of this step is judged by the good agreement between the fits and the observations. The second step is the physical interpretation of the A and $y_0$ parameters. As discussed in the manuscript and illustrated in Figure 2 below, the

derived $y_0$ values provides a great deal of information regarding the U.S. background ozone, but we do not argue that the two quantities are identical in all situations. Hence, the four values in the high value tail, which includes the Connecticut $y_0$ value. We do not re-set the U.S. background ozone to a lower value; rather we discuss why the derived Connecticut $y_0$ value is not equal to the U.S. background ozone. This is not an attempt to compensate for the simple model's shortcomings; rather it is a discussion of an issue identified by the simple model that deserves further investigation. It is not an admission that the original simple model cannot be used to distinguish between US background ozone and US anthropogenic ozone; in the majority of cases the model provides a robust distinction.

*Ultimately, I have concluded that the assertions claiming that US background and anthropogenic increment can be derived from the simple exponential function are not compelling, especially for the northeastern states. It is possible that the analysis could be re-worked, by changing the ozone metric to the annual 98th percentile, but the failure of the method to derive the three parameters of the model even with three-year running averages, and its inability to distinguish between US background and US anthropogenic in the northeastern states suggests that this simple approach is flawed for the regions to which it has been applied in this study. I thought that the study was interesting, and the author did a commendable job in explaining the uncertainties and possible shortcomings of the approach. This admirable transparency in describing the methods is worthy of emulation, but I do not think the study should be published in its present form.*

The author appreciates the referee's positive comments. However, as discussed in detail above, I believe that the objections raised by the referee are generally not valid. Hence, the overall conclusion stated here is not justified.
* * *
[Figure]

**Fig. 1.** Time series of nonattainment area ODVs (solid symbols) and the ODVs reported from all monitoring sites (open symbols) in the four nonattainment areas included within the northeastern states.

Fig. 2 shows:

- median = 47.7 ppb
- Std. dev = 4.5 ppb

Labels on plot: New York/maximum, Connecticut/all sites, Connecticut/coastal, Mt. Washington

Axis labels: cumulative probability distribution (y-axis), $y_0$ (ppb) (x-axis)

**Fig. 2.** Cumulative probability plot of the y0 determinations listed in Table 2 of the manuscript. The line is a linear regression fit to the open points, which defines a normal distribution.

**Table 1. Results of least-squares fits to Eq. 1 illustrated in Figure 1; RMSD indicates the root-mean-square deviation between the observed ODVs and the derived fit.**

| Nonattainment area/sites | $y_0$ (ppb) | $A$ (ppb) | RMSD (ppb) | years fit |
|---|---|---|---|---|
| NY-NJ-LI-CT area/ODVs | 63 ± 12 | 40 ± 16 | 3.0 | 2000-2017 |
| NY-NJ-LI-CT/all sites | 53 ± 4 | 42 ± 6 | 5.8 | 2000-2017 |
| Greater CT area/ODVs | 59 ± 12 | 36 ± 17 | 3.2 | 2000-2017 |
| Greater CT/all sites | 54 ± 7 | 39 ± 10 | 4.1 | 2000-2017 |
| Jamestown, NY area/ODVs | 45 ± 14 | 51 ± 19 | 3.7 | 2000-2017 |
| Jamestown, NY/all rural upwind sites | 42 ± 7 | 50 ± 10 | 5.1 | 2000-2017 |
| Dukes Co, MA/all sites | 44 ± 9 | 52 ± 13 | 3.2 | 2000-2017 |

**Fig. 3.** Table 1

---

## Author Comment (AC3) · 8 Jan 2019

The discussion paper currently under review has important scientific and policy implications. Scientifically, the paper argues that a simple, mathematically based, conceptual model can accurately estimate two contributions to U.S. ozone design values (ODVs) in the northeastern U.S.: the first from U.S. background ozone, and the second from enhancements produced by photochemical ozone production from U.S. anthropogenic ozone precursor emissions. Assuming that the first is constant and the second is decreasing exponentially allows these two contributions to be estimated separately. I believe that this is currently the only observationally based approach for estimating

background ozone contributions to U.S. ODVs. Further, the paper argues that U.S. background ozone estimates from regional photochemical models have large uncertainties, as evidenced by the large differences between the results from different models. These two results are important from an air quality policy perspective, since policy makers consider estimates of the background ozone contribution in formulating ozone air quality standards. Given its scientific importance and policy relevance, this paper should be published provided that it is scientifically sound.

The analysis in the paper relies on fitting long-term trends of ozone design values (ODVs) in the northeastern U.S. to an exponential decrease with a constant positive offset (Equation 1 of the text). The ODV time series in the northeastern U.S. are too short to allow all 3 model parameters to be accurately extracted from the fits, so the exponential time constant derived from prior work for southern California (21.9 years) is assumed to be appropriate for the northeastern U.S. as well. The U.S. background ODVs are then estimated from the constant offset ($y_0$) values derived from the nonlinear regression fits to Equation 1 with that time constant. The results indicated no statistically significant difference in the derived U.S. background ODVs over the northeastern U.S., giving a regional average U.S. background ODV of $45.8 \pm 1.7$ ppb.

Both referees question various aspects of this analysis approach, particularly the fitting of the anthropogenic ODV enhancements to an exponential decrease with a time constant equal to that found in southern California. Importantly, there is an independent analysis that can also estimate the U.S. background ODV. Section 2.3 of Parrish et al. (2017) demonstrates a different, somewhat more general approach that is based upon correlations of ODVs between different regions. This approach does not assume any specific functional form for the time dependence of the ODV enhancements. Instead, the time series of ODVs from one region is selected as a reference, and other time series are linearly correlated with that reference. A different assumption does underlie this approach, namely that the U.S. background ODVs vary in a similar manner in all regions under investigation. Given this assumption, the U.S. background ODV for a region is taken as the ODV where the line derived from a linear correlation of that region's ODVs with those of the reference region equals 1; at that point the ODVs from the two time series are equal, which is necessarily at that regionally uniform U.S. background ODV. Section 2.3 of Parrish et al. (2017) show that the results of this approach for seven southern California air basins are nearly identical to the results from the fits to Equation 1.

The results from applying this second approach to the northeastern U.S. are shown in Figures S1-S7 of the Supplement for the 13 regional data sets given by the black lines in Figure 8 of the discussion paper. Here the time series of maximum observed ODVs in the New York City urban area (included in Figure 12a of the discussion paper) is selected as the reference, because they are some of the largest ODVs recorded in the northeastern U.S., and because after 2000 this time series closely follows an exponential decrease with little interannual variability. Figure 1 below collects all of the linear regressions for the 13 regional data sets, and Table 1 summarizes the results, which have significant variability (36 to 62 ppb) and wide confidence limits (4 to 13 ppb), but with an average of $49.2 \pm 3.9$ (95% confidence limit) ppb. One reason for the high bias in this second determination is that the fits were obtained from standard linear regressions, which assign all of the uncertainty to the data set plotted on the ordinate. The analysis can also be done utilizing a reduced major axis regression with equal weighting of the two correlated data sets, which gives a corresponding average of $43.2 \pm 5.7$ ppb. These two results bracket the result ($45.8 \pm 1.7$ ppb) reported in the discussion paper, and neither average is statistically significantly different from the original result. The analysis presented in this comment demonstrates that the fitting of long-term trends of ODVs to Equation 1 of the text does not compromise the accuracy of the results presented in the discussion paper.

**Reference**

Parrish, D.D., Young, L.M., Newman, M.H., Aikin, K.C., and Ryerson, T.B.: Ozone design values in Southern California's air basins: Temporal evolution and U.S.

[Figure]

background contribution. Journal of Geophysical Research: Atmospheres, 122, 11,166–11,182. https://doi.org/10.1002/2016JD026329, 2017.

Please also note the supplement to this comment:
https://www.atmos-chem-phys-discuss.net/acp-2018-1174/acp-2018-1174-AC3-supplement.pdf

––––––––––––––––––––––––––––––––––––

**Fig. 1.** Results of standard linear regressions between the ODVs from 13 regions in the north-eastern U.S. and the maximum ODVs recorded in New York City as illustrated in Figures S1-S7 of the Supplement.

**Table 1. Results of the intercepts of the linear regressions illustrated in Figure 1 with the 1:1 line.**

| State/sites | U.S. background ODV (ppb) | $r^2$ | years fit |
|---|---|---|---|
| New York/rural upwind | 43 ± 6 | 0.75 | 2000-2017 |
| New Jersey/all sites | 52 ± 4 | 0.83 | 2000-2017 |
| Rhode Island/all sites | 62 ± 6 | 0.84 | 2000-2017 |
| Massachusetts/Boston | 47 ± 11 | 0.75 | 2000-2017 |
| Massachusetts/suburban | 50 ± 10 | 0.82 | 2000-2017 |
| Massachusetts/coastal | 53 ± 7 | 0.89 | 2000-2017 |
| New Hampshire/coastal | 55 ± 8 | 0.77 | 2000-2017 |
| New Hampshire/northwest | 49 ± 6 | 0.60 | 2000-2017 |
| Vermont /all sites | 51 ± 7 | 0.79 | 2000-2017 |
| Maine/interior | 39 ± 11 | 0.44 | 2000-2017 |
| Maine/NE coast | 36 ± 10 | 0.89 | 1991-2017 |
| Maine/SW coast | 56 ± 6 | 0.77 | 2000-2017 |
| Maine/Cadillac Mtn. | 48 ± 13 | 0.85 | 2000-2017 |

**Fig. 2.** Table 1

---

## Author Response (AR1)

**Author's Response to interactive comments by Anonymous Referees #1 and #2 on "Estimating background contributions and U.S. anthropogenic enhancements to maximum ozone concentrations in the northern U.S." by David D. Parrish**

The author is grateful for the referees' thoughtful comments regarding this paper. Responses to all comments are included here, and where appropriate the manuscript has been revised as described herein, and indicated in the "tracked changes" manuscript copy at the end of this response. Many of the referees' comments are incorrect in some respects, or are not germane to the issues that this manuscript investigates; in such cases detailed responses are included below, and some additions have been made to the manuscript to more fully discuss these issues.

I believe that this manuscript is very important and deserves publication. U.S. policy makers must set national ambient air quality standards (NAAQS) for criteria air pollutants including ozone. A major uncertainty they face is the contribution to measured ambient concentrations made by transported background ozone. It is possible that a standard could be set at an ozone concentration below that background contribution, at least in some regions of the country, and thus be impossible to meet through control of U.S. precursor emissions. If it is also possible, indeed likely, that the background contribution varies substantially over the country, so any standard would be more difficult to meet in some regions of the country than others. Currently, policy makers must rely on estimates of the background ozone contribution calculated by models of atmospheric transport and chemistry. However, these estimates vary widely among different models, and are recognized to be so uncertain that their utility to policy makers is limited. The value of this paper is that it presents an observationally based estimate of the background ozone contribution that policy makers can compare and contrast with model estimates. As noted by the referees (and acknowledged and discussed in the paper) this observationally based approach is not perfect, but all analyses suggest that the results are more accurate than the currently available model results, and that the paper deserves publication. The comments of each referee are reproduced below (*in blue italics*) followed by my response (in plain text).

**Referee #1 Comments**

For reference, I have numbered the paragraphs from the first referee's comments; when a paragraph included multiple comments, those are addressed separately with numbers and letters (e.g., paragraph 3 is divided into 3a and 3b).

*1) This paper fits long-term trends of ozone design values (ODVs) in the northeastern and rural western US to exponential decay forms with a pre-derived decay scale (21 years) from prior work for Los Angeles, and infers US background ODVs from the asymptote. It concludes that the ODV in the northeastern US is 45.8 ppb, points out that it represents a large fraction (65%) of the current NAAQS over which the US has no regulatory authority, and that it is much larger than models implying that models have large errors.*

This summary is mostly accurate, if perhaps overly concise. However, the last phrase is not accurate; the response to comment 5 below discusses the issues that imply large model errors.

*2) I have a number of problems with this paper, and not sure that they can be fixed, so ACP may need to do arbitration or seek another reviewer. As I see it, there is no reason that the 21-year ODV decay time scale from LA would apply to other regions. The rural western US is mostly flat, and the northeastern US has a very different trend history initiated with the early 2000s NOx SIP Call. In fact, it seems from Figure 6 that the trend in the Northeast since 2000 could be fit to a linear decrease just as well as to an exponential decrease, and the linear decrease would imply a zero backgound ODV – which does not make sense of course*

*but makes the point that there is no robustness to the estimate of background ODV presented here from the aymptote to the exponential decay curve. As the paper points out, a 10% change in the decay time scale would lead to a 5 ppb change in background ODV – but there is much more than 10% leeway to the fit in Figure 6.*

The referee correctly identifies one of the more difficult aspects of the analysis presented in this paper. In the northeastern U.S., the long-term decreases of ozone design values have not continued in a consistent manner long enough for all three parameters of Equation 1 to be precisely extracted from the available data sets. Thus, applying the exponential decay time from the entire southern California region (not just LA) is a convenient, and as we demonstrate below, accurate approximation. Section 4.2 of the original manuscript did discuss reasons that the long-term ozone changes may be similar to those in southern California; this discussion is retained in the revised manuscript. The northeastern U.S. does have a mixed trend history before 2000, but ODVs over much of this region have decreased similarly to those in California. Figure 12a illustrates the great value of comparing the two most populous U.S. urban areas in this manner.

The referee suggests that a linear fit to the data would be appropriate. Indeed, the estimation of background ODVs in the northeastern U.S. can be based on linear fits to the long-term changes as illustrated in Figure 1. Importantly, a linear decrease by itself does not imply a zero backgound ODV, or any other specific value; rather it implies that the ODVs can decrease

[Figure]

**Figure 1**: Comparison of linear fits (solid black lines) and exponential fits (colored lines) to the maximum observed ODVs in the New York City urban area (from Figure 12 of the original paper), and all of the ODVs observed in Vermont (from Figure S7 of the original Supporting Information). The dotted lines are extrapolations of the linear fits, with the red circle indicating their intersection at an ODV value of 48.5 ppb.

indefinitely to increasingly negative ODVs – which indeed does not make sense. However, two linear fits to different time series of ODVs in a region with a uniform background ODVs, do imply a particular background ODV value, which is given by the intersection of the extrapolations of the two linear fits. This follows because the background ODV does not depend upon the magnitude of the anthropogenic enhancement of ODVs, so both time series should meet at that intersection when U.S. emissions of anthropogenic ozone precursors are reduced to zero. For two extreme northeastern U.S. data sets (the maximum ODVs observed in the New York City urban area, and all of the ODVs observed in the much more rural state of Vermont) the background ODV from the intersection of the linear fits  (48.5 ppb) is in reasonable accord with the asymptotes approached by the two exponential curves for New York City and Vermont (45.8 and 45.6 ppb, respectively). The important point of this figure and discussion is not that the linear fits are realistic (they are not), but rather that the estimation of the background ODV is not strongly dependent on the validity of the exponential fits, because they can be estimated nearly as well from linear fits. Thus, contrary to the referee's assertion, the estimates of background ODVs presented in the manuscript are robust.

Importantly, there is an independent analysis that can also estimate the U.S. background ODVs without the assumption of any specific functional form for the time dependence of the ODV enhancements. Section 2.3 of Parrish et al. (2017) discusses this different, somewhat more general approach, which is based upon correlations between ODVs recorded in different regions with substantially different anthropogenic enhancements. The time series of ODVs from one region is selected as a reference, and other time series are linearly correlated with this reference. A different assumption does underlie this approach, namely that the

U.S. background ODVs vary in a similar manner, but not necessarily as an exponential decrease, in all regions under consideration. Given this assumption, the U.S. background ODV for a region is taken as the ODV where the line derived from a linear correlation of that region's ODVs with those of the reference region intercepts the 1:1 line; at that point the extrapolated ODVs from the two time series would be equal, which is necessarily at that regionally uniform U.S. background ODV. Section 2.3 of Parrish et al. (2017) show that the results of this approach for seven southern California air basins are nearly identical to the results from the fits to Equation 1.

The results from applying this second approach to the northeastern U.S. are shown in Figures S11-S18 of the revised Supplement for the 13 regional data sets given by the black lines in Figure 8 of the revised manuscript. Here the time series of maximum observed ODVs in the New York City urban area (included in Figure 12a of the revised manuscript) is selected as the reference, because they are some of the largest ODVs recorded in the northeastern U.S., and because after 2000 this time series closely follows an exponential decrease with little interannual variability. Figure S18 of the Supplement illustrates all of the linear regressions for the 13 regional data sets, and Table S3 summarizes the results. These results have significant variability (25 to 62 ppb), because the correlation slopes are near unity, so it is difficult to determine the intercept of the linear correlation with the 1:1 line. However, the average of the derived background ODVs (49.2 ± 3.9 ppb for standard linear regressions and 42.5 ± 5.7 ppb for reduced major axis regressions, where 95% confidence limits are indicated) bracket the result (45.8 ± 1.7 ppb) reported in the manuscript for the exponential fits, and neither average is statistically significantly different from that result. The analysis presented in this comment demonstrates that the fitting of long-term trends of ODVs to Equation 1 of the text does not compromise the accuracy of the results presented in the discussion paper.

***Changes incorporated in revised manuscript:*** Section 3.2.3 "Alternative approach for estimating U.S. background ODV in northeastern states", has been added to the manuscript; it contains much of the above discussion describing this alternate approach. The results are included in the Supplement - Figures S11-S18 and Table S3. The agreement between the two analysis approaches is now briefly discussed in Section 4.2 "Possible shortcomings of the analysis".

*3a) There is also no physical rationale for a single time scale in the exponential decay of ODV, and in the absence of such a rationale any interpretation or extrapolation can be very foolish. The decrease of ODV in the Northeast is thought to be driven mainly by US NOx emissions, which have decreased linearly since 2000 according to EPA although Jiang et al. (PNAS 2018) suggest that they have been flat since 2009 – in any case, I don't see how either scenario would drive an exponential decay of ozone. Even if the response of ozone to NOx emissions was exponential, we would need a sum of exponentials to describe the ozone decrease because different anthropogenic NOx sources have decreased at different rates, and NOx emissions from fertilizer use and small industries have not decreased at all according to EPA. The effect of anthropogenic emissions on ozone is thus much more complicated than can be explained with a single exponential, and even if one can achieve such a fit to the data there is no rationale for extrapolation without understanding.*

There is a strong rationale for a single time scale in the exponential decay of ODVs, but that rationale is more mathematical than physical. In the entire southern California region considered by Parrish et al. (2017), a single time scale in the exponential decay of ODVs captured 98.4% of the total variability in the maximum ODVs in 7 air basins, as indicated by the $r^2=0.984$ for the correlation between the observed ODVs and the values predicted from the single time scale in the exponential decay of Equation 1 (see Figure 5 of Parrish et al., 2017). Such strong correlations are only rarely encountered in geophysical research. In 4 of the 7 air basins, this exponential decay covered the entire 36-year 1980-2015 time series of ODVs. In the present manuscript, a single time scale in the exponential decay of Equation 1 captures 89% of the total variability in the maximum ODVs in the 8 northeastern U.S. states over the 18-year 2000-2017 period. The shorter time period northeastern U.S. accounts for the smaller $r^2$

found in the present work. This strong agreement between the fits to Equation 1 and the observations in these two diverse U.S. regions is the primary rationale for using a single time scale in the exponential decay, and this strong agreement clearly demonstrates that the effect of anthropogenic emissions on ozone is not more complicated than can be explained with a single exponential. It would be of interest to understand the origin of this strong agreement, but until robust model calculation can reproduce the observed long-term trends in ODVs such an understanding is not available.

***Changes incorporated in revised manuscript:*** The strong agreement between the fits to the exponential decrease and the observations was already discussed in Section 4.2 "Possible shortcomings of the analysis"; no further discussion has been added.

*3b) Considering that NOx emissions from fertilizer use and small industries have not decreased, and that some VOC emissions have not decreased (reference in the paper to McDonald), one must conclude that the background ODV derived in this paper is biased high, possibly by a large amount.*

The referee is correct that the background ODVs derived in this paper can be biased high. This is because the fits to Equation 1 separate the ODV time trend into two contributions, one time independent ($y_0$) and one time dependent ($A \exp\{-(year-2000)/t\}$). Any contribution to ODVs from emissions that have not decreased would be counted in the time independent term, and thus bias its estimate high compared to the true U.S. background ODV. However, generally speaking there are no indications that this bias is significant, and we have some checks on the magnitude of the potential bias. For example, the same $y_0$ value is found throughout the northeastern U.S. in regions that are highly industrialized with no significant agriculture (i.e., the major metropolitan areas), rural areas with significant agriculture (e.g., the Hudson River Valley), and forested regions with little agriculture. If NOx emissions from fertilizer use or small industries were important, their importance is expected to be reflected in spatial variability of the derived $y_0$ values. A similar argument was made by Parrish et al. (2017), where the same $y_0$ value was found for 6 of the 7 California air basins, where the degree of industrialization and land use varied from the Los Angeles urban area, to the rich agricultural areas of the San Joaquin Valley, and to the sparsely populated Mojave desert. Some significant examples of derived $y_0$ values that are biased high have been identified and discussed. In the present paper, the $y_0$ values derived for the most densely populated areas (see discussion of Figure 8) are higher than expected from comparison with nearby areas; this is discussed as possibly caused by emissions of the volatile chemical products discussed by McDonald et al. (2018). Parrish et al. (2017) discuss the source of the high bias of the derived $y_0$ value for the Salton Sea air basin (NOx emissions from the highly fertilized agriculture of California's Imperial Valley). Beyond these limited examples, there is no indication of a significant, general high bias in the results.

***Changes incorporated in revised manuscript:*** This possible bias was already discussed in Sections 4.1 "Implications of the results" and 4.2 "Possible shortcomings of the analysis"; no further discussion has been added.

*4) Indeed, a punch line of the paper is that the background ODVs inferred from the exponential fit are 65-90% of the NAAQS. That seems like a big fraction, but the background ODV estimates are biased high (see above). In addition this is misleading, considering that the ODV is depleted under polluted conditions. In the northeastern US in particular, the ozone background is highest in subsiding northerly flow, whereas the ODV exceedances are under stagnant conditions with southerly flow where background ozone is much lower.*

It is difficult to respond to this comment, as several issues are combined. First, there is no evidence that the background ODV estimates are generally biased high by a significant amount, although there are some specific instances of biases that are discussed, and their influence avoided (see response to comment 3b). It is not clear what the referee means by the statement "that

the ODV is depleted under polluted conditions"; the ODV is defined as the 3-year average of the annual fourth-highest MDA8 ozone concentration, and these highest MDA8 ozone concentrations do occur under polluted conditions. If they are the highest concentrations, how can they be depleted? The analysis presented in the paper makes no assumption about, and does not depend upon, any particular flow regime, whether subsiding northerly flow or stagnant conditions with southerly flow. I can find no relevance in this comment.

***Changes incorporated in revised manuscript:*** No further discussion has been added.

*5) As the paper points out, model estimates of background ozone are much lower than what is presented in this paper. The paper attributes this to error in the models. It is fair to say that there is a ±10 ppb uncertainty in model estimates, as quoted in the paper. But that uncertainty is not a bias, whereas the background estimate in this paper is unarguably biased high. The paper does point out to some extent the uncertainties in its estimate of background ODVs, but that disappears in the abstract where the message is about the high contribution of the background to the NAAQS and how the models need to be corrected.*

Model estimates of background ozone are both higher and lower than the result presented in this paper. The $45.8 \pm 1.7$ ppb estimate of the background ODV derived in our analysis for the northeastern US is smaller than the model results illustrated in a recent assessment of background ozone over the U.S. (Jaffe et al., 2018); the color scale of their figure 3 indicates a U.S. background ODV in the northeastern U.S. in the range of 50-60 ppb. Our paper does cite other model calculations that give results that average 4 to 12 ppb lower than the 45.8 ppb estimate.

The uncertainty in model estimates of ODVs is actually significantly larger than ±10 ppb. The paper does have the sentence: "Jaffe et al. (2018) estimate an uncertainty in modeled seasonal mean U.S. background ozone of about ±10 ppb, with greater uncertainty for individual days (such as those that define the ODV), and Guo et al. (2018) find biases as high as 19 ppb in modeled seasonal mean MDA8 ozone."  Much of the uncertainty in model results is indeed due to biases. The referee's reference to a high bias in our analysis is discussed in the response to comment 3b) above.

It is clear from the large systematic, average differences among the results from different models that models have large errors, and that to be more useful to policy makers, the models do need to be improved.

The abstract does focus on the high contribution of the background to the NAAQS and the need for further model improvement, since those are among the primary conclusions of the paper. The abstract also states "The approach used in this work has some unquantified uncertainties that are discussed."

***Changes incorporated in revised manuscript:*** The last sentence of the abstract has been changed to read "Further model improvement is required until their output can accurately reproduce the time series and variability of observed ODVs. Ideally, the uncertainties in the modeling and observational based approaches can then be reduced through additional comparisons."

*6) Aside from these basic issues of scientific content, I found the paper to be much longer than it needs to be. Figures 1 and 2 show the Pacific Northwest and the Midwest but these then drop from the radar screen, why even bother? Descriptive discussions of population, topography, etc. don't seem necessary. There's a lot of chattiness and repetition.*

The description of the Pacific Northwest and the Midwest are necessary for two reasons: first, to show that the ozone design values measured throughout the entire tier of northern U.S. states follow a common pattern (i.e., the magnitude of ODV enhancements correlate with population density, and have decreased in at least a qualitatively similar manner), and second, to justify the focus on two of the four regions, namely on the two regions presenting the greatest extremes. This paper is intended

for a journal with an international readership, so I believe that descriptive discussions of population and topography are needed for readers that are not familiar with these U.S. details.

***Changes incorporated in revised manuscript:*** The paper has been revised to remove chattiness and repetition where they could be found.

**Referee #2 Comments**

*The analysis is designed to separately quantify the U.S. background ozone design values (ODVs) and the enhancements of the ODVs above that background contribution due to U.S. anthropogenic precursor emissions. The U.S. background ozone design value is assumed to be the maximum ozone DV that would exist in these regions in the absence of U.S anthropogenic precursor emissions. The US background and US anthropogenic increment are derived from a simple exponential function, analogous to the function derived for California subregions in Parrish et al. (2017).*

*Although the idea of a simple model to describe design value behavior is appealing, there are several problems with this approach.*

*(1) The simple exponential function has been applied to separate the contributions of anthropogenic US precursor emissions to the ozone design values from the contributions by US background ozone (i.e., ozone that would be present in the absence of US anthropogenic precursor emissions). This formulation of the ozone problem is based upon a chemical transport modeling definition of US background ozone; the US background ozone can be estimated by "zeroing-out" US anthropogenic emissions in a chemical transport model. In areas far from the Pacific Coast, where US background concentrations enter the country, it is difficult to see how this simple observational model can untangle the interactions between US biogenic emissions (part of the background) and US anthropogenic emissions (part of the anthropogenic component). It is not at all clear that the asymptotic value approached by the exponential equation in this manuscript represents US background, or some mixture of US background (e.g., biogenic VOCs) combined with an especially persistent US anthropogenic (NOx) component that hasn't yet been substantially reduced by control strategies.*

The referee speculates that there may be "an especially persistent US anthropogenic (NOx) component that hasn't yet been substantially reduced by control strategies" that can confound the analysis presented in the manuscript. As discussed in the paper, this is a concern. However, there is no evidence for such a component with a NOx emission magnitude that is comparable to the well-known emission sectors (mobile, industrial and power plant sources) that have been effectively reduced by control strategies. The possible influence of such an emission component must be kept under consideration, but in that absence of any evidence of such a component, there is no justification for rejecting the results of the analysis presented in the manuscript.

Notably, the emission inventories typically used for regional photochemical modeling do not include any such "especially persistent US anthropogenic (NOx) component". If such a component did exist, photochemical modeling results also would be biased. Such a possibility further emphasizes the value of the present observationally based estimate of U.S. background ozone, so that biases in either the model results or the observationally based estimate can be understood through comparison of results, and the results obtained from each approach improved.

***Changes incorporated in revised manuscript:*** The manuscript has already discussed biases due to anthropogenic precursors that have not been reduced by emission controls. No further discussion of this issue has been added.

*(2) Interannual variation of ozone data is smoothed because three years of data are averaged together to get a design value. The rationale for using the three-year average of the fourth high is that attainment of the ozone standard is linked to a three-year average, and hence, it is important to study the behavior of this somewhat unwieldy metric in order to reach policy-relevant conclusions.*

*But a design value is defined for a specific metropolitan area. The highest three-year average of the fourth high at any monitoring site within the metro area is the design value. Since the analysis does not examine ODVs for individual metro areas, or select the fourth high for each metro area for each year, the ODVs described in the paper are not actually the ODVs used in regulatory applications. It can be argued that this distinction is scientifically trivial, but in this case we are discussing policy, not science, so the distinction is important. The author could redefine the regions according to the EPA's definition of nonattainment areas to match the policy definition.*

The referee's description of ozone design values is not entirely complete. As discussed in the manuscript, an ODV is defined for each monitor in the U.S. The ODVs analyzed in this paper are indeed the ODVs that are considered in regulatory applications. EPA's process for defining nonattainment areas (rural as well as metropolitan areas) involves many considerations, including the ODV for the area; the area ODV is simply the highest ODV of those recorded at any of the sites within the area during each year. Since the highest site ODVs are included in our analysis, we are analyzing the ODVs actually used in regulatory applications. To assuage the referee's concern, Figure 2 illustrates the analysis for nonattainment area ODVs, and Table 1 gives the derived parameters. (The four nonattainment areas that lie entirely within the northeastern states are included; the fifth nonattainment area is only partially in the area under consideration, so it is not included.) These nonattainment areas are described in the manuscript. As can be seen from comparing Table 1 below with Table 2 of the manuscript (and properly considering the

[Figure]

**Figure 2:** Time series of nonattainment area ODVs (solid symbols) and the ODVs reported from all monitoring sites (open symbols) in the four nonattainment areas included within the northeastern U.S. states examined in our analysis. The curves are fits of Eq. 1 to respective colored symbols, or to all ODVs (black curves) in the areas.

**Table 1.** Results of least-squares fits to Eq. 1 illustrated in Figure 1; RMSD indicates the root-mean-square deviation between the observed ODVs and the derived fit.

| Nonattainment area/sites | $y_0$ (ppb) | $A$ (ppb) | RMSD (ppb) | years fit |
|---|---|---|---|---|
| NY-NJ-LI-CT area/ODVs | 63 ± 12 | 40 ± 16 | 3.0 | 2000-2017 |
| NY-NJ-LI-CT/all sites | 53 ± 4 | 42 ± 6 | 5.8 | 2000-2017 |
| Greater CT area/ODVs | 59 ± 12 | 36 ± 17 | 3.2 | 2000-2017 |
| Greater CT/all sites | 54 ± 7 | 39 ± 10 | 4.1 | 2000-2017 |
| Jamestown, NY area/ODVs | 45 ± 14 | 51 ± 19 | 3.7 | 2000-2017 |
| Jamestown, NY/all rural upwind sites | 42 ± 7 | 50 ± 10 | 5.1 | 2000-2017 |
| Dukes Co, MA/all sites | 44 ± 9 | 52 ± 13 | 3.2 | 2000-2017 |

parameter confidence limits included in the tables), the analysis presented in the manuscript is directly relevant to the nonattainment area ODVs. The results for the NY-NJ-LI-CT and Greater CT nonattainment areas in Table 1 are in close accord with the Connecticut and New York/maximum results, respectively, given in Table 2 of the manuscript. Similarly, the results for the Jamestown, NY and Dukes Co, MA nonattainment areas in Table 1 are in close accord with the New York/rural upwind and Massachusetts/coastal results, respectively, given in Table 2 of the manuscript. The analysis for the nonattainment areas is entirely consistent with the analysis and discussion included in the manuscript; nothing new would be gained by including this additional analysis in the manuscript.

***Changes incorporated in revised manuscript:*** No further discussion of this issue has been added to the manuscript.

*But the statistical analysis would still be somewhat clumsy; the three-year averages smooth out much of the interannual variability, and can cause autocorrelation issues, as the author notes. There are, however, other metrics just as relevant to policy as the ODVs used in this study. A better metric for individual monitoring sites would be the 4th high (98th percentile) maximum daily eight-hour ozone concentration for each year at each site. If the fourth-high/98th percentile metric for each year at each site were analyzed, there would be no overlap among years, eliminating problems with autocorrelation and excessive smoothing of interannual variations among years, yet the analysis would be at least as relevant to regulatory status as the current analysis. The physical interpretation of the data would be simplified as well, because the metric itself would be more closely tied to the observations of a single site and single year, instead of being smeared over three years.*

*Using a different metric would help resolve an issue related to the smoothing of interannual variation. The author asserts that the simple exponential model of ODV trends has achieved a degree of success in describing the variation, based upon the confidence intervals. These confidence intervals have been modified to account for covariance due to lack of independence among ODVs. But the interannual variation would be larger if the analysis had been performed on annual 98th percentiles rather than ODVs for each site, and it is unclear whether the modification of confidence intervals to account for covariance also accounts for the reduction of interannual variation. The results would be more compelling if the interannual variation had not been shaved down by using three-year running averages.*

The preceding two paragraphs criticize the statistical fitting technique utilized in the manuscript; however, the referee's discussion is not entirely correct. The goal of the statistical analysis is to extract the systematic, long-term change in a set of observed ODVs (or fourth-high/98th percentile) as accurately as possible, given the interannual variability about those long-term changes. Importantly, no statistically significant information regarding the long-term change is lost by working with 3-year means (i.e., the ODVs) rather than annual mean data (i.e, fourth-high/98th percentile). A qualitative explanation for this can be

given. Deriving 3-year means from annual data involves an averaging process, which minimizes the sum of the squares of the deviations of the annual data from the derived 3-year means. The fitting procedure employed in the analysis to derive the long-term change minimizes the sum of the squares of the deviations of the 3-year ODVs from the derived long-term change. The final result is independent of whether the sum of the squares of the deviations is minimized in two steps (3-year mean calculation followed by the fit to the long-term change), or in one step (extracting the long-term change directly from the higher frequency annual data). We work with 3-year mean ODVs because they are of most policy relevance, and we have properly dealt with the autocorrelation issues. And the confidence limits derived from the procedure for fitting the long-term changes properly account for the uncertainty arising from interannual variations, whether that fit is to 3-year or annual means.

Additionally, working with annual mean data (i.e, fourth-high/98th percentile) would worsen a separate autocorrelation concern. Interannual meteorological and climate variations can drive differences in ozone photochemical production and transport, differences that can persist over multiple years, so that annual mean data may be autocorrelated. Working with 3-year mean ODVs reduces the influence of this additional autocorrelation.

***Changes incorporated in revised manuscript:*** No further discussion of these issues has been added to the manuscript.

*(3) Increasing the interannual variation, however, would probably worsen another issue: the inability of the model to converge on a solution for the three model parameters. As the author notes on page 7, lines 10-16, the shorter data record for the northern regions appears to be preventing estimation of the three parameters of the exponential function. To resolve this issue, the author has assumed that one of the parameters can be set at the same value as derived for California. As the author notes, the value of $\tau$ = 21.9 years derived from California implicitly assumes that control strategies have produced approximately equal relative reductions in anthropogenic ozone enhancements throughout the country. This assumption is questionable, and the results for the northeastern states seems to show that it is unwarranted.*

As discussed in my response immediately preceding this comment, no statistically significant information regarding the long-term change is lost working with 3-year means. Increasing the interannual variation by working with annual means would not worsen the uncertainty in deriving the three model parameters; the greater number of independent data available for the fitting procedure would closely compensate for the larger interannual variation.

The referee is correct that the assumption of $\tau$ = 21.9 years is questionable, and that question is addressed in detail in my response to Comment 2 of referee #1. Notably, the referee does not state in what way "the results for the northeastern states seems to show that it (the assumption) is unwarranted"; thus I am unable to respond to the last part of this comment.

***Changes incorporated in revised manuscript:*** No further discussion of this issue has been added to the manuscript.

*(4) Table 2 shows the derived values of y0 (US background ozone) and A (US anthropogenic component) for subsets of monitoring sites. The values of US background ozone for low altitude sites vary from 41±10 ppb in suburban Massachusetts to 61±6 ppb in coastal Connecticut. This is a large variation over a short distance for a value that is supposed to reflect relatively unvarying background ozone.*

The referee correctly points out that the derived $y_0$ values in Table 2 of the manuscript for the low altitude sites vary from 41±10 ppb in suburban Massachusetts to 61±6 ppb in coastal Connecticut and suggests that this variation is large. This large variation arises from two sources. First, as reflected in the 95% confidence limits given in the table, there is statistical uncertainty in deriving the parameter values from the fits of Equation 1 to the ODVs; the confidence limits for the two extreme results that the referee quotes indicate that the statistically significant portion of the variation is likely not as large as the range in the extreme

values might indicate. Second, as discussed in the paper, some of the variation does arise from real departures of the derived $y_0$ values from the true U.S. background ozone. Figure 2 below illustrates an analysis of the distribution of the derived $y_0$ values that allows these two sources of variation to be examined separately. The ordinate scale of this figure is designed so that a normal distribution will lie on a straight line. In this figure, 13 of the 17 derived $y_0$ values do define a normal distribution with a median of 47.7 ppb and a standard deviation of 4.5 ppb. We believe that the median of the 13 values accurately reflects U.S. background ozone and the standard deviation reflects the uncertainty in determining each $y_0$ value. This result is consistent with the 45.8 ± 1.7 ppb estimate of the U.S. background ODV given in the manuscript. The variation of the results from these 13 groups of sites is consistent with the statistically derived confidence limits given in the manuscript. The highest 4 of 17 derived $y_0$ values in Figure 2 define a high value tail; these are the 4 $y_0$ values that are discussed separately in the manuscript. Figure 2 nicely illustrates the value of our analysis based upon the simple model encapsulated in Equation 1 – the large majority of the results provide a robust estimate of the regional U.S. background ozone, and deviations from a uniform result point toward issues that deserve further investigation.

[Figure]

**Figure 3:** Cumulative probability plot of the $y_0$ determinations listed in Table 2 of the manuscript. The line shows a linear regression fit to the open points, which defines a normal distribution with the median and standard deviation annotated in the figure. The four points at the higher $y_0$ values correspond to the colored curves in Figure 8 of the manuscript.

*Changes incorporated in revised manuscript:* Figure 2 is now included as Figure S10 in the supplement, and a short description of this analysis has been added to Section 3.2.1 of the manuscript.

*One interpretation is that the Connecticut "background O3" includes a lot of ozone generated in the New York City area, and that therefore the simple exponential model cannot determine US background. The author chose a different interpretation, and re-set the US background to a lower value, which increased the US anthropogenic component to more acceptable levels. This portion of the analysis is not convincing, and seems to be an attempt to compensate for the simple model's shortcomings. In fact, it is essentially an admission that the original simple model cannot be used to distinguish between US background ozone and US anthropogenic ozone.*

This interpretation suggested by the referee cannot be correct. Connecticut ODVs certainly do reflect a lot of ozone generated in the New York City area. However, this New York City generated ozone has decreased over the years; thus, our analysis would properly include it in Connecticut's time varying term ($A*\exp\{-(\text{year-2000})/t\}$), not in Connecticut's constant $y_0$ term. The simple exponential model can indeed determine U.S. background ozone, within the caveats thoroughly discussed in the manuscript.

The referee does not correctly describe the reasoning in our manuscript. We followed two clearly distinct steps. The first step is the purely mathematical fitting of Equation 1 to the observed ODVs. The success of this step is judged by the good agreement between the fits and the observations. The second step is the physical interpretation of the A and $y_0$ parameters. As discussed in the manuscript and illustrated in Figure 3 above, the derived $y_0$ values provide substantial information regarding the U.S. background ozone, but we do not argue that the two quantities are identical in all situations. Hence, the four values in the high value tail, which includes the Connecticut $y_0$ value discussed by the referee, require separate discussion. In the physical interpretation of the A and $y_0$ parameters we do not re-set the U.S. background ozone to a lower value; rather we discuss why the derived Connecticut $y_0$ value is not equal to the U.S. background ozone. This is not an attempt to compensate for the shortcomings of the simple model; rather it is an identification of an issue identified by the simple model that deserves further investigation. It is not an admission that the original simple model cannot be used to distinguish between US background ozone and US anthropogenic ozone; in the majority of the ODV time series investigated the model does provide a robust distinction, and the few exceptions identify interesting issues.

***Changes incorporated in revised manuscript:*** No further discussion of this issue has been added to the manuscript.

*Ultimately, I have concluded that the assertions claiming that US background and anthropogenic increment can be derived from the simple exponential function are not compelling, especially for the northeastern states. It is possible that the analysis could be re-worked, by changing the ozone metric to the annual 98th percentile, but the failure of the method to derive the three parameters of the model even with three-year running averages, and its inability to distinguish between US background and US anthropogenic in the northeastern states suggests that this simple approach is flawed for the regions to which it has been applied in this study. I thought that the study was interesting, and the author did a commendable job in explaining the uncertainties and possible shortcomings of the approach. This admirable transparency in describing the methods is worthy of emulation, but I do not think the study should be published in its present form.*

The author appreciates the referee's positive comments. However, as discussed in detail above and in the response to comment 2 of referee #1, the referee's comments do not raise valid concerns regarding the analysis presented in the manuscript. The referee's overall recommendations stated here are not justified.

The critical comments provided by the referees have sharpened the analysis presented, and formulating the above responses has (hopefully) clarified the discussion. All of the objections raised by the referees have been answered. This is an important paper; please consider this resubmitted manuscript for publication.

[revised manuscript text omitted]

---

## Referee Report (RR1)

Review of acp-2018-1174
  "Estimating background contributions and U.S. anthropogenic enhancements to
  maximum ozone concentrations in the northern U.S." by D. D. Parrish

There is a considerable amount of interesting and potentially valuable information and
analysis in this manuscript, but it is not presented in a form which allows the reader to
appreciate its value.

In addition to the difficulty I had in reading this work and trying to extract its main
points, I, like the other two reviewers, think the geophysical interpretation of the
mathematical analysis may be overstated beyond what is truly warranted by the data
presented. Perhaps this feeling is exacerbated by the repetition of sections of text, but for
me it fundamentally derives from the significant difference in the temporal evolution of
ODVs over the entirety of the record for the regions studied here in comparison to the
"reference" region of southern California.

I have four points of concern with regard to the content, which I think any resubmitted
manuscript should address:

1) The inability of the single exponential analysis to describe the NEUS and MWUS data
before 2000 seems to cast doubt on the geophysical interpretation of the mathematical
fitting parameters and/or on the rigid adherence to the California tau parameter, which is
derived in Parrish et al. 2017 (JGR) from observations that are well-described back to
1980. Pg. 13, line 4 ascribes this change in the temporal behavior in the NEUS to
regulatory efforts around the turn of the millennium in the eastern US, thereby suggesting
that the California behavior (or at the very least the time constant) should not be used as a
model for the present region(s).

It seems that finding $y_0$ values is a primary goal of this work, so the author must present
a more complete sensitivity analysis than 10% change in tau mentioned briefly in section
4.2. Given the shortness of the data record being fit, what is the range of tau values that
could reasonably describe the observed decay? If the time constant were set to, e.g., 10
yrs. or 40 yrs., instead of 21.9 yrs., how much would $y_0$ change?

The alternative analysis approach described in AC-3 may be worth including in a revised
version of this manuscript, if doing so can be accomplished in a concise and compelling
way. (Could a revised manuscript be structured around the alternative method as the
primary analysis approach?) This alternative method seems to provide similar insight into
the data without requiring the use of the CA tau parameter. The author would, however,
still need to discuss the implications and limitations of analyzing only the recent portion
of the data record, regardless of the method used.

2) In light of the author's 2107 GRL paper showing that the transported contribution to
NA background ozone is now decreasing, I would like to see a discussion of how those

findings do or do not affect the interpretation here, given that a constant y0 value is a fundamental presumption of the present analysis.

3) The three low-altitude "exceptions" should not be excluded from discussion. If they are not failures of the method but rather sites in a category of their own (where the "y0 value is not equal to the U.S. background ozone" –AC-2), they should be explained and analyzed, not buried in a sentence in section 4.1. Why is y0 different at these sites than others in NEUS (or at least for the CT sites)? What does y0 represent in these cases, if not US background ODV? Or is this just merely evidence that the US background ODV has different values in different locations?

Why does NY/max (y0=53) get singled out as an exception, but Maine/Cadillac Mtn (y0=52) does not? To my eye, Fig. 2 in AC-2 suggests Maine could equally belong to the category containing NY/max. What criteria were used to determine which groups of sites belonged in the category of "exceptions"?

4) Why are all the remaining USNE y0 values then averaged together, despite the fact that they have a wider spread than the Rural West values which are never averaged together in Section 3.1? The discussion of coastal sites in the text led me to expect a sub-category that included NH/coastal, MA/coastal, and ME/coasts, rather than just having all those sites lumped into a single average y0 value with the inland locations.

Comments related to Presentation:

Substantial revision of the text will allow the reader to focus on the important messages the author wishes to convey. Perhaps a colleague with "fresh eyes" can help the author frame the discussion for a scientist, rather than for a local expert at a regulatory agency.

I recommend that the author focus first on presenting only the western and northeastern US data, followed by the method in brief. The caveats, while important, are distracting when presented in the Introduction. The sections which discuss limitations should all be combined together somewhere in the body of the manuscript and only be enumerated once.

There is a tremendous amount of detail presented in the Introduction that does not really build the author's case but which does consume the reader's attention and capacity to manage information (e.g., page 3, lines 17-32 present many names and numbers that seem to require a deep understanding and attention. Yet only a few pieces of information from this paragraph are truly critical to understanding and appreciating the message.)

Similarly, the other two geographic regions do not seem critical to present in such detail. Figure 1 and Figure 2/top panels are not the most important figures, but by placing them

so early in the manuscript the implication is that they should be read carefully and all four panels digested fully.

After spending so much effort to read the early sections of the manuscript, I had little patience or focus left by the time I got to the discussion of A and A* values – which surely are more central to the message of the study than are the nuances of the geography of Martha's Vineyard, for example.

There are many examples of redundancy throughout the text where content is repeated in essentially the same format as presented in earlier sections.

In conclusion, in its present form, I do not believe this manuscript meets the standards of *Atmospheric Chemistry and Physics*, nor the expectations of its readers. But I do believe it could become an interesting contribution to the community's understanding of ozone trends if the author addresses the substantive concerns identified above (as well as some or all of those identified by R1 & R2) and invests in crafting a more streamlined manuscript that is much easier for the reader to understand and digest.

References:
Parrish, D.D., Young, L.M., Newman, M.H., Aikin, K.C., and Ryerson, T.B.: Ozone design values in Southern California's air basins: Temporal evolution and U.S. background contribution. Journal of Geophysical Research: Atmospheres, 122, 11,166–11,182. https://doi.org/10.1002/2016JD026329, 2017.

Parrish, D. D., Petropavlovskikh, I., & Oltmans, S. J. (2017). Reversal of longterm trend in baseline ozone concentrations at the North American West Coast. Geophysical Research Letters, 44. https://doi.org/10.1002/2017GL074960

---

## Referee Report (RR2)

August 2019 Review of acp-2018-1174
  "Estimating background contributions and U.S. anthropogenic enhancements to maximum ozone concentrations in the northern U.S." by D. D. Parrish and C. A. Ennis

This manuscript is much improved! My concerns have been adequately addressed, and the readability is an order of magnitude better than the last version. This work is now presented in a way that others can understand and evaluate it. I look forward to the substantive and reasoned discussion which may follow in ACP and in the community at large.

Given the substantial amount of new text that the authors have generated, I have just a few comments that may improve the presentation.

A) The new section 2.3 describing the additional analyses was confusing to me.
  • Should line 11 read "we also derive $y_0$ through THREE somewhat different approaches that…" ?
  • I recommend you remove the last sentence of the first paragraph, as it is repeated in the third paragraph.
  • The third and fourth paragraphs seem to jump around a bit and are not presented in the same order as they are used later in the manuscript. Please see if this suggested re-organization describes what you intended:

Two additional approaches can approximately quantify the value of $\tau$ in the northeastern states; both of these approaches assume that constant values of $y_0$ and $\tau$ are appropriate for all ODV time series included in each analysis. First, a linear fit to the initial period of decreasing ODVs provides direct information regarding the magnitude of $\tau$ and $y_0$. The absolute value and the time derivative of Equation 1 when evaluated at year 2000 are $y_0 + A$ and $-A/\tau$, respectively. Fits to two ODV time series provide four parameters ($\tau$, $y_0$, $A_1$ and $A_2$) if the $\tau$ and $y_0$ values are the same for the two time series. Algebraic manipulation gives $\tau = - \Delta_{intercept} / \Delta_{slope}$, where $\Delta$ indicates the difference in the subscripted parameter between the two linear fits, and $y_0 = (\Sigma_{intercept} + \tau * \Sigma_{slope})/2$, where $\Sigma$ indicates the sum of the subscripted parameter from the two fits. A complication with this approach is that the linear fits to time periods of significant length give biased measures of the derivative and year 2000 value of Equation 1; however, this bias can be corrected to first order through numerical comparison of a linear fit to the selected period of the exponential fit. The second approach is described in Section 2.4 of Parrish et al. (2017a) and is adapted here to the northeastern U.S. ODV time series. It uses an iterative, non-linear regression analysis that simultaneously derives values for $\tau$ and $y_0$, plus the A parameter for each ODV time series included in the analysis. These two additional approaches help to constrain the uncertainty of the assumed value of $\tau$ (21.9 years).

B) When the results of the method shown in Fig 13 are described in section 3.3.2, there is insufficient information to allow the reader to evaluate your "bias correction". The description of that "first order" correction "through numerical comparison of a linear fit" in section 2.3 does not shed much additional light. If it is possible to give a bit more information (or a reference?), the reader can have more confidence that your "corrections" reduce uncertainty, rather than increase it.

And for clarity of communication, I would suggest that "intercept" is not strictly the correct noun here, as the plots you show in Figure 13 would have fitting parameters with an intercept at Year 0. I recommend you either replot with the horizontal axis as Year Minus 2000 instead of Year, or stick to using the descriptor "value" or "absolute value at Year 2000" (e.g., revised manuscript, page 8, line 4). This will also remove any confusion about if the "intercept" for the Maine data set is to be chosen at 2000 or at 1991.

C) "Variance" is discussed often in this manuscript, and it seems that the meaning is slightly different in different contexts. On page 6, variance (in units of ppb) is described in terms of RMSD between a dataset and its fit. But then on page 9 and in Table 1, variance has units of ppb^2 and appears to be (but is never actually*) defined as the square of the standard deviation of a dataset. Then both types are used back and forth through the remainder of the manuscript. It's especially tricky to parse at the top of page 19, where you are using both the ppb^2 values (like 251 and 13.4), but are also referring to Figure 9, which gives RMSDs of 3.5 and 5.6 ppb. I don't have a nice, tidy suggestion, but I would ask that the authors take a few minutes to think about how they might make their dual-use of this word a little bit easier for the reader.

*It is given quite succinctly on page 6 of the Response document, but I couldn't find it in the manuscript itself. I recommend including it in the manuscript, especially since the rounding effects are just enough to make it questionable (e.g., pg 18: 3.7 * 3.7 = 13.69 = 13.7, not 13.4).

And by the way, the Figure 3 legend says 252 ppb^2, not 251 ppb^2.

D) And a few small suggestions:

- Pg 9, line 14, add a comma: "Figures 3 and 4, and averages with standard deviations…"
- Pg 13, line 15 change to: Figure 10 plots the time series of these state maximum ODVs recorded in each year with respective fits over the 2010-2017 period.
- Is that really supposed to be 2010? In Fig 10, the solid fit lines start at 2000.
- Pg 13, line 19 change to: only the largest of the state's ODVs in a given year… {to match singular "across the state" later in the sentence"}
- Pg 13, line 27: is this "NYC urban maximum" a new subset? Or is it the same as the data which generated the red dashed line in Fig 7? This is not a big deal, but I got distracted for a while trying to figure it out.
- And a related question: In Fig S13, Does the fact that the 1:1 line goes through NJ data points at all values of "NYC urban max" indicate that the

"NYC urban max" data is actually entirely from NJ? (I understand the geography; that's not my point. If all the "reference" data is contained in the data plotted against it in Fig S13, that seems a bit circular.)

- Pg 13, line 28, remove comma after "is selected".
- Suggested clarification and nuance regarding spatial variability of ODVs on page 16: the derived US background ODV has significant variability on a continental scale. Within..... significantly smaller than in any of the western US regions, but shows no discernable spatial variability within this region. For context... NAAQS of 70 ppb. In contrast, in the northeastern US the A...
- Pg 18, top line has an extra space
- Pg 18, line18, strike "by" before (Fiore et al., 2015)
- Pg 18, line 21: change "limit" to "limited"
- Competing Interests section needs to be updated, as does the last line of the Acknowledgements.
- Fig 9 caption: The explanation of dashed, dotted, and y0 is awkward. Maybe try rewriting with dashed, then dotted, then y0 last.
- Fig 14, vertical axis label: Is this really US background, not NAB?
- Supplemental, pg 2, line 11: One...area
- Supplemental, line 21: some sites in the NY...
- Supplemental, line 32: neighboring states until 2013.
- Figs S3-S10: the gray circles are a bit too faint to really see. Could they be darker?
- Fig S10 legend, lower panel: green sites are called "interior" or "rural inland" elsewhere. I recommend choosing one of those in the lower legend for consistency.
- Fig S16: the blue dashed line is different from all the other blue dotted lines.

---

## Author Response (AR2)

**Author's Response to interactive comments by Editor and 4 Anonymous Referees on "Estimating background contributions and U.S. anthropogenic enhancements to maximum ozone concentrations in the northern U.S." by David D. Parrish**

The author greatly appreciates the many comments regarding this paper, the time and effort that clearly went into these reviews, and the Editor's recommendations. We have taken these comments to heart, and have made significant changes to the analysis and the manuscript. Responses to all comments follow, and where appropriate the manuscript has been revised as described herein, and indicated in the "tracked changes" manuscript copy at the end of this response.

To organize this response, we first give an overview of the four major issues raised in the reviews, and briefly summarize how we have addressed each. This is followed by more extensive discussion of each major issue. Finally, point-by-point responses to all reviewer comments are given, often simply referring back the more extensive discussion of the major issues.

**Overview:** Major Scientific Issues Raised by the Reviewers and Our Response**

A. Reviewers questioned our choice of Equation 1, which incorporates an exponential decrease above a constant offset, to quantify urban ODV changes.

Response and Changes Made: We agree that additional support for using Equation 1 will strengthen the paper. We discuss this choice from four perspectives that support its use, provide context for the analysis, and guide the interpretation of the derived parameter values. We have expanded the discussion in Section 2.2 of the paper to address this comment.

B. Reviewers questioned our decision to set  $\tau$  equal to 21.9 years (i.e., the value derived in a southern California ODV analysis).

Response and Changes Made: The reviewers raised valid concerns and questions about this choice of  $\tau$ . In response to their comments, we have now included two approaches to estimate  $\tau$  in the northeastern U.S. to provide support for the assumed value of  $\tau$  of 21.9 years. One approach includes a linear analysis, similar to that suggested by Referee #1 in the first round of reviews. These 2 approaches provide values of  $\tau$  that, within uncertainties, overlap the original value of 21.9 years for  $\tau$ . We describe these analyses in Section 2.3 and the results in Section 3.3. The original analysis remains in the main text as our primary choice, but we hope the reviewers will agree that this choice is now is strengthened by the additional analyses. However, we acknowledge that these additional analyses do support other reviewer comments suggesting that the uncertainties in our results are likely larger than we stated. Therefore we have provided additional uncertainty analysis of the magnitude of  $\tau$  and the derived value of  $y_{\theta}$  in the northeastern U.S. The result is a larger confidence interval ( $\pm$  3.0 ppb for  $y_{\theta}$ ), which is incorporated into the uncertainty discussion in the revised manuscript (new Section 3.3.4).

C. Reviewers point out that our analysis neglects that temporal changes likely occur in the background ODV contribution (i.e., our assumption of a constant  $y_0$  may not be valid).

Response and Changes Made: The discussion in Section 3.1 that shows  $y_0$  has remained nearly constant in the western rural states over the past 3 decades has been clarified. The variance due to varying background ODV contributions is shown to be small relative to that from the changing anthropogenic contribution in the northeastern states. A detailed comparison of the ODVs at western rural sites with a baseline ozone site on the U.S. west coast is included in the expanded discussion below. These comparisons show that applying Equation 1 with an assumed constant background is well justified for the northeast U.S.

**D. Reviewers were critical of the success of Equation 1 in describing spatial and temporal ODV distribution.**

Response and Changes Made: U.S. ODVs result from a very complicated system of chemical and physical atmospheric processes. Equation 1 provides a basis to quantify the role of only one of these processes - changing anthropogenic emissions. We agree with the reviewers that judging the degree of success (and failure) of this description is important for the informative interpretation of the derived parameters. Section 3.3 of the paper discusses this judgment, and that discussion is further extended below.

**Discussion of Major Issues**

**Issue A: Choice of exponential decrease above a constant offset (Equation 1) to quantify urban ODV changes**

Four considerations guide our choice of Equation 1, which contains an exponential decrease to quantify long-term changes of U.S. urban pollution ozone contributions:

 Examination of ozone observations in U.S. urban areas reveals similarity of trends throughout the country, with general decreases in all areas, albeit occurring at different rates and after varying onset times. Our goal is to quantitatively evaluate these decreases in order to elucidate the processes controlling U.S. urban ozone concentrations. Fitting observational data to a simple functional form is a common tool utilized for quantitative observational analysis; linear trend analysis (i.e., fitting observational data to a linear function) is one example. In this work we select a functional form that a) is consistent with our general understanding of urban ozone, and b) is as simple as possible (i.e., has the fewest unknown parameters). There are three general features of ozone in U.S. urban (and rural) areas that must be consistent with the selected functional form: first, there is a background contribution, below which ozone cannot be reduced by U.S. precursor emission controls; second, maximum ozone concentrations have been enhanced above that background due to pollutiongenerated ozone; and third, the pollution enhancements are presently generally decreasing due to on-going emission reductions. To the best of our understanding, Equation 1 of our paper,

$$O_3 = y_0 + A \exp\{-(y_{ear}-2000)/\tau\},$$
(1)

with three undetermined parameters, is the simplest possible functional form consistent with these three features. (A linear fit with only two undetermined parameters – slope and intercept – is simpler, but a linear fit to a decreasing trend will eventually go negative, and therefore cannot fit a positive background contribution).

Equation 1 gives excellent fits to the last two or more decades of ozone observed in U.S. urban areas. Parrish et al. [2017a] show that Equation 1 captures 98.4% of the total variance in the maximum ODVs in 7 southern California air basins over the 1980 to 2015 period. Figure 9b of the revised manuscript shows that Equation 1 captures 89% of the total variance

in the maximum ODVs in the 8 northeastern U.S. states from 2000 to 2017. These fits suggest that the derived parameter values must contain useful information regarding the processes driving temporal evolution of U.S. urban (and rural) ozone concentrations.

- 3) Since Equation 1 is chosen to be consistent with our general understanding of urban ozone changes, the parameter values can be directly related to the processes controlling U.S. ozone concentrations. For example,  $y_0$  in Equation 1 is designed to quantify the background ozone contribution. However, this design does not necessarily mean that a parameter value can be directly interpreted as Equation 1 implies; the simple, direct interpretation of  $y_0$  (as well as the other two parameters) must proceed with careful consideration of potentially complicating factors.
- 4) A simple intuitive argument suggests that an exponential decrease in the pollution ozone contribution is to be expected. When emission controls are initiated, early progress can be rapid, since there are large emission sources that evolved with no plans for their control. As an illustrative example, it might be possible to reduce the pollution ozone contribution by half in the first 15 years of control efforts. After that period reducing emissions will be harder, since the most easily controlled emissions have been addressed. During the next 15 years, it might be possible to again reduce the remaining pollution ozone contribution by half (i.e., to 25% of the original). If this example were realistic, then the emission reductions would follow Equation 1 exactly with  $\tau = 21.6$  years, close to the value of  $\tau = 21.9 \pm 1.2$  years reported by Parrish et al. [2017a]. Simply put, the expected increasing difficulty in reducing emissions by an absolute amount implies an approximately exponential decrease in the impact of those emissions. Equation 1 captures this expectation.

As the above discussion makes clear. Equation 1 can only be applied to time periods after maximum urban ozone concentrations were reached, and emission controls had advanced to the point that ozone concentrations began to decrease consistently. It is true that this decrease began ~1980 in California, but not until ~2000 in many regions of the northeast U.S.; thus the present analysis is primarily limited to post-2000. Four comments are relevant here. First, the single exponential fit must fail at some point if extended to earlier times, as a maximum urban ozone concentration must have occurred at some time in the past in any given region. Second, some regions in the northeast do approximately follow Equation 1 to earlier times, e.g., Connecticut data in Figure S3, where the ODV maximum of 169 ppb (off scale in figure S3) occurred in 1982. Other northeastern U.S. regions are fit for years before 2000 (see Table 2). Third, it would be of interest to investigate the timing of the ozone maxima in different regions (and to understand the possible role played by evolution of urban monitoring site networks), but such investigation is beyond the scope of this paper. Finally, the literature contains many examples of analyses based on linear fits to ozone trends after maxima were reached [e.g., see Fig. 17a of Lin et al., 2017; Fig. 5 of Jaffe et al., 2018]; the inability of the linear trend to describe data before the maxima is not interpreted as casting doubt on the geophysical interpretation of the mathematical fitting parameters for the period after the maximum – the analysis presented in this paper must be considered similarly.

This discussion leads to the conclusion that fitting to Equation 1 is a valid approach for observational analysis of ozone in both California and the northeast U.S., but that complicating factors must be carefully considered in geophysical interpretation of the derived values of the

fitting parameters. In response to reviewer comments, our revised paper goes further to acknowledge and explain these complications and assumptions.

**Issue B: Choice of $\tau$ set equal to 21.9 years (i.e., the value derived in a southern California ODV analysis).**

It is difficult to precisely determine the three parameters of Equation 1 from the relatively short (~2000-2017) record of consistently decreasing northeastern U.S. ODVs. The primary analysis (retained in the revised paper) is based on the simplest approach – assuming that  $\tau$  in the northeastern U.S. is the same as derived for southern California ( $\tau = 21.9 \pm 1.2$  years). To test this assumption and address reviewer comments, we now include two analyses to investigate the  $\tau$  value appropriate for the northeastern U.S.; both analyses assume constant  $\tau$  and  $y_0$  values across that entire region.

The first analysis is an iterative, non-linear regression analysis similar to that described in Section 2.4 of Parrish et al. [2017a] that simultaneously fits seven data sets to Eq.1 to allow extraction of nine parameters. The data sets are the 2000-2017 maximum ODVs recorded in seven states shown in Figure 10 of the revised paper. (Note that since its recent ODV behavior is different from the other states, as discussed in the paper, Connecticut is not included here.) This fit gives a value for  $\tau$ , a value of  $y_0$ , and values of A for each of seven states (Table 1). The derived  $\tau$  value is larger than the California value, although they agree within the derived 95% confidence limit. Correspondingly, the derived  $y_0$  value is

| Table 1. Results of iterative, |                |  |
|--------------------------------|----------------|--|
| non-linear regression analysis |                |  |
| $\tau$ (years)                 | $26.0\pm6.0$   |  |
| $y_{\theta}$ (ppb)             | $41.8 \pm 3.0$ |  |
| State                          | A (ppb)        |  |
| Maine                          | $51 \pm 10$    |  |
| Massachusetts                  | $56 \pm 10$    |  |
| New Hampshire                  | $46 \pm 10$    |  |
| New Jersey                     | $66 \pm 10$    |  |
| New York                       | $61 \pm 10$    |  |
| Rhode Island                   | $55 \pm 10$    |  |
| Vermont                        | $39 \pm 10$    |  |

smaller than derived earlier ( $45.8 \pm 1.7$  ppb), and the *A* values are larger (compare to *A*\* values in Table 3 of manuscript).

The second approach uses linear fits to the initial period of decreasing ODVs. Figure 1 shows three of the ODV data sets selected to give the largest possible contrast in the contribution of pollution ozone and the smallest variability about the fits to Equation 1. Linear fits are shown for the initial periods (2000-2017 for two, and 1991-2017 for the data set with a large gap in the ODVs near 2000). The absolute value and the time derivative of Equation 1 when evaluated at year 2000 are  $y_0 + A$  and  $-A/\tau$ , respectively. Four parameters ( $\tau$ ,  $y_0$ ,  $A_1$  and  $A_2$ ) are required to fit two ODV data sets if the  $\tau$  and  $y_0$  values are common. The slopes and intercepts (given in Table 2) of the two linear fits in each panel of Figure 1 provide 4 relationships that allow the calculation of the 4 exponential fit parameters. Algebraic manipulation gives expressions for  $\tau = -\Delta_{intercept} / \Delta_{slope}$ , where  $\Delta$  indicates in the difference in the subscripted parameter between the two linear fits, and  $y_0 = (\Sigma_{intercept} + \tau^* \Sigma_{slope})/2$ , where  $\Sigma$  indicates the sum of the subscripted

parameter from the two fits. These equations allow calculation of estimates of  $\tau$  and  $y_0$ from the two linear fits in each panel of

| Table 2. Results from linear fits to early years of three ODV data sets |                 |                 |                  |                  |
|-------------------------------------------------------------------------|-----------------|-----------------|------------------|------------------|
| Data set                                                                | intercept (ppb) |                 | slope (ppb/yr)   |                  |
|                                                                         | original        | corrected       | original         | corrected        |
| New York City                                                           | $108.6\pm3.8$   | $110.1 \pm 3.8$ | $-2.16 \pm 0.38$ | $-2.91 \pm 0.38$ |
| Vermont                                                                 | $78.2 \pm 1.8$  | $79.0 \pm 1.8$  | $-1.07 \pm 0.18$ | $-1.44 \pm 0.18$ |
| Maine NE coast                                                          | $70.6 \pm 1.5$  | $69.8 \pm 1.5$  | $-0.89 \pm 0.13$ | $-1.05 \pm 0.13$ |

Figure 1. One complication is that the linear fits give biased estimates of the year 2000 slopes and intercepts required in the above derivation. First order corrections for these biases are made from numerical evaluation of linear fits to an exponential function, and included as corrected parameters in Table 2. The resulting estimates of  $\tau$  are 21.1 ± 5.9 and 21.7 ± 5.0 years for the fit parameters in the upper and lower Figure 1 panels, respectively.

In summary, the  $\tau$  estimates from these two analyses do agree with the southern California value within their confidence limits, indicating that there is no evidence for a different exponential rate of decrease of U.S. anthropogenic ODV enhancements between these two U.S. regions. Importantly, these two analyses give estimates of  $y_0$  that are independent of the assumed  $\tau$  value.

**Issue C: Neglect of temporal changes in the background ODV contribution (y0).**

Parrish et al. [2017b] report that seasonal average baseline ozone concentrations at the U.S. west coast were increasing before a maximum was reached in the mid-2000s, and are now decreasing. Since transported baseline ozone is the primary source of the U.S. background ODV contribution, assuming a constant  $y_0$

contribution as done in the present analysis may compromise the results. Figure 2 compares the ODVs recorded at Lassen Volcanic NP [one of the sites considered by Parrish et al. 2017b] with

those at two of the rural western state sites. Table 3 gives the coefficients of quadratic polynomial fits, such as Parrish et al. [2017b] utilized to quantify these long-term changes. There are no statistically significant differences in the temporal trends of these ODVs, as quantified by the *b* and *c* parameters (although there are differences in the absolute concentrations as quantified by the *a* parameters.) Figure 2 is consistent with Section 3.1 of the paper, where we conclude that the ODVs in the rural western states are dominated by the U.S. background contribution.

The impact of neglecting background temporal changes can be judged by comparing the ODV variance over the 2000-2017 period between the northeastern states (Figure 3 of the manuscript), where

**Figure 1.** Comparison of linear and exponential fits to three ODV data sets.

**Figure 2.** Comparison of ODVs recorded at three relatively isolated rural sites.

anthropogenic ODV contributions dominate, and North Dakota (Figure 4 of the manuscript), where only the U.S. background contribution contributes significantly. The sample variance in each region is calculated from the square of the standard deviation -  $107.3 \text{ ppb}^2$  and 3.7 ppb2 in the northeastern states and North Dakota, respectively. The North Dakota variance can be taken as a rough approximation for the background ODV contribution to the ODV variance in the northeastern states, which would amount to a contribution of no more than a few percent. Table 3 compares the variance of ODV series in Figure 2, which are similar to that of North Dakota. In summary the neglect of temporal changes in  $y_0$  in the northeastern states is of negligible consequence in the determination of the  $y_0$  and A values in that region.

| Table 3. Comparison of quadratic fits to ODVs at a baseline site on the U.S. west coast and |                |                           |                           |            |                              |
|---------------------------------------------------------------------------------------------|----------------|---------------------------|---------------------------|------------|------------------------------|
| two sites in the rural western states                                                       |                |                           |                           |            |                              |
| Site                                                                                        | a (ppb)        | b (ppb yr -1 ) | c (ppb yr -2 ) | RMSD (ppb) | Variance (ppb 2 ) |
| Lassen Vol. NP                                                                              | $73.0 \pm 1.6$ | $-0.01 \pm 0.20$          | $-0.027 \pm 0.019$        | 3.0        | 9.9                          |
| Yellowstone NP                                                                              | $65.6 \pm 1.8$ | $-0.14 \pm 0.49$          | $-0.005 \pm 0.027$        | 1.7        | 4.4                          |
| Glacier NP                                                                                  | $55.1 \pm 0.7$ | $0.11 \pm 0.10$           | $-0.013 \pm 0.008$        | 1.2        | 2.0                          |
|                                                                                             |                |                           |                           |            |                              |

**Issue D: Judging the success of Equation 1 in describing spatial and temporal ODV distribution**

U.S. ODVs are determined by a very complicated system of chemical and physical atmospheric processes. Equation 1 ignores nearly all (including varying influences from stratospheric intrusions, increasing Asian anthropogenic emissions, increasing frequency of wildfires, variable meteorological conditions, increasing methane concentrations, a warming climate, etc.), while considering only two ozone contributions - background ozone and anthropogenic ozone pollution. Further, these two contributions are treated in a simplified manner – the background contribution is assumed constant, and the anthropogenic contribution is assumed to have decreased exponentially since the time that urban ozone concentrations began decreasing consistently. This simple approach is only justified if it successfully quantifies the spatial and temporal distribution of U.S. ODVs.

The discussion of Figure 9 in Section 3.2.2 of the paper demonstrates the success of Equation 1 in capturing the variability of 18 years of ODVs recorded in 8 northeastern U.S. states. These ODVs include 1719 individual values. Notably, there are no free parameters in the two fits shown in Figure 9: constant values of  $y_0$  and  $\tau$  were derived from other analyses, and the A values are interpolated from Figure 9 (for the individual monitoring sites) and taken from Table 3 for the state maxima. Equation 1 then captures 89% and 71% of the variance in state maxima and all 1719 ODVs, respectively. This indicates that no more than the remaining 11% and 29% of the variance in state maxima and all 1719 ODVs, respectively, can be accounted for by the combined influences of:

- The atmospheric processes listed above, and ignored by Equation 1.
- Departures from the assumed constant background contribution and the assumed simple exponential decrease of the anthropogenic contribution at the assumed California  $\tau$  value.
- Spatial variability of the *A* parameter not captured by the contour plot of Figure 9.
- Any measurement issues such as temporally evolving monitoring networks giving inconsistent temporal changes, instrumental errors, etc.

The success of Equation 1 in quantitatively describing the spatial and temporal distribution of U.S. ODVs is a strong indication of the utility of the derived parameters for estimating aspects of the ODV distribution.

**Point-by-Point Responses**

Notes:

- The comments from the Editor and the Referees are reproduced below in black regular font with our responses in *blue italic font*.
- Our revisions have been guided by several of the constructive comments and suggestions offered by Referee #4; therefore we have responded to that review first in our discussion below.
- Some of the Reviewers' comments have been divided into separate paragraphs to facilitate responses; (*Para*) indicates where such divisions have been made.

**Anonymous Referee #4**

Review of acp-2018-1174

"Estimating background contributions and U.S. anthropogenic enhancements to maximum ozone concentrations in the northern U.S." by D. D. Parrish

There is a considerable amount of interesting and potentially valuable information and analysis in this manuscript, but it is not presented in a form which allows the reader to appreciate its value.

In addition to the difficulty I had in reading this work and trying to extract its main points, I, like the other two reviewers, think the geophysical interpretation of the mathematical analysis may be overstated beyond what is truly warranted by the data presented. Perhaps this feeling is exacerbated by the repetition of sections of text, but for me it fundamentally derives from the significant difference in the temporal evolution of ODVs over the entirety of the record for the regions studied here in comparison to the "reference" region of southern California.

I have four points of concern with regard to the content, which I think any resubmitted manuscript should address:

1) The inability of the single exponential analysis to describe the NEUS and MWUS data before 2000 seems to cast doubt on the geophysical interpretation of the mathematical fitting parameters and/or on the rigid adherence to the California tau parameter, which is derived in Parrish et al. 2017 (JGR) from observations that are well-described back to 1980. Pg. 13, line 4 ascribes this change in the temporal behavior in the NEUS to regulatory efforts around the turn of the millennium in the eastern US, thereby suggesting that the California behavior (or at the very least the time constant) should not be used as a model for the present region(s).

Please see the **Issue** *A* discussion above regarding the use of the single exponential and the geophysical interpretation of the derived parameters. Our revisions acknowledge the complications that must be considered when applying the exponential. Please also see **Issue B**

**and the next point below, which describes our additional tests of the primary approach of using the California tau parameter.**

It seems that finding y0 values is a primary goal of this work, so the author must present a more complete sensitivity analysis than 10% change in tau mentioned briefly in section 4.2. Given the shortness of the data record being fit, what is the range of tau values that could reasonably describe the observed decay? If the time constant were set to, e.g., 10 yrs. or 40 yrs., instead of 21.9 yrs., how much would y0 change?

Please see the **Issue B** discussion above regarding the appropriateness of application of the California time constant to the northeastern U.S. We now incorporate two additional analyses that provide insight into the range of tau values that apply to the northeastern U.S., and an expanded confidence limit for  $y_0$  is now discussed in the revised manuscript.

The alternative analysis approach described in AC-3 may be worth including in a revised version of this manuscript, if doing so can be accomplished in a concise and compelling way. (Could a revised manuscript be structured around the alternative method as the primary analysis approach?) This alternative method seems to provide similar insight into the data without requiring the use of the CA tau parameter. The author would, however, still need to discuss the implications and limitations of analyzing only the recent portion of the data record, regardless of the method used.

Please see Issue B and the discussion of additional analyses completed in response to this comment and comments of the other Reviewers. The alternative analysis approach is included in our revised manuscript (Section 3.3.1). We have worked to improve the concise and compelling way the material is described. Given that each approach involves assumptions, and considering that the alternative approaches give values of tau that overlap the assumed California tau value (21.9 years) of our original analysis, we have retained our original analysis as the primary analysis in the paper. We hope Referee #4 agrees that the additional analyses have strengthened the case for our primary approach. Please see the **Issue** A discussion above regarding the implications and limitations of analyzing only the recent portion of the data record. We have briefly discussed this issue in Section 3.2 of the revised paper.

2) In light of the author's 2107 GRL paper showing that the transported contribution to NA background ozone is now decreasing, I would like to see a discussion of how those findings do or do not affect the interpretation here, given that a constant y0 value is a fundamental presumption of the present analysis.

**Please see the **Issue** *C* discussion above regarding the how changing background ODV contributions affect the interpretation of the present analysis. This issue is now specifically discussed in Section 4.2.**

3) The three low-altitude "exceptions" should not be excluded from discussion. If they are not failures of the method but rather sites in a category of their own (where the "y0 value is not equal to the U.S. background ozone" -AC-2), they should be explained and analyzed, not buried in a sentence in section 4.1. Why is y0 different at these sites than others in NEUS (or at least for the

CT sites)? What does y0 represent in these cases, if not US background ODV? Or is this just merely evidence that the US background ODV has different values in different locations?

The second paragraph of Section 4.1 now explains and analyzes these exceptions as fully as possible: "The cause of this difference is not understood. Whether this difference is simply a statistical fluctuation cannot be determined at this time; however, random fluctuations of similar magnitude are only rarely apparent in the temporal records of ODVs in the states discussed. McDonald et al. (2018) have recently discussed a class of ozone precursor emissions, i.e., volatile chemical products - including pesticides, coatings, printing inks, adhesives, cleaning agents, and personal care products - that have not been addressed by emission controls to the same extent as other emission sectors. The impact of this emission sector on ODVs has not been quantified, but is expected to be most significant in areas of largest population density, exactly the regions where the significant differences in temporal evolution of ODVs are noted." Unfortunately, further discussion is not possible without undue speculation.

Why does NY/max (y0=53) get singled out as an exception, but Maine/Cadillac Mtn (y0=52) does not? To my eye, Fig. 2 in AC-2 suggests Maine could equally belong to the category containing Maine/Cadillac Mtn. What criteria were used to determine which groups of sites belonged in the category of "exceptions"?

This raises a good point. Figure 2 of AC-2 (now included as Figure S11 in the revised Supplementary Information) indicates no clear reason to separate Maine/Cadillac Mtn from NY/max as their  $y_0$  values are similar. However, the separation must be made somewhere, and these two data sets lie near the "knee" of the cumulative probability distribution plot where the separation naturally belongs. The differentiation was made for two reasons. First, the other two low-altitude "exceptions" are either highly urbanized or just downwind from highly urbanized areas, so it makes sense to include NY/max with these two exceptions. In contrast, Maine/ Cadillac Mtn is one of the furthest downwind areas, so it makes sense to include it with the other areas downwind form the major urban center. Second, the confidence limits on the Maine/Cadillac Mtn  $y_0$  are so wide that it may well lie in the range of the  $y_0$ 's found at the other downwind areas. In the interest of brevity, this discussion has not been added to the revised paper.

4) Why are all the remaining USNE y0 values then averaged together, despite the fact that they have a wider spread than the Rural West values which are never averaged together in Section 3.1? The discussion of coastal sites in the text led me to expect a subcategory that included NH/coastal, MA/coastal, and ME/coasts, rather than just having all those sites lumped into a single average y0 value with the inland locations.

Figure 2 of AC-2 (now included as Figure S11 in the revised Supplementary Information) shows that remaining USNE  $y_0$  values approximately define a Gaussian distribution with a median of 47.7 ppb and a standard deviation of 4.5 ppb. The 95% confidence limits of the remaining USNE y0 values in Table 2 range from 4 to 10 ppb (16 ppb including the Maine/Cadillac Mtn confidence limit). This rough correspondence between the standard deviation of the  $y_0$ distribution and the standard deviation that corresponds to the 95% confidence limits suggests that there is no statistically significant difference between the remaining USNE  $y_0$  values, so they are all lumped into a single average  $y_0$  value. In contrast, the Rural West  $y_0$  values do exhibit statistically significant differences, so it is appropriate to discuss them separately rather than averaging. In the interest of brevity, this discussion has not been added to the revised paper.

**Comments related to Presentation:**

We agree with the Referee's suggestions below to reduce redundancy, focus attention more fully on the main points, and reduce distractions of secondary information. We have made the following revisions to address these comments and suggestions (answers interspersed with Referee's comments):

Substantial revision of the text will allow the reader to focus on the important messages the author wishes to convey. Perhaps a colleague with "fresh eyes" can help the author frame the discussion for a scientist, rather than for a local expert at a regulatory agency.

This was an extremely helpful suggestion. A colleague not previously involved in the manuscript was asked to join as a coauthor. After working through the paper to understand all aspects of the analysis, she led the reorganization of the entire manuscript, Supplement and this response to the comments.

I recommend that the author focus first on presenting only the western and northeastern US data, followed by the method in brief. The caveats, while important, are distracting when presented in the Introduction. The sections which discuss limitations should all be combined together somewhere in the body of the manuscript and only be enumerated once.

We have focused the discussion on the two regions as suggested, which involved eliminating text from the Introduction and moving Figures 1 and 2 of the prior manuscript to the Supplementary Information.

*Caveats are now collected and discussed in one section (4.2).*

There is a tremendous amount of detail presented in the Introduction that does not really build the author's case but which does consume the reader's attention and capacity to manage information (e.g., page 3, lines 17-32 present many names and numbers that seem to require a deep understanding and attention. Yet only a few pieces of information from this paragraph are truly critical to understanding and appreciating the message.)

**We agree and we have eliminated this paragraph.**

Similarly, the other two geographic regions do not seem critical to present in such detail. Figure 1 and Figure 2/top panels are not the most important figures, but by placing them so early in the manuscript the implication is that they should be read carefully and all four panels digested fully.

**As mentioned, we agree and we have eliminated Figures 1 and 2 from the main manuscript and moved them to the Supplementary Information.**

After spending so much effort to read the early sections of the manuscript, I had little patience or focus left by the time I got to the discussion of A and A\* values – which surely are more central

to the message of the study than are the nuances of the geography of Martha's Vineyard, for example.

The Referee's point is well taken. We hope we have removed the distractions successfully so that the paper is now more focused on the central messages of the study.

There are many examples of redundancy throughout the text where content is repeated in essentially the same format as presented in earlier sections.

We agree with the Referee's suggestion. We have made the discussion more concise and removed redundancy wherever possible. Specifically, we only briefly mention caveats in the introduction, and collected the discussion in one section (4.2).

In conclusion, in its present form, I do not believe this manuscript meets the standards of *Atmospheric Chemistry and Physics*, nor the expectations of its readers. But I do believe it could become an interesting contribution to the community's understanding of ozone trends if the author addresses the substantive concerns identified above (as well as some or all of those identified by R1 & R2) and invests in crafting a more streamlined manuscript that is much easier for the reader to understand and digest.

We appreciate the very constructive criticisms of Referee #4 on the scientific issues and the presentation of the manuscript. We are also appreciative of the Referee's statement that the essence of the paper can be an interesting contribution to the community's understanding.

**Anonymous Referee #3**

Parrish uses a simple mathematical fit and an overly specific interpretation. The author has responded to previous comments by adding an acknowledgment of weaknesses of the technique and interpretation. The acknowledgment is not sufficient given the reliance of the manuscript on the specific interpretation.

As noted in our Overview, other responses above, and further responses below, we have revised our manuscript to provide alternative analyses as well as additional discussion of the assumptions of those analyses and the primary exponential analysis. Any analysis of anthropogenic and background ozone, whether observation-based or model-based, involves assumptions and simplifications. We acknowledge, as do all such studies, that the full complexity of "reality" is not represented in the analyses. Nevertheless, the application of new approaches such as the ones we have presented here serve as a useful way to advance understanding of the scientific issues and concurrently shed new light on the uncertainties inherent in all studies. We have widened the confidence limits of our results as a result of the scientific issues raised by the referees and based on our alternative analyses.

In response to previous comments, the author asserts that there is likely no persistent emissions. The NEI shows a decay of emissions from 2002 to 2014 (a comparable period) with a lifetime of 18 years (see https://gispub.epa.gov/neireport/2014/) -- which is in reasonable agreement with the authors tau. However, the interpretation requires two more assumptions. The first assumption is that the trends are explainable by single exponential decay function. The second major assumption is that ozone trends will continue responding to future reductions as they have now.

First assumption: The interpretation by Parrish is based on exponential decay of local contributions, which are related to emissions. If the emissions reduction is not reasonably fit by a single exponential function, then it is over interpretation to conclude that y0 is background. The NEI trends data shows that some sectors are increasing while others are decreasing. Individual exponential fits would create tau values ranging from -50 to 22 years. For example, "Industrial and Other Process" category has gone up Nationally by 25% over the time period being analyzed (tau=-50years). If each sector is separately allowed to grow to 66 years (3 x tau in this manuscript), using the single exponential function would predict just 12% of the sum of individual functions. The "Industrial and Other Process" sector is likely linear not exponential; even holding it constant would mean that the single exponential function would predict just 34% of the future emissions. This highlights that the exponential decay is actually the sum of equations. Even the exponential decay of certain sectors could be due to an exponential decay in controlled processes.

We make no assumption regarding the time dependence of ozone precursor emissions. Given the shortcomings of emission inventories (e.g., Miller et al. 2006), quantifying emission changes is uncertain. We do know from ambient measurements in Los Angeles that emissions have decreased approximately exponentially, but at different rates for different species (and likely for different sectors as well). For example, in Los Angeles over the past 50 years, ambient VOC concentrations have decreased at about 7.5% yr-1 or a factor of ~50 over 5 decades (Warneke et al., 2012) while NOx concentrations have decreased at about 2.6% yr-1 or a factor of ~4 over 5 decades (Pollack et al., 2013). It is widely recognized that the response of ozone to changes in precursor emissions is complicated; we do not rely upon any quantitative information regarding precursor changes.

**Please see the* **Issue** *A discussion above regarding our choice of the exponential decrease of the U.S. anthropogenic ODV enhancements.**

Suggesting that an exponential decays that corresponds with specific sector controls can be used to characterize all sector contributions is over interpretation. There is evidence of this overinterpretation in the manuscript itself. Figure 7 is an excellent example of where the interpretation is obviously flawed. If this approach were robust, you could apply it to a time interval of observations and get the same result as if you applied it again after accumulating more data. If you had applied the model in 2003 (before the NOx SIP call took effect), the "Background ODV" appears as though it would have converged on 80-90 ppb. The exponential decay after 2003 highlights that the decay is in response to controls put in place over that time period. During that same time, motor vehicle emissions have also decreased. If controls targeted all sectors, the interpretation would be more reasonable. There is no evidence provided by the author in the manuscript.

Responding to this comment is difficult, as it is not clear to which data set the referee is referring. Taking the Massachusetts coastal data set as an example, Figure 3 shows fits for three time intervals with the derived  $y_0$  values annotated: 4 years (2000-2003) as the referee suggested, 10 years (2000-2009), and the full 18-year period (2000-2017). The 4-year period gives physically unrealistic results; interannual variability gives an overall increase in ODVs over that period. Nevertheless, the derived  $y_0$  value agrees (nearly) with the other two results within the derived confidence limits, despite the implication of a negative pollution ozone contribution. When 10 vears of data are considered, there is close agreement with the 18-year period, but the confidence limits are so large that the results are of limited usefulness. This exercise does not show an obvious flaw in the interpretation, nor does it present any *evidence of an over-interpretation – close* attention must be paid to the confidence *limits in this (or really in every)* observational analysis. The exercise does show that one can apply this approach to a time interval of observations, and then get a statistically consistent (not the same) result if applied again after accumulating more data. Thus, the approach does pass this "robustness" test suggested by the referee, and this exercise demonstrates the value of additional years of data.

---

## Author Response (AR3)

**Authors' Response to interactive comments by 2 Anonymous Referees on "Estimating background contributions and U.S. anthropogenic enhancements to maximum ozone concentrations in the northern U.S." by David D. Parrish and Christine A. Ennis**

The authors greatly appreciate the additional comments regarding our paper and the Editor's recommendations. The reviews of this paper have required much more time and effort than is usually the case, and the authors appreciate everyone's willingness to make these efforts. Responses to all new comments follow, and where appropriate the manuscript has been revised as described herein, and indicated in the "tracked changes" manuscript copy at the end of this response.

**Point-by-Point Responses**

Notes:

- The comments from the Referees are reproduced below in black regular font with our responses in *blue italic font*.
- The revised draft that was the subject of the present review included some changes that were not incorporated into Figure 14. That figure has now been updated in this new revision, and the discussion revised accordingly; this change did not lead to any changes in interpretation.

**Report #3, by Anonymous Referee #4**

August 2019 Review of acp-2018-1174

"Estimating background contributions and U.S. anthropogenic enhancements to maximum ozone concentrations in the northern U.S." by D. D. Parrish and C. A. Ennis

This manuscript is much improved! My concerns have been adequately addressed, and the readability is an order of magnitude better than the last version. This work is now presented in a way that others can understand and evaluate it. I look forward to the substantive and reasoned discussion which may follow in ACP and in the community at large.

Thank you for these positive comments, and the very great effort that you devoted to your two reviews.

Given the substantial amount of new text that the authors have generated, I have just a few comments that may improve the presentation.

A) The new section 2.3 describing the additional analyses was confusing to me.

• Should line 11 read "we also derive y0 through THREE somewhat different approaches that..."?

• I recommend you remove the last sentence of the first paragraph, as it is repeated in the third paragraph.

These changes have been made. The first paragraph now reads:

"Acknowledging the uncertainty introduced by the assumptions required to implement the exponential analysis described in Section 2.2, we derive  $y_0$  through three additional, somewhat different approaches that also provide two estimates of  $\tau$  appropriate for the northeastern states."

• The third and fourth paragraphs seem to jump around a bit and are not presented in the same order as they are used later in the manuscript. Please see if this suggested re-organization describes what you intended:

Two additional approaches can approximately quantify the value of  $\tau$  in the northeastern states; both of these approaches assume that constant values of y0 and  $\tau$  are appropriate for all ODV time series included in each analysis. First, a linear fit to the initial period of decreasing ODVs provides direct information regarding the magnitude of  $\tau$  and v0. The absolute value and the time derivative of Equation 1 when evaluated at year 2000 are y0 + A and  $-A/\tau$ , respectively. Fits to two ODV time series provide four parameters ( $\tau$ , v0, A1 and A2) if the  $\tau$  and v0 values are the same for the two time series. Algebraic manipulation gives  $\tau = -\Delta$  intercept /  $\Delta$  slope, where  $\Delta$  indicates the difference in the subscripted parameter between the two linear fits, and  $y0 = (\Sigma intercept + \tau * \Sigma slope)/2$ , where  $\Sigma$  indicates the sum of the subscripted parameter from the two fits. A complication with this approach is that the linear fits to time periods of significant length give biased measures of the derivative and year 2000 value of Equation 1; however, this bias can be corrected to first order through numerical comparison of a linear fit to the selected period of the exponential fit. The second approach is described in Section 2.4 of Parrish et al. (2017a) and is adapted here to the northeastern U.S. ODV time series. It uses an iterative, non-linear regression analysis that simultaneously derives values for  $\tau$  and y0, plus the A parameter for each ODV time series included in the analysis. These two additional approaches help to constrain the uncertainty of the assumed value of  $\tau$  (21.9 vears).

Thank you very much for this suggested reorganizing; it has been incorporated exactly as suggested.

B) When the results of the method shown in Fig 13 are described in section 3.3.2, there is insufficient information to allow the reader to evaluate your "bias correction". The description of that "first order" correction "through numerical comparison of a linear fit" in section 2.3 does not shed much additional light. If it is possible to give a bit more information (or a reference?), the reader can have more confidence that your "corrections" reduce uncertainty, rather than increase it. *A 3-sentence explanation of the bias correction has been added to Section 3.3.2; see tracked changes manuscript for those changes.*

And for clarity of communication, I would suggest that "intercept" is not strictly the correct noun here, as the plots you show in Figure 13 would have fitting parameters with an intercept at Year 0. I recommend you either replot with the horizontal axis as Year Minus 2000 instead of Year, or stick to using the descriptor "value" or "absolute value at Year 2000" (e.g., revised manuscript, page 8, line 4). This will also remove any confusion about if the "intercept" for the Maine data set is to be chosen at 2000 or at 1991.

Corrected by consistently using the descriptive phrase "absolute value at Year 2000" in place of "intercept" throughout the discussion of the method and its results.

C) "Variance" is discussed often in this manuscript, and it seems that the meaning is slightly different in different contexts. On page 6, variance (in units of ppb) is

described in terms of RMSD between a dataset and its fit. But then on page 9 and in Table 1, variance has units of ppb2 and appears to be (but is never actually\*) defined as the square of the standard deviation of a dataset. Then both types are used back and forth through the remainder of the manuscript. It's especially tricky to parse at the top of page 19, where you are using both the ppb2 values (like 251 and 13.4), but are also referring to Figure 9, which gives RMSDs of 3.5 and 5.6 ppb. I don't have a nice, tidy suggestion, but I would ask that the authors take a few minutes to think about how they might make their dual-use of this word a little bit easier for the reader.

Thank you. On page 6 we have added the definition of variance as the square of the standard deviation of a dataset and we discuss that the square of the RMSD is an estimate of the variance not captured by a fit to Equation 1. We have gone through the paper to ensure that the use of "variance" is now consistent throughout.

\*It is given quite succinctly on page 6 of the Response document, but I couldn't find it in the manuscript itself. I recommend including it in the manuscript, especially since the rounding effects are just enough to make it questionable (e.g., pg 18:  $3.7 \times 3.7 = 13.69 = 13.7$ , not 13.4).

And by the way, the Figure 3 legend says 252 ppb2, not 251 ppb2.

Thank you for the very careful reading. The definition of variance is now included (see previous response). However, we cannot repair the roundoff errors without giving an undue number of significant figures in the numbers. However, we now have 252 ppb^2 in the text as well as the figure legend.

**D) And a few small suggestions:**

Thank you for the very detailed reading evidenced by your suggestions below.

• Pg 9, line 14, add a comma: "Figures 3 and 4, and averages with standard deviations..."

**Comma added.**

• Pg 13, line 15 change to: Figure 10 plots the time series of these state maximum ODVs recorded in each year with respective fits over the 2010-2017 period.

Suggested change made.

- Is that really supposed to be 2010? In Fig 10, the solid fit lines start at 2000. *Thank you. It has now been changed to 2000-2017.*
- Pg 13, line 19 change to: only the largest of the state's ODVs in a given year... {to match singular "across the state" later in the sentence"}

Suggested change made.

• Pg 13, line 27: is this "NYC urban maximum" a new subset? Or is it the same as the data which generated the red dashed line in Fig 7? This is not a big deal, but I got distracted for a while trying to figure it out.

This is a new data subset. I apologize for the confusion, but it seems necessary to include both the subset that generated the curve in Figure 7 (since this was the choice made when the subsets of data were initially chosen) and this "NYC urban maximum", which is designed to have as little

*interannual variability about the exponential fit to optimize the analysis described in Section 3.3.1.*

• And a related question: In Fig S13, Does the fact that the 1:1 line goes through NJ data points at all values of "NYC urban max" indicate that the "NYC urban max" data is actually entirely from NJ? (I understand the geography; that's not my point. If all the "reference" data is contained in the data plotted against it in Fig S13, that seems a bit circular.)

Actually many but not all "NYC urban max" are from the NJ data points. I do not think that this is circular. Rather it makes the fit in this figure more uncertain, because the method relies on the intercept of the fitted slope with the 1:1 line, which requires a slope differing from unity. The close correspondence between the "NYC urban max" and the NJ data points causes the fitted slope to approximate unity, giving an uncertain intercept. This comparison is nevertheless included for completeness.

• Pg 13, line 28, remove comma after "is selected".

Comma removed.

• Suggested clarification and nuance regarding spatial variability of ODVs on page 16: the derived US background ODV has significant variability on a continental scale. Within.... significantly smaller than in any of the western US regions, but shows no discernable spatial variability within this region. For context... NAAQS of 70 ppb. In contrast, in the northeastern US the A...

Clarifications added.

• Pg 18, top line has an extra space

Corrected.

- Pg 18, line18, strike "by" before (Fiore et al., 2015) *Corrected.*
- Pg 18, line 21: change "limit" to "limited" *Corrected.*

• Competing Interests section needs to be updated, as does the last line of the Acknowledgements.

Corrected.

• Fig 9 caption: The explanation of dashed, dotted, and y0 is awkward. Maybe try rewriting with dashed, then dotted, then y0 last.

Revised.

- Fig 14, vertical axis label: Is this really US background, not NAB? Good point; axis relabeled.
- Supplemental, pg 2, line 11: One...area *Corrected.*
- Supplemental, line 21: some sites in the NY... *Corrected.*
- Supplemental, line 32: neighboring states until 2013.

Corrected.

**• Figs S3-S10: the gray circles are a bit too faint to really see. Could they be darker?**

The lightest symbols have been darkened.

• Fig S10 legend, lower panel: green sites are called "interior" or "rural inland"

elsewhere. I recommend choosing one of those in the lower legend for consistency.

Corrected.

• Fig S16: the blue dashed line is different from all the other blue dotted lines. *Corrected.*

**Report #1, by Anonymous Referee #3**

First, I would like to thank the author for taking the time to consider many different pieces of feedback.

Thank you for this comment. Indeed the feedback has been very valuable to us in improving the manuscript.

This reviewer cannot overlook the assumptions necessary to support the implications and conclusions. In many ways, the additional mathematical and uncertainty estimates obfuscates the real problem. In my estimation, the value of the author's work is the predictive power and not the physical interpretation. Yet, the paper's implications and conclusions focus on the interpretation. The authors application and interpretation is not unlike Hubbert's peak oil curve, which likewise sparked much controversy around the pop cultural interpretation.

The similarity of your fit to Hubbert's peak oil curve has both strengths and weaknesses. Even when fitting observations, both curves equally support two competing interpretations. The first interpretation is, as the author focuses, y0 occurs when US contribution is zero. By analogy, Hubbert's peak oil curve can be interpreted as approaching zero when there is no oil is left in the US (only other countries). The second interpretation is that y0 is when US regulations have been applied to the sectors where and the extent to which it will be applied -- leaving some residual (USRES). By analogy, Hubbert's peak oil curve can be interpreted as no \*practically recoverable\* oil is left in the US. In the second interpretation, y0 is really the sum of  $y_{USB}$  +  $y_{USRES}$ . The strength of such the simple mathematical fits is their predictive power, and not the conjecture about the composition of the intercept.

Thank you for this interesting analogy, which we appreciate because we believe that it supports, rather than calls into question, our analysis and its interpretation. The prediction of the temporal evolution toward ultimate total possible petroleum production is analogous to the prediction of the temporal evolution toward complete elimination of the anthropogenic contributions to ozone in an urban area. However, Hubbert's peak oil hypothesis suffers from a major difficulty - the total recoverable petroleum reserves is quite uncertain, and is subject to technological or exploration driven "surprises". This quantity has increased by a significant factor over past decades; Wikipedia reports that "The ratio between proven oil reserves and current production has constantly improved, passing from 20 years in 1948 to 35 years in 1972 and reaching about 40 years in 2003." This is primarily due to improved petroleum recovery technology, and discovery of some new oil fields. When the ultimate total possible petroleum production (i.e., the proven reserves) is not accurately known, prediction of the temporal evolution toward that ultimate limit is very likely to be inaccurate.

In contrast, elimination of the anthropogenic ozone contribution requires the reduction of anthropogenic emissions of NOx emissions to zero. (Reduction of VOC emissions may also be important, but the presence of important quantities of biogenic VOCs implies that ultimately the

NOx emissions must be reduced to zero to completely eliminate the anthropogenic ozone contribution.) We do know very accurately the "proven NOx emission reserves" simply from measurements of ambient NOx or NOy measurements. For example, Pollack et al. (2013) show that NOx concentrations in California's South Coast Air Basin have decreased by a factor of ~4 between 1960 and 2010. Thus, we are at least about 3/4 of the way toward complete elimination of all anthropogenic NOx emissions in at least that area (and similar relative decreases are likely in all U.S. regions, although we are not aware of similar analyses for other regions). Since ambient NOx concentrations in rural areas of the U.S. are small with respect to urban areas, there is no possibility of discovering unknown reservoirs of NOx emissions (and certainly no likelihood of improving technology increasing NOx emissions.) In summary, our predictive task is much less vulnerable to surprises than was Hubbert's.

There is value in the predictive power of the exponential model proposed. The exponential model does fit recent data and one may conjecture that it will fit near-term future reductions. As the author points out, the curve may represent diminishing reductions of US contribution. The predictive power is independent of the physical interpretation of the composition of y0. Parrish and Ennis project an attainment date of 2021 and that prediction is not predicated upon the physical interpretation y0, which does seem valuable -- as was their prediction for SoCal. *Thank you for these supportive thoughts.*

Regarding your response to my critique of fitting to data for Massachusetts (old Fig 7, new Fig 6). If in 2003 you been applying your exponential model, you would have only had the historical record to that date. My visual interpretation of the historical record between 1984-2003 is that your curve would have had some limited explanatory power. If I am not much mistaken, your y0 would have been substantially higher (80 ppb?) because the record as a local plateau between 1995 and 2003. This illustrates clearly that the physical interpretation of y0 predicted from that segment (1984-2003) would have been better interpreted as  $y_{USB} + y_{USRES}$ . For the 2003 inflection point, it is easy to guess that is caused by the NOx SIP call. How can we justify assuming there are not other controllable sectors that would produce a similar inflection in the future?

Thank you for this clarification. Your visual interpretation is at least qualitatively correct. Figure 1 below reproduces Figure 3 from the previous response with the fits that you specify above replacing the earlier fits. The  $y_0$  derived from the 1984-2003 record is substantially higher ( $68 \pm 9$  ppb), at least partly due to the reason you suggest. The  $y_0$  derived from the total 1984-2017 record is also is substantially higher ( $60 \pm 4$  ppb).

Figure 1. Analysis of the Massachusetts coastal data set (violet points from Figure 6 of the revised manuscript) over the time periods indicated in the figure annotation. The fits to Equation 1 for the different time periods are indicated by the superimposed

periods are indicated by the superimposed, color-coded curves

We struggled with this issue throughout the analysis. Comparisons of all of the data sets, including the urban and rural areas of the northern U.S. as well as other regions of the country, indicated that the 2000-2017 period does give consistent results over nearly two decades of measurements for all data sets considered. This choice is thoroughly discussed in the manuscript.

The additional sections with all the analysis suggest that this fundamental problem of interpretation can be solved by more math. The largest uncertainty in  $y_{USB}$  is how well it is approximated by  $y_0$ . If  $y_{USRES}$  is large, then the uncertainty is swamped by the interpretation. If  $y_{USRES}$  is small, then the interpretation is reasonable.  $y_{USRES}$  is some combination of the technologically challenging reductions and, as illustrated by the previous paragraph, the sectors that may not yet have been the focus of controls. No effort is made to consider or estimate the magnitude of US emission sectors that are not decreasing. In response to a previous revision, this point was made by highlighting specific sectors that are not decreasing. No amount of regression uncertainty analysis will address the uncertainty associated with a physical interpretation.

As discussed above, we do have ambient NOx and VOC measurements that provide guidance regarding the magnitude of  $y_{USRES}$ , and all indications are that it is relatively small with the possible exceptions of intense, fertilized agricultural regions (e.g., California's Imperial Valley) and very densely populated urban centers, which may lead to the Connecticut and NYC ODV deviations from exponential decreases, as we discuss in the paper.

In my view, this manuscript has valuable analysis mixed with what seems like an arbitrary interpretation. The author has made token statements that allow for an alternative interpretation, but the conclusions and implications are so intertwined with the interpretation and those have not changed. This manuscript would be far easy to accept if the conclusions and implications it promotes/asserts were not dependent on an arbitrary physical interpretation of a parameter with an unknowable composition. Given that there are two equally likely physical interpretations (y0 =  $y_{USB}$  or  $y_0 = y_{USB} + y_{USRES}$ ), this reviewer cannot support a manuscript whose conclusions and implications are based on a seemingly arbitrary choice between the two. *We believe that the two alternative physical interpretations are not equally likely. As discussed above and in much of the previous responses to reviews of this paper, all indications are that y\_{USRES} is generally quite small, with the possible exceptions discussed in the response to the previous comment. Of course, only future monitoring results can definitively resolve this issue.*

**References:**

Pollack, I. B., T. B. Ryerson, M. Trainer, J. A. Neuman, J. M. Roberts, and D. D. Parrish (2013), Trends in ozone, its precursors, and related secondary oxidation products in Los Angeles, California: A synthesis of measurements from 1960 to 2010, J. Geophys. Res. Atmos., 118, 5893–5911, doi:10.1002/jgrd.50472.

[revised manuscript text omitted]

-